# Neurosymbolic Diffusion Models

**Emile van Krieken**[1]     **Pasquale Minervini**[1,2,*]     **Edoardo Ponti**[1,*]     **Antonio Vergari**[1,*]
[1]School of Informatics, University of Edinburgh          [2]Miniml.AI
e.van.krieken@vu.nl, {p.minervini, eponti, avergari}@ed.ac.uk

## Abstract

Neurosymbolic (NeSy) predictors combine neural perception with symbolic reasoning to solve tasks like visual reasoning. However, standard NeSy predictors assume conditional independence between the symbols they extract, thus limiting their ability to model interactions and uncertainty — often leading to overconfident predictions and poor out-of-distribution generalisation. To overcome the limitations of the independence assumption, we introduce *neurosymbolic diffusion models* (NESYDMS), a new class of NeSy predictors that use discrete diffusion to model dependencies between symbols. Our approach reuses the independence assumption from NeSy predictors at each step of the diffusion process, enabling scalable learning while capturing symbol dependencies and uncertainty quantification. Across both synthetic and real-world benchmarks — including high-dimensional visual path planning and rule-based autonomous driving — NESYDMS achieve state-of-the-art accuracy among NeSy predictors and demonstrate strong calibration.

## 1   Introduction

Neurosymbolic (NeSy) methods aim to develop reliable and interpretable AI systems by augmenting neural networks with symbolic reasoning [25, 26, 77]. In particular, *probabilistic neurosymbolic predictors* [52, 54, 57, 80] learn neural networks that extract high-level symbols, also called *concepts*, from raw inputs. These concepts are latent variables used in interpretable symbolic programs to reason and predict output labels. However, recent work highlights that the reliability of NeSy predictors is not guaranteed, especially under certain common architectural choices.

More specifically, in many real-world settings, NeSy predictors fail silently: they can learn the wrong concepts while achieving high accuracy on output labels [22, 27]. This issue arises when the data and program together admit multiple concept assignments that are indistinguishable [54, 56]. How do we design NeSy predictors that handle this ambiguity? Marconato et al. [55] argued that NeSy predictors should express uncertainty over the concepts that are consistent with the data. Then, uncertainty can guide user intervention, inform trust, or trigger data acquisition when the model is uncertain [55].

However, most existing NeSy predictors cannot properly model this uncertainty, as they rely on neural networks that assume *(conditional) independence* between concepts [10, 80, 86]. While this assumption enables efficient probabilistic reasoning [6, 73, 80, 86], it also prevents these NeSy predictors from being aware of concept ambiguity and thus reliably generalising out-of-distribution [39, 79]. Therefore, designing expressive, scalable and reliable NeSy predictors is an open problem.

To fill this gap, we design *neurosymbolic diffusion models* (NESYDMS). NESYDMS are the first class of diffusion models that operate over the concepts of a NeSy predictor in conjunction with symbolic programs. In theory, discrete diffusion models [9, 68] are particularly suited for NeSy predictors, as each step of their denoising process involves predicting a discrete distribution that fully factorises.

---

*: Shared supervision.

Code is available at https://github.com/HEmile/neurosymbolic-diffusion.

39th Conference on Neural Information Processing Systems (NeurIPS 2025).

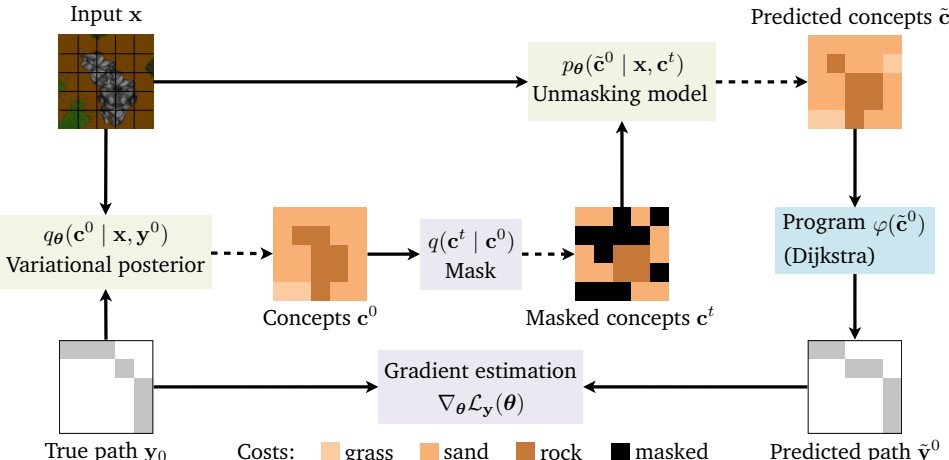

Figure 1: **NESYDMs integrate masked diffusion models (orange boxes) with symbolic programs (blue box)** to learn to predict the minimum cost path in a visual path-planning task. A variational posterior (Section 3.3) first obtains a candidate concept $\mathbf{c}^0$, that represents the costs of traversing each cell of the grid. Then, we partially mask $\mathbf{c}^0$ using the masking process $q(\mathbf{c}^s \mid \mathbf{c}^0)$ to obtain masked concepts $\mathbf{c}^{\frac{1}{2}}$. We feed this to the discrete diffusion model's *unmasking model* $p_{\boldsymbol{\theta}}(\tilde{\mathbf{c}} \mid \mathbf{x}, \mathbf{c}^{\frac{1}{2}})$ to predict the unmasked concepts $\tilde{\mathbf{c}}^0$. We use the symbolic program $\varphi$, which we choose as Dijkstra's algorithm, to map the predicted concepts $\tilde{\mathbf{c}}^0$ to the predicted path $\tilde{\mathbf{y}}^0$. Finally, we use gradient estimation to update the parameters of the unmasking model. Dotted arrows denote samples from a distribution.

We use this *local* independence assumption to profit from the insights and machinery of classical NeSy predictors, while modelling concepts as dependent entities *globally*. In practice, designing a diffusion process for NeSy predictors is highly non-trivial, as it requires dealing with a symbolic program and marginalising over all possible concepts, a task that is intractable in general. We show how to solve both aspects effectively by devising a novel continuous-time loss function for diffusion that incorporates symbolic programs, for which training scales gracefully.

**Contributions.** After discussing the background on NeSy predictors and (masked) diffusion models in Section 2, we **(c1)** introduce NESYDMs in Section 3, a class of scalable NeSy predictors that model concept dependencies by formalising a *masked diffusion process* [68]. Then in Section 3.2, we **(c2)** derive a principled loss function for NESYDMs and present an efficient gradient estimator for training it. To derive this loss, we prove that the continuous-time losses of masked diffusion models extend to non-factorised distributions. Finally, in Section 4, we **(c3)** empirically show that NESYDMs are (i) both calibrated and performant on tasks from the RSBench suite of visual reasoning problems [11] while (ii) scaling beyond the state-of-the-art on the complex visual path-planning task [64].

## 2 Background

### 2.1 Neurosymbolic predictors

We aim to learn a parametrised predictive model $p_{\boldsymbol{\theta}}(\mathbf{y} \mid \mathbf{x})$ that maps high-dimensional inputs $\mathbf{x}$ to $Y$-dimensional discrete labels $\mathbf{y} \in [V_{\mathbf{y}}]^Y$, where each label can take a value in $[V_{\mathbf{y}}] = \{1, 2, \ldots V_{\mathbf{y}}\}$. A typical *(probabilistic) NeSy predictor* implements $p_{\boldsymbol{\theta}}(\mathbf{y} \mid \mathbf{x})$ by first (i) using a *concept extractor*, i.e., a neural network $p_{\boldsymbol{\theta}}(\mathbf{c} \mid \mathbf{x})$ that maps the input $\mathbf{x}$ to a $C$-dimensional vector of symbolic *concepts* $\mathbf{c} \in [V_{\mathbf{c}}]^C$, i.e., discrete variables encoding high-level information that can take $V$ values.[1] Then, (ii) the NeSy predictor maps concepts $\mathbf{c}$ through a program $\varphi : [V_{\mathbf{c}}]^C \to [V_{\mathbf{y}}]^Y$ to obtain output predictions $\hat{\mathbf{y}}$. As usual in NeSy [10, 43, 52, 80], we only assume access to training data for input-output pairs $(\mathbf{x}, \mathbf{y})$ but no labelled data for concepts $\mathbf{c}$, i.e., concepts $\mathbf{c}$ are *latent variables*. Formally, we define the predictor $p_{\boldsymbol{\theta}}(\mathbf{y} \mid \mathbf{x})$ by marginalising over all concepts $\mathbf{c} \in V_{\mathbf{c}}{}^C$ that are consistent with

---

[1]For simplicity of presentation, we assume that the number of possible values $V$ is the same for both concepts and labels, but this is not necessary for the paper.

the output $\mathbf{y}$, summing their probability masses:

$$p_{\boldsymbol{\theta}}(\mathbf{y} \mid \mathbf{x}) := \sum_{\mathbf{c}} p_{\boldsymbol{\theta}}(\mathbf{c} \mid \mathbf{x}) \mathbb{1}[\varphi(\mathbf{c}) = \mathbf{y}]. \tag{1}$$

The equation above is also known as computing a conditional *weighted model count* (WMC), and it is central to several probabilistic neurosymbolic methods [6, 43, 52, 80, 86].

**Example 2.1** ([65]). Consider the visual path-planning task in Fig. 1 where the task is to predict a minimum cost path $\mathbf{y}$ from the top-left corner to the bottom-right corner of the visual map $\mathbf{x}$. $\mathbf{y}$ is encoded as a binary matrix, where cells traversed form a path. A neural network extracts concepts that represent discrete costs $\mathbf{c}$ for each cell on the grid, then a search algorithm $\varphi(\mathbf{c})$, like Dijkstra, is used to find the shortest path $\mathbf{y}$ according to costs $\mathbf{c}$.

**Reasoning shortcuts.** Recent work proved NeSy predictors are susceptible to *reasoning short-cuts* [RSs; 56], which is when a model $p_{\boldsymbol{\theta}}(\mathbf{y} \mid \mathbf{x})$ learns to predict the output labels $\mathbf{y}$ correctly given the input $\mathbf{x}$, but incorrectly maps inputs to concepts $\mathbf{c}$. Since we cannot catch RSs on the training data, it can dramatically harm model performance on unseen data [53]. Mitigating RSs is challenging and potentially costly [54, 56]. However, models can be made *aware of their RS* by properly expressing uncertainty over all concepts that are consistent with the input-output mapping, improving reliability and generalisation [39, 54, 55]. Then we can, for example, deploy NeSy predictors in an active learning setting where uncertain concepts are queried for extra labelling.

**Example 2.2.** Consider an input $\mathbf{x}$ containing two MNIST digits that are either 0 or 1. The unseen concepts $\mathbf{c}$ are the digits, and $\varphi(\mathbf{c})$ returns 1 if the two digits are different, otherwise 0. A neural concept extractor $p_{\boldsymbol{\theta}}(\mathbf{c} \mid \mathbf{x})$ that maps MNIST digits of 0 to 1s and MNIST digits of 1s to 0s will perfectly fit the input-output mapping.

The configuration in Example 2.2 maximises Eq. 1 without learning the ground-truth concepts. Given only input-output pairs, it is not possible to distinguish this RS from the correct input-concept mapping. Instead, given ground-truth concepts $\mathbf{c}^* = (0, 1)$, an RS-aware model would assign some belief to both options $(0, 1)$ and $(1, 0)$.

**Independence assumption and its limitations.** Unfortunately, in practice, the vast majority of NeSy predictors make an architectural assumption that prevents RS awareness: the *conditional independence* of concepts $\mathbf{c}$ given inputs $\mathbf{x}$ [43, 80, 86]. Formally, this assumption implies that $p_{\boldsymbol{\theta}}(\mathbf{c} \mid \mathbf{x})$ in Eq. 1 factorises as $\prod_{i=1}^{C} p_{\boldsymbol{\theta}}(c_i \mid \mathbf{x})$. NeSy predictors use this assumption to perform efficient probabilistic reasoning via WMC solvers and *knowledge compilation* techniques [15, 18, 63], or by developing efficient approximation algorithms [73, 80].

Recent work proved that such models cannot simultaneously represent the relevant uncertainty over different concepts while maximising Eq. 1 [39]. To see why, consider Example 2.2, with true concepts $\mathbf{c}^* = (0, 1)$. The only maximisers of Eq. 1 for the independent model are to either deterministically return $(0, 1)$ or $(1, 0)$ [39, 79]. However, there is no maximiser that can simultaneously assign probability mass to *both cases*, meaning independent models cannot be RS-aware. To overcome this limitation, we should design a NeSy predictor that can express dependencies between concepts, which we address next.

## 2.2 Which expressive model class for NeSy?

Previous work on NeSy predictors without the independence assumption explored mixture models and their generalisation as probabilistic circuits [6, 16]. An example is BEARS [55], which is specifically designed for RS-awareness. A related approach is to add extra variables and constraints to the WMC. This can, for instance, be done using a probabilistic programming language [43, 51]. However, these methods require (i) compiling the program into a logic circuit via knowledge compilation and (ii) ensuring the probabilistic circuit is compatible with this logic circuit [83]. The first step can require exponential time in the worst case, and as such scaling to high-dimensional spaces can be challenging [4, 80]. Furthermore, these methods require the neural concept extractor to predict many more additional parameters for the different mixture components.

Alternatively, autoregressive models are a common type of expressive model, but using these in NeSy predictors based on Eq. 1 is computationally hard, as the marginalisation over concepts does not commute with autoregressive conditioning [3, 5]. While this limitation also holds for diffusion

models, they *do* use a conditional independence assumption *locally* at every denoising step. This local assumption is sufficient to encode *global* dependencies. Furthermore, the locality allows us to design neural models that predict only $C$ parameters, just like NeSy predictors with the independence assumption. Thus, we use *masked diffusion models* [68] that achieve expressiveness by iteratively unmasking a discrete sample. We discuss in Section 3 how to extend their local independence assumption to realise NeSy predictors.

**Masked diffusion models.** Diffusion models encode an expressive joint distribution over concepts $\mathbf{c}$ by defining a *forward process* that a neural network modelling a *reverse process* will learn to invert. As our concepts are symbolic, we need a diffusion process for discrete data [9, 90]. We choose masked diffusion models (MDMs) [68, 72], a type of discrete diffusion model with promising results on language modelling [61, 89] and reasoning [88]. MDMs allow us to derive a principled loss using the program $\varphi$ (Section 3.2) and to develop scalable approximations (Section 3.4). We first review MDMs in their vanilla form, i.e., to model an unconditional distribution over concepts, $p_{\boldsymbol{\theta}}(\mathbf{c})$.

MDMs consider a continuous time diffusion process [9, 14], where the forward process gradually masks dimensions of a data point $\mathbf{c}^0$ into a partially masked data point $\mathbf{c}^t \in [V_{\mathbf{c}} + 1]^C$ at time steps $t \in [0, 1]$. We extend the vocabulary size to include a placeholder $\mathrm{m} = V_{\mathbf{c}} + 1$ for masked dimensions. The data point becomes fully masked as $\mathbf{c}^1 = \mathbf{m} = [\mathrm{m}, \dots, \mathrm{m}]^\top$ at time step 1. More formally, for $0 \le s < t \le 1$, the forward process $q$ masks a partially masked concept $\mathbf{c}^s$ into $\mathbf{c}^t$ with

$$q(\mathbf{c}^t \mid \mathbf{c}^s) = \prod_{i=1}^{C} \frac{\alpha_t}{\alpha_s} \mathbb{1}[c_i^t = c_i^s] + \left(1 - \frac{\alpha_t}{\alpha_s}\right) \mathbb{1}[c_i^t = \mathrm{m}], \tag{2}$$

where $\alpha : [0, 1] \to [0, 1]$ is a strictly decreasing noising schedule with $\alpha_0 = 1$ and $\alpha_1 = 0$. $q(\mathbf{c}^t \mid \mathbf{c}^s)$ masks each dimension with probability $1 - \frac{\alpha_t}{\alpha_s}$, leaving it unchanged otherwise. Importantly, once masked, a dimension remains masked. MDMs learn to *invert* the forward process $q(\mathbf{c}^t \mid \mathbf{c}^s)$ using a trained reverse process $p_{\boldsymbol{\theta}}(\mathbf{c}^s \mid \mathbf{c}^t)$. The reverse process starts at a fully masked input $\mathbf{c}^1 = \mathbf{m}$ at time step 1, and gradually unmasks dimensions by assigning values in $\{1, ..., V_{\mathbf{c}}\}$.

The reverse process $p_{\boldsymbol{\theta}}(\mathbf{c}^s \mid \mathbf{c}^t)$ is usually parameterised with conditionally independent *unmasking models* $p_{\boldsymbol{\theta}}(\tilde{\mathbf{c}}^0 \mid \mathbf{c}^t) = \prod_{i=1}^{C} p_{\boldsymbol{\theta}}(\tilde{c}_i^0 \mid \mathbf{c}^t)$ that predict completely unmasked data $\tilde{\mathbf{c}}^0$ given (partially) masked versions $\mathbf{c}^t$. Then, MDMs remask some dimensions using the so-called *reverse posterior* $q(\mathbf{c}^s \mid \mathbf{c}^t, \mathbf{c}^0 = \tilde{\mathbf{c}}^0)$ (see more details in Eq. 10 in Section A):

$$p_{\boldsymbol{\theta}}(\mathbf{c}^s \mid \mathbf{c}^t) := \sum_{\tilde{\mathbf{c}}^0} p_{\boldsymbol{\theta}}(\tilde{\mathbf{c}}^0 \mid \mathbf{c}^t) \, q(\mathbf{c}^s \mid \mathbf{c}^t, \mathbf{c}^0 = \tilde{\mathbf{c}}^0), \tag{3}$$

The standard loss function masks $\mathbf{c}^0$ partially to obtain $\mathbf{c}^t$, and then uses the conditionally independent unmasking model $p_{\boldsymbol{\theta}}(\tilde{\mathbf{c}}^0 \mid \mathbf{c}^t)$ to attempt to reconstruct $\mathbf{c}^0$. This loss function requires that $p_{\boldsymbol{\theta}}(\tilde{\mathbf{c}}^0 \mid \mathbf{c}^t)$ implements the *carry-over unmasking assumption*, meaning it should assign a probability of 1 to the values of previously unmasked dimensions. We provide additional background on MDMs in Section A. Next, we discuss how to design novel MDMs tailored for NeSy prediction.

## 3 Neurosymbolic Diffusion Models

To overcome the limitations of the independence assumption haunting NeSy predictors, our neurosymbolic diffusion models (NESYDMS) use MDMs to learn an expressive distribution over concepts and labels while retaining this assumption locally, enabling scaling. To develop NESYDMS, we extend MDMs by (i) conditioning on the input $\mathbf{x}$, (ii) acting on both concepts $\mathbf{c}$ and outputs $\mathbf{y}$, treating concepts as latent variables and (iii) providing differentiable feedback through the program $\varphi$. We first define this model in Section 3.1 and then derive a principled loss in Section 3.2. We discuss how to optimise this loss in Sections 3.3 and 3.4, and finish by discussing inference in Section 3.5. Finally, Fig. 1 provides an overview of the loss computation of NESYDMS.

### 3.1 Model setup

We define NESYDMS using a conditionally independent unmasking model $p_{\boldsymbol{\theta}}(\tilde{\mathbf{c}}^0 \mid \mathbf{c}^t, \mathbf{x})$ and a program $\varphi$ that maps concepts to outputs. We use forward processes for both the concepts $q(\mathbf{c}^t \mid \mathbf{c}^s)$ and the outputs $q(\mathbf{y}^t \mid \mathbf{y}^s)$, each defined as in Eq. 2. The *concept reverse process* $p_{\boldsymbol{\theta}}(\mathbf{c}^s \mid \mathbf{c}^t, \mathbf{x})$ is

parameterised as in Eq. 3 with a conditional *concept unmasking model* $p_{\boldsymbol{\theta}}(\tilde{\mathbf{c}}^0 \mid \mathbf{c}^s, \mathbf{x})$, and the *output reverse process* $p_{\boldsymbol{\theta}}(\mathbf{y}^s \mid \mathbf{c}^s, \mathbf{y}^t, \mathbf{x})$ is parameterised by reusing the concept unmasking model:

$$p_{\boldsymbol{\theta}}(\mathbf{y}^s \mid \mathbf{c}^s, \mathbf{y}^t, \mathbf{x}) := \sum_{\tilde{\mathbf{c}}^0} p_{\boldsymbol{\theta}}(\tilde{\mathbf{c}}^0 \mid \mathbf{c}^s, \mathbf{x}) q(\mathbf{y}^s \mid \mathbf{y}^t, \tilde{\mathbf{y}}^0 = \varphi_{\mathbf{y}^t}(\tilde{\mathbf{c}}^0)). \tag{4}$$

$p_{\boldsymbol{\theta}}(\mathbf{y}^s \mid \mathbf{c}^s, \mathbf{y}^t, \mathbf{x})$ takes the concept unmasking model and marginalises over all concepts $\tilde{\mathbf{c}}^0$ that are consistent with the partially masked output $\mathbf{y}^s$. To implement the carry-over unmasking assumption, we use $\varphi_{\mathbf{y}^t}$ to refer to a variation of the program $\varphi$ that always returns $y_i^t$ if dimension $i$ is unmasked in $\mathbf{y}^t$. We refer to Section D.1 for details. The neural network for the concept unmasking model $p_{\boldsymbol{\theta}}(\tilde{\mathbf{c}}^0 \mid \mathbf{c}^t, \mathbf{x})$ can be readily adapted from NeSy predictors as defined in Eq. 1 by additionally conditioning the neural network $p_{\boldsymbol{\theta}}(\mathbf{c} \mid \mathbf{x})$ on the currently unmasked concepts $\mathbf{c}^t$.

Since we do not have direct access to ground-truth concepts $\mathbf{c}^0$, we will use a variational setup and derive a lower-bound for the intractable data log-likelihood $p_{\boldsymbol{\theta}}(\mathbf{y}^0 \mid \mathbf{x})$ (fully defined in Eq. 45). In particular, we use a variational distribution $q_{\boldsymbol{\theta}}(\mathbf{c}^0 \mid \mathbf{y}^0, \mathbf{x})$ that shares parameters $\boldsymbol{\theta}$ with the MDM to approximate the posterior $p_{\boldsymbol{\theta}}(\mathbf{c}^0 \mid \mathbf{y}^0, \mathbf{x})$. To implement this, we repurpose our concept unmasking model $p_{\boldsymbol{\theta}}(\mathbf{c}^s \mid \mathbf{c}^t, \mathbf{x})$ with the controlled generation method from [29], which we describe in Section 3.3. We provide more details and a full derivation of the log-likelihood in Section D.1.

## 3.2 Loss function

We next derive a NELBO for NESYDMS. Intuitively, we define the NESYDM reverse process over $T$ discrete steps, and then consider the data log-likelihood as $T$ goes to infinity, giving a NELBO for a continuous-time process. This NELBO will be the base for the loss function used to train NESYDMS.

**Theorem 3.1.** *Let $p_{\boldsymbol{\theta}}(\tilde{\mathbf{c}}^0 \mid \mathbf{c}^t, \mathbf{x})$ be a concept unmasking model, $\varphi : [V_{\mathbf{c}}]^C \to [V_{\mathbf{y}}]^Y$ a given program, $q_{\boldsymbol{\theta}}(\mathbf{c}^0 \mid \mathbf{y}^0, \mathbf{x})$ a variational distribution, and $\alpha_t$ a noising schedule. Then, we have that the data log-likelihood as $T \to \infty$ is bounded as $\lim_{T\to\infty} -\log p_{\boldsymbol{\theta}}^{\mathrm{NESYDM}}(\mathbf{y}^0 \mid \mathbf{x}) \leq \mathcal{L}_{\mathrm{NESYDM}}$, where*

$$\mathcal{L}_{\mathrm{NESYDM}} = \mathbb{E}_{t\sim[0,1], q_{\boldsymbol{\theta}}(\mathbf{c}^0 \mid \mathbf{x}, \mathbf{y}^0), q(\mathbf{c}^t \mid \mathbf{c}^0)} \left[ \underbrace{\frac{\alpha_t'}{1-\alpha_t} \sum_{i=1}^{C} \log p_{\boldsymbol{\theta}}(\tilde{c}_i^0 = c_i^0 \mid \mathbf{c}^t, \mathbf{x})}_{\mathcal{L}_{\mathbf{c}}:\ \text{concept unmasking loss}} \right.$$

$$\left. + \underbrace{\alpha_t' \sum_{i=1}^{Y} \log \sum_{\tilde{\mathbf{c}}^0} p_{\boldsymbol{\theta}}(\tilde{\mathbf{c}}^0 \mid \mathbf{c}^t, \mathbf{x}) \mathbb{1}[\varphi(\tilde{\mathbf{c}}^0)_i = y_i^0]}_{\mathcal{L}_{\mathbf{y}}:\ \text{output unmasking loss}} \right] - \underbrace{\mathrm{H}[q_{\boldsymbol{\theta}}(\mathbf{c}^0 \mid \mathbf{y}^0, \mathbf{x})]}_{\mathcal{L}_{\mathrm{H}[q]}:\ \text{variational entropy}} \tag{5}$$

We provide a derivation of this NELBO in Section D.2. This NELBO has three components:

• The ***concept unmasking loss*** $\mathcal{L}_{\mathbf{c}}$ is like the unmasking loss used in MDMs (Eq. 14). Since we do not have access to the ground-truth concept $\mathbf{c}^0$, we sample $\mathbf{c}^0$ from the variational distribution $q_{\boldsymbol{\theta}}(\mathbf{c}^0 \mid \mathbf{y}^0, \mathbf{x})$ and ask the model to reconstruct $\mathbf{c}^0$ from a partially masked version $\mathbf{c}^t \sim q(\mathbf{c}^t \mid \mathbf{c}^0)$.

• The ***output unmasking loss*** $\mathcal{L}_{\mathbf{y}}$ is a sum of $Y$ weighted model counts (WMC) like in Eq. 1, one for each dimension $i$ of the output $\mathbf{y}^0$. Unlike Eq. 1, $\mathcal{L}_{\mathbf{y}}$ weights concepts using the concept unmasking model $p_{\boldsymbol{\theta}}(\tilde{\mathbf{c}}^0 \mid \mathbf{c}^t, \mathbf{x})$ that is conditioned on partially masked concepts $\mathbf{c}^t$. Importantly, we use conditionally independent concept unmasking models, meaning we can use standard techniques in the NeSy literature to compute this loss efficiently. Section B provides additional analysis.

• The ***variational entropy*** $\mathcal{L}_{\mathrm{H}[q]}$ is maximised to encourage the variational distribution to cover all concepts $\mathbf{c}^0$ that are consistent with the input $\mathbf{x}$ and output $\mathbf{y}^0$.

To derive the NELBO, we had to prove a new theorem that extends the standard MDM NELBO to *non*-factorised unmasking models $p_{\boldsymbol{\theta}}(\tilde{\mathbf{c}}^0 \mid \mathbf{c}^t)$ (Section C), which can be an interesting result for future MDM architectures even outside NeSy predictors. We need this result because, unlike the concept reverse process, the output reverse process $p_{\boldsymbol{\theta}}(\mathbf{y}^s \mid \mathbf{c}^s, \mathbf{y}^t, \mathbf{x})$ in Eq. 47 does not factorise, and we cannot naively apply the standard MDM NELBO given in Eq. 14.

## 3.3 Variational posterior

To compute the NESYDM NELBO, we require a variational distribution $q_{\boldsymbol{\theta}}(\mathbf{c}^0 \mid \mathbf{y}^0, \mathbf{x})$ to sample likely concepts $\mathbf{c}^0$ that are consistent with the ground-truth output $\mathbf{y}^0$. We achieve this by adapting the sampling algorithm described in Section 3.5 using a concept unmasking model $p_{\boldsymbol{\theta}}(\tilde{\mathbf{c}}^0 \mid \mathbf{c}^t, \mathbf{x})$ that depends on the output $\mathbf{y}^0$ and the program $\varphi$:

$$q_{\boldsymbol{\theta}}(\tilde{\mathbf{c}}^0 \mid \mathbf{c}^t, \mathbf{y}^0, \mathbf{x}) := \frac{p_{\boldsymbol{\theta}}(\tilde{\mathbf{c}}^0 \mid \mathbf{c}^t, \mathbf{x}) \mathbb{1}[\varphi(\tilde{\mathbf{c}}^0) = \mathbf{y}^0]}{\mathcal{Z}(\mathbf{c}^t, \mathbf{x}, \mathbf{y}^0)}, \tag{6}$$

where $\mathcal{Z}(\mathbf{c}^t, \mathbf{x}, \mathbf{y}^0)$ is a normalising constant. This redefines the standard unmasking process from Eq. 3 by only considering valid $\tilde{\mathbf{c}}^0$. Unfortunately, sampling from $p_{\boldsymbol{\theta}}(\tilde{\mathbf{c}}^0 \mid \mathbf{c}^t, \mathbf{x}, \mathbf{y}^0)$ is NP-hard [33, 49]. However, if we have a tractable representation of the program $\varphi$, e.g., a polysize circuit as the output of a knowledge compilation step [63], then we can represent $q_{\boldsymbol{\theta}}(\tilde{\mathbf{c}}^0 \mid \mathbf{c}^t, \mathbf{y}^0, \mathbf{x})$ compactly and exactly sample from it [6]. Without access to such a circuit, we can instead use a relaxation of the constraint similar to [29]. Let $r_{\beta}(\tilde{\mathbf{c}}^0 \mid \mathbf{y}^0) = \exp(-\beta \sum_{i=1}^{Y} \mathbb{1}[\varphi(\tilde{\mathbf{c}}^0)_i \neq y_i^0])$, where $\beta > 0$ and $\beta \to \infty$ approaches the hard constraint. At each step in the reverse process, we resample to approximately obtain samples from $q_{\boldsymbol{\theta}}^{\beta}(\tilde{\mathbf{c}}^0 \mid \mathbf{c}^t, \mathbf{x}, \mathbf{y}^0) \propto p_{\boldsymbol{\theta}}(\tilde{\mathbf{c}}^0 \mid \mathbf{c}^t, \mathbf{x}) r_{\beta}(\tilde{\mathbf{c}}^0 \mid \mathbf{y}^0)$ [29]. This procedure may sample concepts $\tilde{\mathbf{c}}^0$ that are inconsistent with $\mathbf{y}^0$, but prefers samples that reconstruct more dimensions of $\mathbf{y}^0$. We find that reasonably large $\beta > 10$ works in our experiments. In practice, this effectively samples $K$ times from $p_{\boldsymbol{\theta}}(\tilde{\mathbf{c}}^0 \mid \mathbf{c}^t, \mathbf{x})$ and chooses the sample that violates the fewest constraints. See Section F.1 for details.

## 3.4 Loss optimisation and scalability

Next, we describe how we optimise the NESYDM NELBO $\mathcal{L}_{\text{NESYDM}}$ using gradient descent. We design a gradient estimation algorithm that scales to large reasoning problems by approximating intractable computation. Note that, given samples $\mathbf{c}^0, \mathbf{c}^t \sim q_{\boldsymbol{\theta}}(\mathbf{c}^0 \mid \mathbf{x}, \mathbf{y}^0) \, q(\mathbf{c}^t \mid \mathbf{c}^0)$, the empirical **concept unmasking loss** $\mathcal{L}_{\mathbf{c}}$ is tractable, so we only discuss how to backpropagate through the **output unmasking loss** $\mathcal{L}_{\mathbf{y}}$ and the **variational entropy** $\mathcal{L}_{\text{H}[q]}$.

Computing the **output unmasking loss** $\mathcal{L}_{\mathbf{y}}$ involves computing multiple WMCs, which are #P-hard. One option is to compute each WMC exactly using circuits obtained via knowledge compilation [37, 52, 86]. However, to ensure scalability, we develop a sampling-based approach that approximates the WMC gradients [73]. In particular, we use a REINFORCE-based gradient estimator [59], the REINFORCE Leave-One-Out (RLOO) estimator [1, 38]. RLOO is similar to the popular GRPO algorithm [71] while being unbiased. Furthermore, RLOO allows for flexible tradeoffs between variance and computation constraints by choosing the number of samples.

However, methods like RLOO can fail for problems where the probability of getting a sample $\tilde{\mathbf{c}}^0$ consistent with $\mathbf{y}^0$ is very low: when we only sample inconsistent concepts $\tilde{\mathbf{c}}^0$, RLOO does not provide any gradient signal. However, the output unmasking loss is subtly different, as $\mathcal{L}_{\mathbf{y}}$ gives a signal for each of the dimensions of $\mathbf{y}^0$ independently. This helps structure the search for consistent concepts $\tilde{\mathbf{c}}^0$ by decomposing the problem into $Y$ independent subproblems [8, 80]. More precisely, given a time step $t \in [0, 1]$, samples $\mathbf{c}^0, \mathbf{c}^t \sim q_{\boldsymbol{\theta}}(\mathbf{c}^0, \mathbf{c}^t \mid \mathbf{y}^0, \mathbf{x})$ and samples $\tilde{\mathbf{c}}_1^0, \ldots, \tilde{\mathbf{c}}_S^0 \sim p_{\boldsymbol{\theta}}(\tilde{\mathbf{c}}^0 \mid \mathbf{c}^t, \mathbf{x})$, we use:

$$\nabla_{\boldsymbol{\theta}} \mathcal{L}_{\mathbf{y}} \approx \alpha_t' \sum_{i=1}^{Y} \frac{1}{\mu_i(S-1)} \sum_{j=1}^{S} \left( \mathbb{1}[\varphi(\tilde{\mathbf{c}}_j^0)_i = y_i^0] - \mu_i \right) \nabla_{\boldsymbol{\theta}} \log p_{\boldsymbol{\theta}}(\tilde{\mathbf{c}}_j^0 \mid \mathbf{c}^t, \mathbf{x}) \tag{7}$$

where $\mu_i = \frac{1}{S} \sum_{j=1}^{S} \mathbb{1}[\varphi(\tilde{\mathbf{c}}_j^0)_i = y_i^0]$. We provide further details in Section E.

Maximising the **variational entropy** $\mathcal{L}_{\text{H}[q]}$ is challenging: the variational distribution in Section 3.3 samples from a conditioned version of the unmasking model where computing likelihoods, and by extension, maximising the entropy of $q_{\boldsymbol{\theta}}$, is highly untractable. We therefore experimented with two biased approximations of this loss which sufficed for our experiments, and leave more sophisticated approximations for future work:

• **conditional 1-step entropy:** If we have access to a tractable constraint circuit of $\varphi$, we can use it to compute the entropy of an independent distribution over $\mathbf{c}^0$ conditioned on $\mathbf{y}^0$ and $\mathbf{x}$ [7, 83]. Then, we maximise the entropy over the variational distribution when performing time discretisation with a single step ($T = 1$): $\text{H}[q_{\boldsymbol{\theta}}(\tilde{\mathbf{c}}^0 \mid \mathbf{c}^1 = \mathbf{m}, \mathbf{y}^0, \mathbf{x})]$ using the distribution defined in Eq. 6.

- **unconditional 1-step entropy:** Without access to a tractable constraint circuit, we instead maximise the unconditional 1-step entropy $H[q_{\boldsymbol{\theta}}(\tilde{\mathbf{c}}^0 \mid \mathbf{c}^1 = \mathbf{m}, \mathbf{x})]$.

Furthermore, as is common in variational setups [30], we add hyperparameters that weight the contribution of each loss component $\mathcal{L}_{\mathbf{c}}$, $\mathcal{L}_{\mathbf{y}}$, and $\mathcal{L}_{H[q]}$. We found these hyperparameters critical to the performance of the model (see Section H.2 for an ablation study). Finally, unbiased optimisation of $\mathcal{L}_{\mathbf{c}}$ and $\mathcal{L}_{\mathbf{y}}$ also requires calculating the gradient through sampling a $\mathbf{c}^0$ from the variational distribution [59, 70]. Like with the variational entropy, we found that sidestepping this part of the gradient, which would be intractable and have high variance otherwise, simplifies optimisation and yields good performance in practice. See pseudocode for the learning algorithm in Algorithm 1 and additional discussion and definitions of the gradient estimation algorithm in Section E.

---

**Algorithm 1** Algorithm for estimating the gradients of the NELBO for training NESYDM

1: **Given** datapoints $(\mathbf{x}, \mathbf{y}^0)$ and unmasking model $p_{\boldsymbol{\theta}}(\tilde{\mathbf{c}}^0 \mid \mathbf{x}, \mathbf{c}^t)$ with current parameters $\boldsymbol{\theta}$
2: $\mathbf{c}^0 \sim q_{\boldsymbol{\theta}}(\mathbf{c}^0 \mid \mathbf{x}, \mathbf{y}^0)$                                          ▷ Sample from variational distribution (Section 3.3).
3: $t \sim \mathcal{U}(0, 1)$                                                                                    ▷ Sample a random time step.
4: $\mathbf{c}^t \sim q(\mathbf{c}^t \mid \mathbf{c}^0)$                                                          ▷ Mask the concept $\mathbf{c}^0$ to $\mathbf{c}^t$ (Eq. 9).
5: $\tilde{\mathbf{c}}_1^0, \ldots, \tilde{\mathbf{c}}_S^0 \sim q_{\boldsymbol{\theta}}(\tilde{\mathbf{c}}^0 \mid \mathbf{x}, \mathbf{c}^t)$            ▷ Sample $S$ samples from unmasking model.
6: $\mathbf{g}_{\mathbf{y}} \leftarrow g_{\mathbf{y}^0}(\tilde{\mathbf{c}}_1^0, \ldots \tilde{\mathbf{c}}_S^0)$        ▷ Estimate gradient of $\mathcal{L}_{\mathbf{y}}$ using Eq. 60.
7: $\mathbf{g}_{\mathbf{c}} \leftarrow \frac{\alpha'_t}{1 - \alpha_t} \sum_{i \in M_{\mathbf{c}^t}} \nabla_{\boldsymbol{\theta}} \log p_{\boldsymbol{\theta}}(\tilde{w}_i^0 = c_i^0 \mid \mathbf{x}, \mathbf{c}^t)$        ▷ Compute gradient of $\mathcal{L}_{\mathbf{c}}$.
8: $\mathbf{g}_H \leftarrow \nabla_{\boldsymbol{\theta}} \mathcal{L}_H$                                            ▷ Compute gradient of $\mathcal{L}_H$.
9: **return** $\frac{\gamma_{\mathbf{c}}}{C} \mathbf{g}_{\mathbf{c}} + \frac{\gamma_{\mathbf{y}}}{Y} \mathbf{g}_{\mathbf{y}} + \frac{\gamma_H}{C} \mathbf{g}_H$        ▷ Return the weighted sum of the gradients.

---

### 3.5 Sampling and Inference

Next, we describe how we sample from trained NESYDMs to make predictions of $\mathbf{y}$ given $\mathbf{x}$. Exactly computing the mode $\operatorname{argmax}_{\mathbf{y}^0} p_{\boldsymbol{\theta}}^{\text{NESYDM}}(\mathbf{y}^0 \mid \mathbf{x})$ is intractable even for representations supporting tractable marginals [2, 84], therefore we need to approximate it. We use a majority voting strategy, where we sample $L$ concepts $\mathbf{c}_l^0$ from the trained MDM, compute the output with the program $\varphi$, and take the most frequent output:

$$\hat{\mathbf{y}} = \operatorname{argmax}_{\mathbf{y}} \sum_{l=1}^{L} \mathbb{1}[\varphi(\mathbf{c}_l^0) = \mathbf{y}], \quad \mathbf{c}_1^0, \ldots, \mathbf{c}_L^0 \sim p_{\boldsymbol{\theta}}(\mathbf{c}^0 \mid \mathbf{x}, \mathbf{c}^1 = \mathbf{m}). \tag{8}$$

If the concept dimension $C$ is not too large, we use the first-hitting sampler from [94] to sample from $p_{\boldsymbol{\theta}}(\mathbf{c}^0 \mid \mathbf{x}, \mathbf{c}^1 = \mathbf{m})$ exactly in $C$ steps. Otherwise, we use a $T$-step time-discretisation of the reverse process [68], for pseudocode see Algorithm 2. For implementation details, we refer to Section F. Additionally, we experimented with different majority voting strategies, which we discuss in Section H.1. These mainly study whether to do majority voting before or after running the program.

---

**Algorithm 2** Standard time-discretised output prediction for NESYDM

1: **Given** datapoint $\mathbf{x}$ and unmasking model $p_{\boldsymbol{\theta}}(\tilde{\mathbf{c}}^0 \mid \mathbf{x}, \mathbf{c}^t)$ with parameters $\boldsymbol{\theta}$
2: **for** $l \leftarrow 1$ **to** $L$ **do**
3:     $\mathbf{c}^1 = \mathbf{m}$
4:     **for** $k \leftarrow T$ **to** $1$ **do**
5:         $\tilde{\mathbf{c}}^0 \sim p_{\boldsymbol{\theta}}(\tilde{\mathbf{c}}^0 \mid \mathbf{x}, \mathbf{c}^t)$        ▷ Sample from unmasking model (Section 3.3).
6:         $\mathbf{c}^s \sim q(\mathbf{c}^s \mid \mathbf{c}^t, \mathbf{c}^0 = \tilde{\mathbf{c}}^0)$        ▷ Sample from remasking process (Eq. 10).
7:     $\mathbf{c}_l^0 \leftarrow \mathbf{c}^0$                                              ▷ Store the sampled concept
8:     $\mathbf{y}_l \leftarrow \varphi(\mathbf{c}_l^0)$                                    ▷ Compute program output for this sample
9: $\hat{\mathbf{y}} \leftarrow \operatorname{argmax}_{\mathbf{y}} \sum_{l=1}^{L} \mathbb{1}[\mathbf{y}_l = \mathbf{y}]$        ▷ Majority vote
10: **Return** $\hat{\mathbf{y}}$                                                            ▷ Return the most frequent output

---

## 4 Experiments

We aim to answer the following research questions: (**RQ1:**) "Can NESYDMs scale to high-dimensional reasoning problems?" and (**RQ2:**) "Does the expressiveness of NESYDMs improve

Table 1: Accuracy of predicting the correct sum on MNIST Addition with $N = 4$ and $N = 15$ digits. Methods above the horizontal line are exact, and below are approximate. We bold the best-scoring methods in the exact and approximate categories separately.

| METHOD | $N = 4$ | $N = 15$ |
|---|---|---|
| DEEPSOFTLOG [48] | **93.5** $\pm$ **0.6** | 77.1 $\pm$ 1.6 |
| PLIA [21] | 91.84$\pm$ 0.73 | **79.00**$\pm$ **0.73** |
| SCALLOP [21, 43] | 90.88$\pm$ 0.48 | T/O |
| EXAL [85] | 91.65$\pm$ 0.57 | 73.27$\pm$ 2.05 |
| A-NESI [80] | **92.56**$\pm$ **0.79** | **76.84**$\pm$ **2.82** |
| NESYDM (*ours*) | **92.49**$\pm$ **0.98** | **77.29**$\pm$ **1.40** |

Table 2: **NESYDM significantly scales beyond current NeSy predictors.** Accuracy of predicting a shortest path on visual path planning with different grid sizes. Above the horizontal line are methods predicting continuous costs, while below are approximate NeSy methods that predict discrete, binned costs.

| METHOD | $12 \times 12$ | $30 \times 30$ |
|---|---|---|
| I-MLE [62] | 97.2 $\pm$ 0.5 | 93.7 $\pm$ 0.6 |
| EXAL [85] | 94.19$\pm$ 1.74 | 80.85$\pm$ 3.83 |
| A-NESI [80] | 94.57$\pm$ 2.27 | 17.13$\pm$ 16.32 |
| A-NESI+RL [80] | **98.96**$\pm$ **1.33** | 67.57$\pm$ 36.76 |
| NESYDM (*ours*) | **99.41**$\pm$ **0.06** | **97.40**$\pm$ **1.23** |

reasoning shortcut awareness compared to independent models?" Since there are currently no scalable RS-aware NeSy methods, the baselines we use are separated for the two research questions. We match experimental setups of the baselines, using the same datasets and neural network architectures for a fair comparison. To approximate the variational entropy (Section 3.4), we use the unconditional entropy for the experiments, as the conditional entropy is intractable. For the RSBench experiments, we tried both. We use the linear noising schedule $\alpha_t = 1 - t$ for all experiments.

For all experiments, we repeat runs with 10 different random seeds. In all tables, we find the best-performing methods with bold font. In particular, we bold all methods that are not statistically different from the highest-scoring method according to an unpaired one-sided Mann-Whitney U test at a significance level of 0.05. We provide additional experimental details in Section G. Code is available at https://github.com/HEmile/neurosymbolic-diffusion.

## 4.1 RQ1: Scalability of NESYDM

To evaluate the scalability of NESYDM, we consider two NeSy benchmark tasks with high combinatorial complexity: multidigit MNIST Addition and visual path planning. We compare to current approximate NeSy methods that use the independence assumption and are not RS-aware, namely A-NeSI [81], Scallop [43], and EXAL [85].

**Multidigit MNIST Addition.** The input $\mathbf{x}$ is a sequence of 2 numbers of $N$ digits, and the output $\mathbf{y}$ is the sum of the two numbers, split up into $N + 1$ digits. The goal is to train a neural network that recognises the individual digits $\mathbf{c} \in \{0, 1, \ldots, 9\}^{2N}$ in the input from input-output examples. There are no dependencies between the digits and the problem is not affected by reasoning shortcuts, so we do not expect NESYDM to improve significantly over NeSy methods that use the independence assumption. Still, we find in Table 1 that NESYDM, which uses a much more expressive model than the baselines, performs similar to the state-of-the-art approximate method A-NeSI, and is competitive with exact methods [19, 48]. Therefore, the expressivity does not come at a cost of performance and scalability in traditional NeSy benchmarks.

**Visual path planning.** We study the problem described in Example 2.1. Specifically, we train a neural network to predict the correct cost $c_{i,j}$ at each of the $N \times N$ grid cells. Then, we use Dijkstra's algorithm to find the shortest path $\mathbf{y} \in \{0, 1\}^{N \times N}$, where $y_{i,j} = 1$ if the shortest path passes through cell $i, j$ and 0 otherwise. Like other NeSy methods, we predict costs with a 5-dimensional categorical variable $\mathbf{c} \in \{1, \ldots, 5\}^{N \times N}$. We also compare to I-MLE, the state-of-the-art method that predicts costs as a single continuous variable [62]. We find in Table 2 that NESYDM significantly outperforms all baselines on the challenging $30 \times 30$ problem, including I-MLE. This problem has a combinatorial space of $5^{900}$ and is considered very challenging for NeSy and neural models [65]. On the $12 \times 12$ problem, we cannot reject the null hypothesis that NESYDM outperforms A-NeSI + RLOO, but it does have much lower variance, highlighting the reliability of our method.

## 4.2 RQ2: RS-awareness of NESYDM

To evaluate the RS awareness of NESYDM, we use the RSBench dataset [56] of reasoning problems that cannot be disambiguated from data alone. We consider two synthetic problems and a real-

Table 3: **NᴇSʏDM is a performant and RS-aware NeSy predictor** as shown on several tasks from the RSBench dataset. We report relevant performance metrics for each task, and concept calibration using ECE to evaluate RS-awareness (see Section G.4.2 for a motivation for this metric). We underline the second-best-scoring method if there is only a single statistically significant best-scoring method. The first two methods use the independence assumption. Note that SL does not support BDD-OIA.

| | Method | $PNP^{\perp\!\perp}$ | $SL^{\perp\!\perp}$ | BEARS [55] | | NᴇSʏDM (*ours*) | |
| | | | | PNP | SL | Uncond H | Cond H |
|---|---|---|---|---|---|---|---|
| **MNIST HALF** | $ACC_y$ ↑ | 98.24± 0.12 | **99.62± 0.12** | 99.19± 0.12 | **99.76± 0.00** | 99.12± 0.10 | 99.12± 0.10 |
| | $ACC_c$ ↑ | 42.76± 0.14 | 42.88± 0.09 | 43.26± 0.75 | 42.86± 0.00 | **79.41± 6.58** | 71.16± 1.77 |
| | $ACC_{y,OOD}$ ↑ | 5.81± 0.07 | 0.48± 0.21 | 6.31± 1.10 | 0.11± 0.09 | 10.9± 0.05 | **28.44± 0.90** |
| | $ACC_{c,OOD}$ ↑ | 38.97± 0.08 | 38.92± 0.11 | 39.49± 1.07 | 38.88± 0.03 | 57.22± 0.49 | **62.76± 0.89** |
| | $ECE_{c,ID}$ ↓ | 69.40± 0.35 | 70.61± 0.18 | 36.81± 0.17 | 37.61± 1.22 | 39.52± 5.01 | **4.18± 2.56** |
| | $ECE_{c,OOD}$ ↓ | 86.67± 0.18 | 87.95± 0.14 | 37.89± 2.18 | 35.99± 2.88 | 35.07± 2.67 | **11.74± 1.18** |
| **MNIST E-O** | $ACC_y$ ↑ | 70.77± 0.45 | 97.38± 0.31 | 92.02± 3.14 | **98.67± 0.27** | 97.52± 0.37 | 98.27± 0.44 |
| | $ACC_c$ ↑ | 0.40± 0.04 | 0.33± 0.05 | 0.48± 0.10 | 0.19± 0.08 | 0.36± 0.27 | **20.33± 1.33** |
| | $ACC_{y,OOD}$ ↑ | **7.29± 0.49** | 0.05± 0.06 | 1.60± 2.04 | 0.00± 0.00 | 0.00± 0.00 | 0.02± 0.04 |
| | $ACC_{c,OOD}$ ↑ | 7.50± 0.32 | 7.07± 0.09 | 9.36± 2.13 | 6.25± 1.46 | 4.65± 0.49 | **14.25± 0.76** |
| | $ECE_{c,ID}$ ↓ | 81.04± 1.15 | 82.18± 1.57 | 28.82± 2.19 | 34.51± 1.65 | 20.93± 0.49 | **2.70± 1.21** |
| | $ECE_{c,OOD}$ ↓ | 85.44± 0.72 | 86.96± 1.15 | 26.83± 1.56 | 32.61± 3.32 | 19.13± 0.50 | **5.77± 0.98** |
| **BDD** | $MF1_y$ ↑ | **63.71± 1.50** | – | 60.80± 0.11 | – | 61.67± 0.32 | 62.63± 0.53 |
| | $MF1_c$ ↑ | 10.41± 1.90 | – | **19.25± 0.16** | – | 18.50± 0.21 | 13.77± 0.51 |
| | $ECE_c$ ↓ | 38.89± 1.34 | – | **16.00± 0.20** | – | 18.86± 1.75 | 21.72± 1.83 |

world task. MNIST Half and MNIST Even-Odd (MNIST E-O) are variations of MNIST Addition constructed to ensure disambiguation of concepts is impossible. They have OOD test-sets to diagnose overconfident classifiers. BDD-OIA (BDD) is a self-driving task [87] where a model predicts what actions a car can take given a dashcam image. NeSy predictors extract high-level concepts from the image and use rules to predict the allowed actions. We compare to NeSy predictors using the independence assumption, namely Semantic Loss ($SL^{\perp\!\perp}$) [86] and a standard probabilistic NeSy predictor ($PNP^{\perp\!\perp}$). We also compare to BEARS, an RS-aware ensemble of NeSy predictors with the independence assumption [55].

In Table 3, we find that NᴇSʏDM strikes a good balance between accuracy and RS-awareness throughout the datasets. On the MNIST tasks, it attains significantly better concept accuracy than competitors, both in- and out-of-distribution. Furthermore, NᴇSʏDM, especially using the conditional entropy, has much better concept calibration than both baselines using the independence assumption and RS-aware baselines. We report additional results on these datasets in Section H.1 and find that different majority voting strategies may improve OOD performance. On BDD-OIA, we find that NᴇSʏDM has better predictive performance on outputs than BEARS while significantly improving calibration and concept performance compared to $PNP^{\perp\!\perp}$ using the independence assumption. Furthermore, we note that, unlike the baselines, NᴇSʏDM is much more scalable as highlighted in Section 4.1.

## 5 Further related work

**NeSy predictors.** The field of NeSy predictors is primarily divided into methods using fuzzy logics [10, 17, 28, 78] and those using probabilistic logics [6, 43, 52, 80, 86]. Fuzzy methods implicitly assume a form of independence between concepts, while probabilistic methods can model dependencies. Previous methods that went beyond the independence assumption mixed multiple independent distributions, like in SPL [6] and BEARS [55] which is specifically designed for RS-awareness. Neurosymbolic probabilistic logic programming frameworks like DeepProbLog and Scallop [43, 52] allow modifying the program to increase expressivity compared to the naive independence over concepts. However, these methods are built on exact or top-$k$ inference, which is difficult to scale to high-dimensional reasoning problems like visual path planning when the number of dependencies grows. Relatedly, DeepGraphLog [34] extends DeepProbLog by using graph neural networks to model dependencies between concepts, also relying on exact inference. Conversely, all current methods focussed on approximate inference to scale neurosymbolic predictors assume independence between concepts [73, 80, 85], hence lacking RS-awareness.

**NeSy generative models.** A closely related topic is generating from expressive models like large language models (LLMs) and diffusion models while involving programs and constraints. For LLMs, this was studied with NeSy loss functions encoding the constraints [2, 3, 13] and with constrained decoding, for example using sequential Monte Carlo methods [42, 46, 93] and by combining the LLM with approximations using probabilistic circuits [5, 91, 92]. However, these methods adopt heuristics to steer the LLM towards a constraint, for instance, by using a pseudo-likelihood formulation [2, 3] or training an HMM surrogate that approximates the LLM [91, 92]. Instead, for NESYDM we formulate a principled NELBO, and we do so by exploiting the local structure that diffusion models offer. Furthermore, some methods tackle constrained generation from GANs [24, 75, 76], VAEs [58], deep HMMs [74], and continuous diffusion models [31, 69]. We leave extensions of NESYDM to this generative setting to future work.

## 6 Conclusion

In this paper, we introduced NESYDMS, the first method to integrate masked diffusion models as the neural network extractor in neurosymbolic predictors. We show how to scale NESYDMS by using efficient probabilistic reasoning techniques on *local* unmasking distributions while minimising a *global* NELBO that lower-bounds the data log-likelihood. Empirically, we show that NESYDMS position themselves as one of the best NeSy predictors available that can scale to high-dimensional reasoning problems while being RS-aware. This is a crucial property for NeSy predictors deployed in real-world safety-critical applications, as they need to be well calibrated and generalise robustly.

**Limitations and future work.** The NESYDM NELBO can be extended to incorporate additional exact inference routines if we can obtain an efficient circuit, e.g., as the tractable representation for a symbolic program [63]. Otherwise, as argued in Section 3.4, our sampling-based approach relies on the ability to decompose the output $\mathbf{y}$ into separate dimensions to ensure the search in RLOO is decomposed into independent subproblems. Together, this limits the scalability of NESYDM to tasks with either efficient circuit representations or decomposable output spaces. Understanding how to combine these two aspects, or how to automatically (and approximately) reduce a different setting into one of them, is an interesting and challenging future venue. Two other areas of improvement are our approach to maximising the variational entropy and the influence of the indirect gradient coming from sampling from the variational distribution. Finally, we believe studying how NESYDMS extend to other discrete diffusion models than masked diffusion [9] models is an interesting direction. NESYDM could even be extended to hybrid diffusion models that involve both symbolic, discrete concepts and continuous latent variables by using recent work on generating under continuous constraints [20, 40, 76].

## Acknowledgements

Emile van Krieken was funded by ELIAI (The Edinburgh Laboratory for Integrated Artificial Intelligence), EPSRC (grant no. EP/W002876/1). Pasquale Minervini was partially funded by ELIAI, EPSRC (grant no. EP/W002876/1), an industry grant from Cisco, and a donation from Accenture LLP. Edoardo M. Ponti is supported by the ERC Starting Grant AToM-FM (101222956). Antonio Vergari was supported by the "UNREAL: Unified Reasoning Layer for Trustworthy ML" project (EP/Y023838/1) selected by the ERC and funded by UKRI EPSRC. We would like to express our gratitude to Samuele Bortolotti, Emanuele Marconato, Lennert de Smet, Adrián Javaloy, and Jaron Maene for fruitful discussions during the writing of this paper.

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

# A  Additional background on masked diffusion models

Here, we will discuss additional background and formalisation of masked diffusion models (MDMs). This background is used to derive the NELBO of the masked diffusion model in Section D.2 and the loss with arbitrary joints in Section C.

**Forward process details.** We first define the continuous-time forward process $q(\mathbf{c}^t \mid \mathbf{c}^0)$, which masks the data up to timestep $t \in [0, 1]$ using the forward process defined in Eq. 2.

$$q(\mathbf{c}^t \mid \mathbf{c}^0) = \prod_{i=1}^{C} \alpha_t \mathbb{1}[c_i^t = c_i^0] + (1 - \alpha_t)\mathbb{1}[c_i^t = \mathrm{m}] \tag{9}$$

Secondly, we need the *reverse posterior* $q(\mathbf{c}^s \mid \mathbf{c}^t, \mathbf{c}^0)$, which is the distribution of the initial state $\mathbf{c}^0$ given the state at timestep $t$ and the final state. Here we assume $c_i^t$ is either equal to the mask value $\mathrm{m}$ or to the value of $c_i^0$, as otherwise the probability is not well-defined. The form for each case is (see [68], A.2.1)

$$q(\mathbf{c}^s \mid \mathbf{c}^t, \mathbf{c}^0) = \prod_{i=1}^{C} q(c_i^s \mid c_i^t, c_i^0) \tag{10}$$

$$q(c_i^s \mid c_i^t = c_i^0, c_i^0) = \mathbb{1}[c_i^s = c_i^0] \tag{11}$$

$$q(c_i^s \mid c_i^t = \mathrm{m}, c_i^0) = \frac{1 - \alpha_s}{1 - \alpha_t}\mathbb{1}[c_i^s = \mathrm{m}] + \frac{\alpha_s - \alpha_t}{1 - \alpha_t}\mathbb{1}[c_i^s = c_i^0] \tag{12}$$

We note that $q(c_i^s \mid c_i^t = c_i^0, c_i^0)$ refers to the probability of $c_i^s$ conditioned on some value for the variable $c_i^0$ and where the value of variable $c_i^t$ equals this value. If $c_i^t$ indeed is equal to the value of $c_i^0$, the distribution deterministically returns that value. If it is masked instead, it either stays masked or turns into the value of $c_i^0$ with a probability depending on $\alpha_t$.

**Additional notation.** We let $M_{\mathbf{c}^t} = \{i : c_i^t = \mathrm{m}\}$ refer to the dimensions that are masked in $\mathbf{c}^t$. Similarly, $U_{\mathbf{c}^t} = \{i : c_i^t \neq \mathrm{m}\}$ is the set of unmasked dimensions of $\mathbf{c}^t$. Furthermore, we will use $\mathbf{c}^s \succeq \mathbf{c}^t$ to denote that $\mathbf{c}^s$ is a *(partial) extension* of $\mathbf{c}^t$. This means $\mathbf{c}^s$ agrees on all unmasked dimensions of $\mathbf{c}^t$ with $\mathbf{c}^t$, that is, $w_i^s = w_i^t$ for all $i \in U_{\mathbf{c}^t}$. We will also use $\mathbf{c}^0 \succeq^C \mathbf{c}^t$ to denote that $\mathbf{c}^0$ is a *complete* extension that does not have any masked dimensions. Finally, we use notation such as $\mathbf{c}_{U_{\mathbf{c}^t}}^s$ to index $\mathbf{c}^s$ using the set of indices $U_{\mathbf{c}^t}$, the unmasked dimensions of $\mathbf{c}^t$.

**Reverse process definition.** Using $p_{\boldsymbol{\theta}}(\mathbf{c}^s \mid \mathbf{c}^t)$ (Eq. 3), we can express the intractable generative model $p_{\boldsymbol{\theta}}^{\mathrm{MDM}}(\mathbf{c}^0)$, for time discretisation $T$, as

$$p_{\boldsymbol{\theta}}^{\mathrm{MDM}}(\mathbf{c}^0) := \sum_{\mathbf{C}_{\backslash\{0\}}} \prod_{k=1}^{T} p_{\boldsymbol{\theta}}(\mathbf{c}^{s(k)} \mid \mathbf{c}^{t(k)}), \tag{13}$$

where the sum over $\mathbf{C}_{\backslash\{0\}}$ iterates over all trajectories $\mathbf{c}^1, \ldots, \mathbf{c}^{\frac{T-1}{T}}$ from fully masked $\mathbf{c}^1 = \mathbf{m}$ to unmasked $\mathbf{c}^0$, and $s(k) = \frac{k-1}{T}$ and $t(k) = \frac{k}{T}$ index the timesteps.

Several recent papers [68, 72] proved that this model has a simple negative variational lower bound (NELBO) under a continuous-time process, that is, when $T \to \infty$. Given a dataset of samples $\mathbf{c}^0$, this NELBO resembles a weighted cross-entropy loss:

$$-\log p_{\boldsymbol{\theta}}^{\mathrm{MDM}}(\mathbf{c}^0) \leq \mathcal{L}^{\mathrm{MDM}} = \mathbb{E}_{t \sim [0,1], \mathbf{c}^t \sim q(\mathbf{c}^t \mid \mathbf{c}^0)} \left[ \frac{\alpha_t'}{1 - \alpha_t} \sum_{i \in M_{\mathbf{c}^t}} \log p_{\boldsymbol{\theta}}(\tilde{c}_i^0 = c_i^0 \mid \mathbf{c}^t) \right]. \tag{14}$$

Here $\alpha_t' = \frac{\partial \alpha_t}{\partial t}$, $q(\mathbf{c}^t \mid \mathbf{c}^0)$ is computed with Eq. 9, and the cross-entropy term computes the loss on the factors of the unmasking model $p_{\boldsymbol{\theta}}(\tilde{c}_i^0 \mid \mathbf{c}^t)$. When using the common linear noising schedule, then $\alpha_t = 1 - t$, $\frac{\alpha_t'}{1-\alpha_t} = -\frac{1}{t}$. This bound holds when the unmasking model $p_{\boldsymbol{\theta}}(\tilde{\mathbf{c}}^0 \mid \mathbf{c}^t)$ assigns 0 probability to the mask value (*zero masking probabilities*), and assigns a probability of 1 to unmasked dimensions (*carry-over unmasking*), i.e., for all $i \notin M_{\mathbf{c}^t}$, $p_{\boldsymbol{\theta}}(\tilde{c}_i^0 = c_i^0 \mid \mathbf{c}^t) = 1$ [68].

# B  Analysis of the output unmasking loss

Here, we will discuss the output unmasking loss $\mathcal{L}_{\mathbf{y}}$ in more detail, and relate it to other common loss functions in the NeSy literature. In our problem setup, we assume a program $\varphi : [V]^C \to [V]^Y$ that maps concepts $\mathbf{c}^0$ to outputs $\mathbf{y}^0$. Then, we defined the WMC in Eq. 1 as the probability that some $\mathbf{c}^0$ maps to $\mathbf{y}^0$. This constraint can be understood as

$$\mathbb{1}[\varphi(\mathbf{c}^0) = \mathbf{y}^0] = \mathbb{1}\left[\bigwedge_{i=1}^{Y} \varphi(\mathbf{c}^0)_i = y_i^0\right]. \tag{15}$$

That is, we can see this setup as actually having $Y$ different programs, and we want each program to return the right output. Now, disregarding the weighting and sampling, $\mathcal{L}_{\mathbf{y}}$ is

$$\mathcal{L}_{\mathbf{y}} = \sum_{i=1}^{Y} \log \sum_{\tilde{\mathbf{c}}^0} p_{\boldsymbol{\theta}}(\tilde{\mathbf{c}}^0 \mid \mathbf{c}^t, \mathbf{x})\mathbb{1}[\varphi(\tilde{\mathbf{c}}^0)_i = y_i^0] \tag{16}$$

$$= \sum_{i=1}^{Y} \log p_{\boldsymbol{\theta}}(\tilde{y}_i^0 = y_i^0 \mid \mathbf{c}^t, \mathbf{x}) \tag{17}$$

This loss is a sum of $Y$ different WMC terms, one for each of the $Y$ different programs. $\mathcal{L}_{\mathbf{y}}$ assumes, in a vacuum, that these programs are independent, meaning we can sum the losses for each program independently. How could that be possible?

This is actually a common property of continuous-time losses of discrete diffusion models. For instance, one can observe the same in the NELBO of MDMs in Eq. 14. There, the goal is to reconstruct the (masked) dimensions of $\mathbf{c}^0$ independently. In fact, to perfectly fit an MDM, the goal is merely to perfectly fit each of the $C$ different conditional data marginals $p(\tilde{c}_i^0 \mid \mathbf{c}^t)$ perfectly, without regard for any dependencies between dimensions [44]. The dependencies for the full MDM are handled by the iterative unmasking process, which changes the condition at each step. The same property holds for $\mathcal{L}_{\mathbf{y}}$: the dependencies between the different programs are (ideally) handled by different conditions $\mathbf{c}^0$ at each step.

We highlight that this loss is related to existing loss functions in the NeSy literature. In particular, for programs that implement conjunctive normal forms (CNFs), this loss is equivalent to the logarithm of the product t-norm, which is a common loss function in the NeSy literature [10, 78]. More precisely, if $\mathbf{c} \in \{0,1\}^C$ models the $C$ variables of the CNF and $\mathbf{y} \in \{0,1\}^Y$ the $Y$ clauses consisting of disjunctions of literals $l_{i1} \vee ... \vee l_{i,k_i}$, then $\varphi(\mathbf{c})_i = \bigvee_{j=1}^{k_i} l_{ij}$ computes the truth value of the $i$th clause of the CNF. Under the independence assumption, the probability that disjunction $i$ holds (that is, whether $\varphi(\mathbf{c})_i = 1$) is

$$p_{\boldsymbol{\theta}}(y_i = 1 \mid \mathbf{x}) = 1 - \prod_{j=1}^{k_i}(1 - p_{\boldsymbol{\theta}}(l_{ij} \mid \mathbf{x})) \tag{18}$$

which is equal to the product t-conorm of the probabilities of the literals. Finally, the logarithm product t-norm takes the logarithm over the product of these probabilities, implicitly assuming these clauses are independent:

$$\mathcal{L}^{\text{Log-product}} = -\sum_{i=1}^{Y} \log p_{\boldsymbol{\theta}}(y_i = 1 \mid \mathbf{x}). \tag{19}$$

Note that, outside the reweighting with $\alpha_t'$, this is precisely what $\mathcal{L}_{\mathbf{y}}$ would compute for this problem (Eq. 17).

This equality between $\mathcal{L}_{\mathbf{y}}$ and $\mathcal{L}^{\text{Log-product}}$ holds only for CNFs: for general programs, the product t-norm is not equal to the probability on the output of a program, unlike the disjunction case. For example, the different subprograms used in our experiments are not expressed as CNFs. Furthermore, our setup gives more flexibility even in the CNF case by allowing us to redefine what the dimensions of $\mathbf{y}$ represent. For instance, we can remove the independence assumption between a set of clauses by defining $y_i$ as the conjunction of these clauses. In that sense, it is highly related to Semantic Strengthening [4], which starts from $\mathcal{L}^{\text{Log-product}}$, and then dynamically joins clauses by building a probabilistic circuit to relax the independence assumption. This idea can be directly applied to our setup, which we leave as future work.

# C  Masked Diffusion with Arbitrary Joint Distributions

In this section, we will prove Theorem C.1 which states that the NELBO in Eq. 14 also holds for non-factorised unmasking models $p_{\boldsymbol{\theta}}(\tilde{\mathbf{c}}^0 \mid \mathbf{c}^t)$. We use the notation introduced in Section A and Section 2.2. During this proof, we will derive both discrete- and continuous-time versions of the NELBO. In this appendix, we will use $\mathbf{C}_{\setminus 0}$ to refer to $\mathbf{c}^{1/T}, ..., \mathbf{c}^1$, $t = \frac{k}{T}$ and $s = \frac{k-1}{T}$. This result is related to the tractability result of [14], namely that in a continuous-time process, the probability that two dimensions are unmasked at exactly the same time step in $[0, 1]$ is 0.

**Theorem C.1.** *Let $p_{\boldsymbol{\theta}}(\tilde{\mathbf{c}}^0 \mid \mathbf{c}^t)$ be any conditional joint distribution over $\tilde{\mathbf{c}}^0$ with conditional marginals $p_{\boldsymbol{\theta}}(\tilde{c}_i \mid \mathbf{c}^t)$ that satisfy the following assumptions for all $i \in \{1, \ldots, C\}$:*

*1. Zero masking probabilities: $p_{\boldsymbol{\theta}}(\tilde{c}_i = \mathrm{m} \mid \mathbf{c}^t) = 0$.*

*2. Carry-over unmasking: Given some $\mathbf{c}^t \in (V_{\mathbf{c}} + 1)^C$, $p_{\boldsymbol{\theta}}(\tilde{c}_i = c_i^t \mid \mathbf{c}^t) = 1$.*

*3. Proper prior: $p_{\boldsymbol{\theta}}(\mathbf{c}^1) = \mathbb{1}[\mathbf{c}^1 = \mathbf{m}]$.*

*Let $p_{\boldsymbol{\theta}}(\mathbf{c}^s \mid \mathbf{c}^t)$ be the reverse process defined in Eq. 3 using $p_{\boldsymbol{\theta}}(\tilde{\mathbf{c}}^0 \mid \mathbf{c}^t)$ instead of a fully factorised model. Then as $T \to \infty$,*

$$-\log p_{\boldsymbol{\theta}}^{\mathrm{MDM}}(\mathbf{c}^0) \leq \mathcal{L}^{\mathrm{MDM}} = \mathbb{E}_{t \sim [0,1], \mathbf{c}^t \sim q}\left[\frac{\alpha_t'}{1 - \alpha_t} \sum_{i \in M_{\mathbf{c}^t}} \log p_{\boldsymbol{\theta}}(\tilde{c}_i^0 = c_i^0 \mid \mathbf{c}^t)\right]. \tag{20}$$

*Proof.* We start with a standard variational diffusion models derivation that closely follows those presented in [36, 47].

$$-\log p_{\boldsymbol{\theta}}^{\mathrm{MDM}}(\mathbf{c}^0) = -\log \sum_{\mathbf{C}_{\setminus 0}} p_{\boldsymbol{\theta}}(\mathbf{C}) \leq -\mathbb{E}_{q(\mathbf{C}_{\setminus 0}|\mathbf{c}^0)}\left[\log \frac{p_{\boldsymbol{\theta}}(\mathbf{C})}{q(\mathbf{C}_{\setminus 0} \mid \mathbf{c}^0)}\right]$$

Now we reduce the nominator with Bayes theorem and by conditioning on $\mathbf{c}^0$, which is conditionally independent given $\mathbf{c}^s$:

$$q(\mathbf{C}_{\setminus 0} \mid \mathbf{c}^0) = q(\mathbf{c}^{1/T} \mid \mathbf{c}^0) \prod_{k=2}^{T} q(\mathbf{c}^t \mid \mathbf{c}^s) = q(\mathbf{c}^{1/T} \mid \mathbf{c}^0) \prod_{k=2}^{T} q(\mathbf{c}^t \mid \mathbf{c}^s, \mathbf{c}^0)$$

$$= q(\mathbf{c}^{1/T} \mid \mathbf{c}^0) \prod_{k=2}^{T} \frac{q(\mathbf{c}^s \mid \mathbf{c}^t, \mathbf{c}^0)q(\mathbf{c}^t \mid \mathbf{c}^0)}{q(\mathbf{c}^s \mid \mathbf{c}^0)} = q(\mathbf{c}^1 \mid \mathbf{c}^0) \prod_{k=2}^{T} q(\mathbf{c}^s \mid \mathbf{c}^t, \mathbf{c}^0), \tag{21}$$

where in the last step we use that the $q(\mathbf{c}^t \mid \mathbf{c}^0)$ and $q(\mathbf{c}^s \mid \mathbf{c}^0)$ cancel out in the product over $t$, leaving only $q(\mathbf{c}^t \mid \mathbf{c}^s, \mathbf{c}^0)$. Filling in Eq. 3,

$$= -\mathbb{E}_{q(\mathbf{C}_{\setminus 0}|\mathbf{c}^0)}\left[\log \frac{p_{\boldsymbol{\theta}}(\mathbf{c}^0 \mid \mathbf{c}^{1/T})p(\mathbf{c}^1)}{q(\mathbf{c}^1 \mid \mathbf{c}^0)} \frac{\prod_{k=2}^{T} p_{\boldsymbol{\theta}}(\mathbf{c}^s \mid \mathbf{c}^t)}{\prod_{k=2}^{T} q(\mathbf{c}^s \mid \mathbf{c}^t, \mathbf{c}^0)}\right] \tag{22}$$

$$= -\mathbb{E}_{q(\mathbf{C}_{\setminus 0}|\mathbf{c}^0)}\left[\log p_{\boldsymbol{\theta}}(\mathbf{c}^0 \mid \mathbf{c}^{1/T}) + \log \frac{p(\mathbf{c}^1)}{q(\mathbf{c}^1 \mid \mathbf{c}^0)} + \log \frac{\prod_{k=2}^{T} p_{\boldsymbol{\theta}}(\mathbf{c}^s \mid \mathbf{c}^t)}{\prod_{k=2}^{T} q(\mathbf{c}^s \mid \mathbf{c}^t, \mathbf{c}^0)}\right] \tag{23}$$

$$= \underbrace{\mathbb{E}_{q(\mathbf{c}^{1/T}|\mathbf{c}^0)}\left[-\log p_{\boldsymbol{\theta}}(\mathbf{c}^0 \mid \mathbf{c}^{1/T})\right]}_{\mathcal{L}_{\mathrm{rec},T}: \text{ reconstruction loss}} + \sum_{k=2}^{T} \underbrace{\mathbb{E}_{q(\mathbf{c}^t|\mathbf{c}^0)}\mathrm{KL}[q(\mathbf{c}^s \mid \mathbf{c}^t, \mathbf{c}^0)\|p_{\boldsymbol{\theta}}(\mathbf{c}^s \mid \mathbf{c}^t)]}_{\mathcal{L}_{\mathrm{unm},T,k}: \text{ unmasking loss at timestep } k} + G \tag{24}$$

where $G = \mathbb{E}_{q(\mathbf{c}^1|\mathbf{c}^0)} \log \frac{p(\mathbf{c}^1)}{q(\mathbf{c}^1|\mathbf{c}^0)}$ is a constant and equal to 0 if $p(\mathbf{c}^1) = \mathbb{1}[\mathbf{c}^1 = \mathbf{m}]$.

**Lemma C.2.** *Using the assumptions of Theorem C.1, for any integer $T > 1$,*

$$\mathcal{L}_{\mathrm{rec},T} = \mathbb{E}_{q(\mathbf{c}^{1/T}|\mathbf{c}^0)}[-\log p_{\boldsymbol{\theta}}(\tilde{\mathbf{c}}^0 = \mathbf{c}^0 \mid \mathbf{c}^{1/T})] \tag{25}$$

*Proof.* First note that, using Eq. 12,

$$q(c_i^0 \mid c_i^{1/T} = \mathrm{m}, \tilde{c}_i^0) = \frac{1 - \alpha_0}{1 - \alpha_{1/T}} \mathbb{1}[c_i^0 = \mathrm{m}] + \frac{\alpha_0 - \alpha_{1/T}}{1 - \alpha_{1/T}} \mathbb{1}[\tilde{c}_i^0 = c_i^0]$$

$$= \frac{1 - 1}{1 - \alpha_{1/T}} \cdot 0 + \frac{1 - \alpha_{1/T}}{1 - \alpha_{1/T}} \mathbb{1}[c_i^0 = c_i^0] = \mathbb{1}[\tilde{c}_i^0 = c_i^0]$$

since no elements of $\tilde{\mathbf{c}}^0$ are masked and $\alpha_0 = 1$ by definition, and so combined with Eq. 11, we get $q(c_i^0 \mid c_i^{1/T}, \tilde{c}_i^0) = \mathbb{1}[\tilde{c}_i^0 = c_i^0]$. Therefore,

$$\mathbb{E}_{q(\mathbf{c}^{1/T} \mid \mathbf{c}^0)}[-\log p_{\boldsymbol{\theta}}(\mathbf{c}^0 \mid \mathbf{c}^{1/T})] = \mathbb{E}_{q(\mathbf{c}^{1/T} \mid \mathbf{c}^0)} \left[ -\log \sum_{\tilde{\mathbf{c}}_0} p_{\boldsymbol{\theta}}(\tilde{\mathbf{c}}^0 \mid \mathbf{c}^{1/T}) \prod_{i=1}^{C} q(c_i^0 \mid c_i^{1/T}, \tilde{c}_i^0) \right]$$

$$= \mathbb{E}_{q(\mathbf{c}^{1/T} \mid \mathbf{c}^0)} \left[ -\log \sum_{\tilde{\mathbf{c}}_0} p_{\boldsymbol{\theta}}(\tilde{\mathbf{c}}^0 \mid \mathbf{c}^{1/T}) \prod_{i=1}^{C} \mathbb{1}[\tilde{c}_i^0 = c_i^0] \right]$$

$$= \mathbb{E}_{q(\mathbf{c}^{1/T} \mid \mathbf{c}^0)}[-\log p_{\boldsymbol{\theta}}(\tilde{\mathbf{c}}^0 = \mathbf{c}^0 \mid \mathbf{c}^{1/T})]$$

Where we use that the only nonzero term in the sum is when $\tilde{\mathbf{c}}^0 = \mathbf{c}^0$. $\qquad\square$

Next, we focus on $\mathcal{L}_{\mathrm{unm},T,k}$ in Eq. 24. The standard derivation of the MDM NELBO in [68] computes the dimension-wise KL-divergence between the forward and reverse process, and then sums. This is not possible in our setting because we assume arbitrary joints for the unmasking model, and so the KL-divergence does not decompose trivially.

**Lemma C.3.** *Using the assumptions of Theorem C.1, for any integer $T > 1$ and $k \in \{2, \dots, T\}$,*

$$\mathcal{L}_{\mathrm{unm},T,k} = \mathbb{E}_{q(\mathbf{c}^t \mid \mathbf{c}^0)} \mathrm{KL}[q(\mathbf{c}^s \mid \mathbf{c}^t, \mathbf{c}^0) \| p_{\boldsymbol{\theta}}(\mathbf{c}^s \mid \mathbf{c}^t)]$$
$$= \mathbb{E}_{q(\mathbf{c}^s, \mathbf{c}^t \mid \mathbf{c}^0)} - \log p_{\boldsymbol{\theta}}(\tilde{\mathbf{c}}_{U_{\mathbf{c}^s} \setminus U_{\mathbf{c}^t}}^0 = \mathbf{c}_{U_{\mathbf{c}^s} \setminus U_{\mathbf{c}^t}}^0 \mid \mathbf{c}^t) \tag{26}$$

*Proof.* We first consider what terms in the KL-divergence in $\mathcal{L}_{\mathrm{unm},T,k}$ are nonzero. First, note that $\mathbf{c}^t$ needs to extend $\mathbf{c}^s$ (i.e., $\mathbf{c}^s \succeq \mathbf{c}^t$) as otherwise $q(\mathbf{c}^s \mid \mathbf{c}^t, \mathbf{c}^0) = 0$ by Eq. 11. Next, the unmasked dimensions in $\mathbf{c}^s$ need to be consistent with $\mathbf{c}^0$ by Eq. 12, in other words, $\mathbf{c}^0 \succeq \mathbf{c}^s$. Then, the $|M_{\mathbf{c}^s}|$ dimensions that stay unmasked get a factor of $\frac{1-\alpha_s}{1-\alpha_t}$, while the $|M_{\mathbf{c}^t}| - |M_{\mathbf{c}^s}|$ dimensions that become unmasked get a factor of $\frac{\alpha_s - \alpha_t}{1-\alpha_t}$. Assuming $\mathbf{c}^0 \succeq^C \mathbf{c}^t$, we have

$$q(\mathbf{c}^s \mid \mathbf{c}^t, \mathbf{c}^0) = \begin{cases} \left(\frac{1-\alpha_s}{1-\alpha_t}\right)^{|M_{\mathbf{c}^s}|} \left(\frac{\alpha_s - \alpha_t}{1-\alpha_t}\right)^{|M_{\mathbf{c}^t}| - |M_{\mathbf{c}^s}|} & \text{if } \mathbf{c}^0 \succeq \mathbf{c}^s \succeq \mathbf{c}^t, \\ 0 & \text{otherwise.} \end{cases} \tag{27}$$

Filling this into the KL of $\mathcal{L}_{\mathrm{unm},T,k}$,

$$\mathbb{E}_{q(\mathbf{c}^t \mid \mathbf{c}^0)} \mathrm{KL}[q(\mathbf{c}^s \mid \mathbf{c}^t, \mathbf{c}^0) \| p_{\boldsymbol{\theta}}(\mathbf{c}^s \mid \mathbf{c}^t, \mathbf{c}^0)]$$
$$= \mathbb{E}_{q(\mathbf{c}^t \mid \mathbf{c}^0)} \sum_{\mathbf{c}^0 \succeq \mathbf{c}^s \succeq \mathbf{c}^t} q(\mathbf{c}^s \mid \mathbf{c}^t, \mathbf{c}^0) \log \frac{q(\mathbf{c}^s \mid \mathbf{c}^t, \mathbf{c}^0)}{\sum_{\tilde{\mathbf{c}}^0} p_{\boldsymbol{\theta}}(\tilde{\mathbf{c}}^0 \mid \mathbf{c}^t) q(\mathbf{c}^s \mid \mathbf{c}^t, \tilde{\mathbf{c}}^0)} \tag{28}$$

Now because of the *carry-over unmasking* assumption, we know that the only $\tilde{\mathbf{c}}^0$'s getting positive probabilities are those that extend $\mathbf{c}^t$. Focusing just on the log-ratio above and using Eq. 27 and Eq. 12 we have

$$\log \frac{\left(\frac{1-\alpha_s}{1-\alpha_t}\right)^{|M_{\mathbf{c}^s}|} \left(\frac{\alpha_s - \alpha_t}{1-\alpha_t}\right)^{|M_{\mathbf{c}^t}| - |M_{\mathbf{c}^s}|}}{\sum_{\tilde{\mathbf{c}}^0 \succeq \mathbf{c}^t} p_{\boldsymbol{\theta}}(\tilde{\mathbf{c}}^0 \mid \mathbf{c}^t) \left(\frac{1-\alpha_s}{1-\alpha_t}\right)^{|M_{\mathbf{c}^s}|} \left(\frac{\alpha_s - \alpha_t}{1-\alpha_t}\right)^{|M_{\mathbf{c}^t}| - |M_{\mathbf{c}^s}|} \prod_{i \in U_{\mathbf{c}^s} \setminus U_{\mathbf{c}^t}} \mathbb{1}[\tilde{c}_i^0 = c_i^0]}$$

$$= -\log \sum_{\tilde{\mathbf{c}}^0 \succeq \mathbf{c}^t} p_{\boldsymbol{\theta}}(\tilde{\mathbf{c}}^0 \mid \mathbf{c}^t) \prod_{i \in U_{\mathbf{c}^s} \setminus U_{\mathbf{c}^t}} \mathbb{1}[\tilde{c}_i^0 = c_i^0] = -\log \sum_{\tilde{\mathbf{c}}^0 \succeq \mathbf{c}^s} p_{\boldsymbol{\theta}}(\tilde{\mathbf{c}}^0 \mid \mathbf{c}^t),$$

since the ratio's involving $\alpha_t$ and $\alpha_s$ are independent of $\tilde{\mathbf{c}}^0$ and can be moved out of the sum, dividing away. Then note that $\prod_{i \in U_{\mathbf{c}^s} \setminus U_{\mathbf{c}^t}} \mathbb{1}[\tilde{c}_i^0 = c_i^0]$ also requires that $\tilde{\mathbf{c}}^0$ extends $\mathbf{c}^s$.

Giving the denoising loss:

$$\mathcal{L}_{\text{unm},T,k} = \mathbb{E}_{q(\mathbf{c}^t | \mathbf{c}^0)} \sum_{\mathbf{c}^0 \succeq \mathbf{c}^s \succeq \mathbf{c}^t} -q(\mathbf{c}^s \mid \mathbf{c}^t, \mathbf{c}^0) \log \sum_{\tilde{\mathbf{c}}^0 \succeq \mathbf{c}^s} p_{\boldsymbol{\theta}}(\tilde{\mathbf{c}}^0 \mid \mathbf{c}^t) \tag{29}$$

$$= \mathbb{E}_{q(\mathbf{c}^s, \mathbf{c}^t | \mathbf{c}^0)} - \log \sum_{\tilde{\mathbf{c}}^0 \succeq \mathbf{c}^s} p_{\boldsymbol{\theta}}(\tilde{\mathbf{c}}^0 \mid \mathbf{c}^t) = \mathbb{E}_{q(\mathbf{c}^s, \mathbf{c}^t | \mathbf{c}^0)} - \log p_{\boldsymbol{\theta}}(\tilde{\mathbf{c}}_{U_{\mathbf{c}^s}}^0 = \mathbf{c}_{U_{\mathbf{c}^s}}^t \mid \mathbf{c}^t) \tag{30}$$

$$= \mathbb{E}_{q(\mathbf{c}^s, \mathbf{c}^t | \mathbf{c}^0)} - \log p_{\boldsymbol{\theta}}(\tilde{\mathbf{c}}_{U_{\mathbf{c}^t}}^0 = \mathbf{c}_{U_{\mathbf{c}^t}}^t, \tilde{\mathbf{c}}_{U_{\mathbf{c}^s} \setminus U_{\mathbf{c}^t}}^0 = \mathbf{c}_{U_{\mathbf{c}^s} \setminus U_{\mathbf{c}^t}}^s \mid \mathbf{c}^t) \tag{31}$$

$$= \mathbb{E}_{q(\mathbf{c}^s, \mathbf{c}^t | \mathbf{c}^0)} - \log p_{\boldsymbol{\theta}}(\tilde{\mathbf{c}}_{U_{\mathbf{c}^t}}^0 = \mathbf{c}_{U_{\mathbf{c}^t}}^t \mid \mathbf{c}^t) p_{\boldsymbol{\theta}}(\tilde{\mathbf{c}}_{U_{\mathbf{c}^s} \setminus U_{\mathbf{c}^t}}^0 = \mathbf{c}_{U_{\mathbf{c}^s} \setminus U_{\mathbf{c}^t}}^s \mid \mathbf{c}^t) \tag{32}$$

$$= \mathbb{E}_{q(\mathbf{c}^s, \mathbf{c}^t | \mathbf{c}^0)} - \log p_{\boldsymbol{\theta}}(\tilde{\mathbf{c}}_{U_{\mathbf{c}^s} \setminus U_{\mathbf{c}^t}}^0 = \mathbf{c}_{U_{\mathbf{c}^s} \setminus U_{\mathbf{c}^t}}^s \mid \mathbf{c}^t), \tag{33}$$

where we use the carry-over unmasking assumption twice. In Eq. 33, we use that $p_{\boldsymbol{\theta}}(\tilde{\mathbf{c}}_{U_{\mathbf{c}^t}}^0 = \mathbf{c}_{U_{\mathbf{c}^t}}^t \mid \mathbf{c}^t) = 1$ because $p_{\boldsymbol{\theta}}(\tilde{c}_i^0 = c_i^t \mid \mathbf{c}^t) = 1$ for all $i \in U_{\mathbf{c}^t}$, and so the joint over the variables $\tilde{\mathbf{c}}_{U_{\mathbf{c}^t}}^0$ must also be deterministic and return $\mathbf{c}_{U_{\mathbf{c}^t}}^t$. Similarly, in Eq. 32, we use that $\tilde{\mathbf{c}}_{U_{\mathbf{c}^t}}^0$ is conditionally independent of $\tilde{\mathbf{c}}_{U_{\mathbf{c}^s} \setminus U_{\mathbf{c}^t}}^0$ given $\mathbf{c}^t$ since the support of $\tilde{\mathbf{c}}_{U_{\mathbf{c}^t}}^0$ has only one element. $\qquad \square$

Combining Eq. 24 and Lemmas C.2 and C.3, we get the discrete-time loss:

$$\mathcal{L}_T^{\text{MDM}} = \mathbb{E}_{q(\mathbf{c}^{\frac{1}{T}} | \mathbf{c}^0)} [-\log p(\tilde{\mathbf{c}}^0 = \mathbf{c}^0 \mid \mathbf{c}^{\frac{1}{T}})] +$$
$$\sum_{k=2}^{T} \mathbb{E}_{q(\mathbf{c}^s, \mathbf{c}^t | \mathbf{c}^0)} \left[ -\log p_{\boldsymbol{\theta}}(\tilde{\mathbf{c}}_{U_{\mathbf{c}^s} \setminus U_{\mathbf{c}^t}}^0 = \mathbf{c}_{U_{\mathbf{c}^s} \setminus U_{\mathbf{c}^t}}^s \mid \mathbf{c}^t) \right]. \tag{34}$$

$p_{\boldsymbol{\theta}}(\tilde{\mathbf{c}}_{U_{\mathbf{c}^s} \setminus U_{\mathbf{c}^t}}^0 = \mathbf{c}_{U_{\mathbf{c}^s} \setminus U_{\mathbf{c}^t}}^0 \mid \mathbf{c}^t)$ is the marginal probability of the newly unmasked dimensions in $\mathbf{c}^s$: $U_{\mathbf{c}^s} \setminus U_{\mathbf{c}^t}$. Therefore, computing the discrete-time loss requires being able to be compute conditional marginal distributions over multiple variables. Of course, this is tractable for fully factorised distributions, in which case it's just a product of individual marginals [68]. This loss can be estimated by sampling pairs $\mathbf{c}^s$ and $\mathbf{c}^t$, and can be further simplified depending on the form of $p_{\boldsymbol{\theta}}$.

Next, we consider $\mathcal{L}_T$ as $T \to \infty$. We will show that this allows us to marginalise out $\mathbf{c}^s$, reducing the variance. We will do this by considering the two loss terms individually, and letting $T \to \infty$.

**Lemma C.4.** *Using the assumptions of Theorem C.1,*

$$\lim_{T \to \infty} \mathcal{L}_{\text{rec},T} = 0 \tag{35}$$

*Proof.* Recall that in discrete time this is equal to (see Lemma C.2)

$$\mathcal{L}_{\text{rec},T} = \mathbb{E}_{q(\mathbf{c}^{\frac{1}{T}} | \mathbf{c}^0)} - \log p(\tilde{\mathbf{c}}^0 = \mathbf{c}^0 \mid \mathbf{c}^{\frac{1}{T}}) \tag{36}$$

Note that $q(c_i^{1/T} = \text{m} \mid c_i^0) = 1 - \alpha_{1/T}$. Then, $\lim_{T \to \infty} \alpha_{1/T} = \lim_{t \to 0} \alpha_t = 1$ by continuity and monotonicity of $\alpha_t$, giving $\lim_{T \to \infty} q(c_i^{1/T} = \text{m} \mid c_i^0) = 0$. Therefore, asymptotically, for all $\mathbf{c}^{1/T} \neq \mathbf{c}^0$, we are left with a term that tends to 0 and a constant term independent of $T$, meaning the only relevant element of the sum is $\mathbf{c}^{1/T} = \mathbf{c}^0$:

$$\lim_{T \to \infty} \sum_{\mathbf{c}^{1/T}} -q(\mathbf{c}^{1/T} \mid \mathbf{c}^0) \log \sum_{\tilde{\mathbf{c}}^0} p_{\boldsymbol{\theta}}(\tilde{\mathbf{c}}^0 \mid \mathbf{c}^{1/T}) = \lim_{T \to \infty} -\log \sum_{\tilde{\mathbf{c}}^0} p_{\boldsymbol{\theta}}(\tilde{\mathbf{c}}^0 \mid \mathbf{c}^0) \tag{37}$$

$$= -\log \sum_{\tilde{\mathbf{c}}^0} p_{\boldsymbol{\theta}}(\tilde{\mathbf{c}}^0 \mid \mathbf{c}^0) = \log 1 = 0 \tag{38}$$

where we use the carry-over unmasking assumption to get the last equality. $\qquad \square$

**Lemma C.5.** *Using the assumptions of Theorem C.1,*

$$\lim_{T \to \infty} \sum_{k=2}^{\infty} \mathcal{L}_{\text{unm},T,k} = \mathbb{E}_{t \sim (0,1] q(\mathbf{c}^t | \mathbf{c}^0)} \left[ \frac{\alpha_t'}{1 - \alpha_t} \sum_{i \in M_{\mathbf{c}^t}} \log p_{\boldsymbol{\theta}}(\tilde{c}_i^0 = c_i^0 \mid \mathbf{c}^t) \right] \tag{39}$$

*Proof.* Instead of having a sum over $T-1$ timesteps, each computing a KL, we will now sample some $t \sim \{\frac{2}{T}, ..., 1\}$, redefining $s := t - \frac{1}{T}$. Then, we will weight the result by $T-1$. Using Lemma C.3,

$$\lim_{T \to \infty} \mathcal{L}_{\text{unm},T}$$

$$= \lim_{T \to \infty} \mathbb{E}_{t \sim \{\frac{2}{T}, ..., 1\}} \mathbb{E}_{q(\mathbf{c}^t | \mathbf{c}^0)}[(T-1)\text{KL}[q(\mathbf{c}^s \mid \mathbf{c}^t, \mathbf{c}^0) \| p_{\boldsymbol{\theta}}(\mathbf{c}^s \mid \mathbf{c}^t)]] \tag{40}$$

$$= \mathbb{E}_{t \sim (0,1]} \mathbb{E}_{q(\mathbf{c}^t | \mathbf{c}^0)}[- \sum_{\mathbf{c}^0 \succeq \mathbf{c}^s \succeq \mathbf{c}^t} \lim_{T \to \infty} (T-1)q(\mathbf{c}^s \mid \mathbf{c}^t, \mathbf{c}^0) \log p_{\boldsymbol{\theta}}(\tilde{\mathbf{c}}_{U_{\mathbf{c}^s} \setminus U_{\mathbf{c}^t}}^0 = \mathbf{c}_{U_{\mathbf{c}^s} \setminus U_{\mathbf{c}^t}}^s \mid \mathbf{c}^t)]. \tag{41}$$

Assuming that $\mathbf{c}^0 \succeq \mathbf{c}^s \succeq \mathbf{c}^t$, recall $q(\mathbf{c}^s \mid \mathbf{c}^t, \mathbf{c}^0) = \left(\frac{1-\alpha_s}{1-\alpha_t}\right)^{|M_{\mathbf{c}^s}|} \left(\frac{\alpha_s - \alpha_t}{1-\alpha_t}\right)^{|M_{\mathbf{c}^t}| - |M_{\mathbf{c}^s}|}$, and assume at least one dimension becomes unmasked: $|M_{\mathbf{c}^t}| - |M_{\mathbf{c}^s}| > 0$. Then, using that $\lim_{T \to \infty} \frac{1-\alpha_s}{1-\alpha_t} = 1$, we get

$$\lim_{T \to \infty} (T-1)q(\mathbf{c}^s \mid \mathbf{c}^t, \mathbf{c}^0) = \lim_{T \to \infty} T \left(\frac{1-\alpha_s}{1-\alpha_t}\right)^{|M_{\mathbf{c}^s}|} \left(\frac{\alpha_s - \alpha_t}{1-\alpha_t}\right)^{|M_{\mathbf{c}^t}| - |M_{\mathbf{c}^s}|} \tag{42}$$

$$= \lim_{T \to \infty} \frac{T(\alpha_s - \alpha_t)}{1 - \alpha_t} \left(\frac{\alpha_s - \alpha_t}{1 - \alpha_t}\right)^{|M_{\mathbf{c}^t}| - |M_{\mathbf{c}^s}| - 1} \tag{43}$$

Next, using that $\alpha_t$ is differentiable, consider the first-order Taylor expansion of $\alpha$ around $t$ to evaluate $\alpha_s$: $\alpha_s = \alpha_t - \frac{1}{T}\alpha_t' + O(\frac{1}{T^2})$. Then $T(\alpha_s - \alpha_t) = T(\alpha_t - \frac{1}{T}\alpha_t' + O(\frac{1}{T^2}) - \alpha_t) = -\alpha_t' + O(\frac{1}{T})$. And so $\lim_{T \to \infty} T(\alpha_s - \alpha_t) = -\alpha_t'$.

Now if $|M_{\mathbf{c}^t}| - |M_{\mathbf{c}^s}| \geq 2$, then the following term appears: $T(\alpha_s - \alpha_t)^2 = T(-\frac{1}{T}\alpha_t' + O(\frac{1}{T^2}))^2 = T(\frac{\alpha_t'^2}{T^2} + O(\frac{1}{T^3})) = \frac{\alpha_t'^2}{T} + O(\frac{1}{T^2})$. And so $\lim_{T \to \infty} T(\alpha_s - \alpha_t)^2 = \lim_{T \to \infty} \frac{\alpha_t'^2}{T} + O(\frac{1}{T^2}) = 0$.

Therefore, the only $\mathbf{c}^s$ in the sum with a non-zero contribution are where $|M_{\mathbf{c}^t}| - |M_{\mathbf{c}^s}| \leq 1$, that is, when $\mathbf{c}^s$ unmasks at most one dimension of $\mathbf{c}^t$. When no dimensions are unmasked, $q(\mathbf{c}^s \mid \mathbf{c}^t, \mathbf{c}^0) = 1$, and if $\mathbf{c}^s$ unmasks one dimension, we have $q(\mathbf{c}^s \mid \mathbf{c}^t, \mathbf{c}^0) = -\alpha_t'$.

If $\mathbf{c}^s$ does not unmask any dimensions, then there are no variables in $\tilde{\mathbf{c}}_{U_{\mathbf{c}^s} \setminus U_{\mathbf{c}^t}}^0$ to compute the probability over in $-\log p_{\boldsymbol{\theta}}(\tilde{\mathbf{c}}_{U_{\mathbf{c}^s} \setminus U_{\mathbf{c}^t}}^0 = \mathbf{c}_{U_{\mathbf{c}^s} \setminus U_{\mathbf{c}^t}}^s \mid \mathbf{c}^t)$, giving probability 1 and a term equal to 0. Next, if $\mathbf{c}^s$ unmasks only dimension $i \in M_{\mathbf{c}^t}$ such that $c_i^s = c_i^0$, then $p_{\boldsymbol{\theta}}(\tilde{\mathbf{c}}_{U_{\mathbf{c}^s} \setminus U_{\mathbf{c}^t}}^0 = \mathbf{c}_{U_{\mathbf{c}^s} \setminus U_{\mathbf{c}^t}}^s \mid \mathbf{c}^t) = p_{\boldsymbol{\theta}}(\tilde{c}_i^0 = c_i^0 \mid \mathbf{c}^t)$.

Therefore,

$$\lim_{T \to \infty} \sum_{k=2}^{\infty} \mathcal{L}_{\text{unm},T,k} = \mathcal{L}^{\text{MDM}} = \mathbb{E}_{t \sim (0,1]q(\mathbf{c}^t | \mathbf{c}^0)}\left[\frac{\alpha_t'}{1 - \alpha_t} \sum_{i \in M_{\mathbf{c}^t}} \log p_{\boldsymbol{\theta}}(\tilde{c}_i^0 = c_i^0 \mid \mathbf{c}^t)\right] \tag{44}$$

$\square$

Summing Lemmas C.4 and C.5 completes the proof of Theorem C.1. $\square$

# D  Neurosymbolic diffusion models: formal definition and NELBO derivation

In this section, we will formally define derive the NELBO for neurosymbolic diffusion model. Throughout this section, we assume the same notation as in Section A. We refer to the graphical model in Fig. 2 to set up the model and the notation.

## D.1  Formal model definition

First, we will define the discrete-time data log-likelihood. We let $\mathbf{C}$ be the trajectory of partially masked concepts over timesteps $\mathbf{c}^0, \mathbf{c}^{\frac{1}{T}}, \mathbf{c}^{\frac{2}{T}}, \ldots, \mathbf{c}^{\frac{T-1}{T}}, \mathbf{c}^1$, and similarly $\mathbf{Y}_{\setminus \{0\}}$ is the trajectory

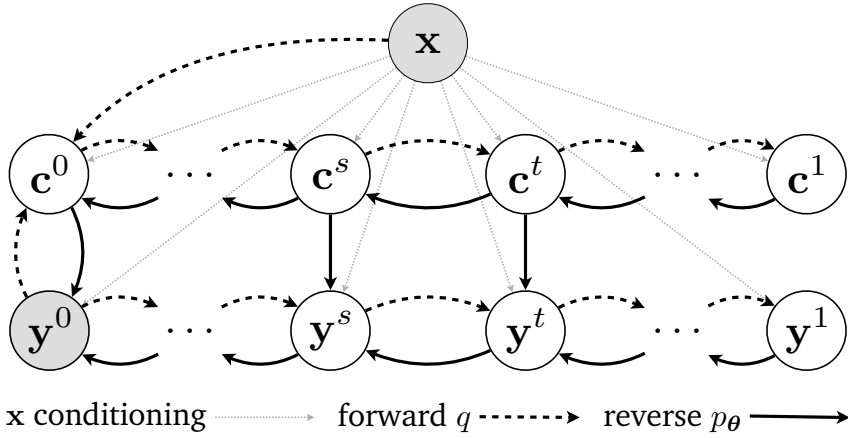

x conditioning ┄┄┄⟶  forward $q$ ┄┄┄⟶  reverse $p_{\boldsymbol{\theta}}$ ⟶

Figure 2: **Probabilistic graphical model for neurosymbolic diffusion model.** The forward process $q$, indicated by striped arrows, masks both concepts $\mathbf{c}$ and outputs $\mathbf{y}$. Since only $\mathbf{y}^0$ is observed, a variational distribution $q_{\boldsymbol{\theta}}$ has to predict $\mathbf{c}^0$ from $\mathbf{y}^0$ and $\mathbf{x}$. The reverse process, with regular arrows, unmasks both concepts $\mathbf{c}$ and outputs $\mathbf{y}$, transforming concepts into outputs at every time step.

$\mathbf{y}^{\frac{1}{T}}, \mathbf{y}^{\frac{2}{T}}, \ldots, \mathbf{y}^{\frac{T-1}{T}}, \mathbf{y}^1$ Marginalising out all latent variables according to the graphical model in the bottom of Fig. 2, we define the data log-likelihood $p_{\boldsymbol{\theta}}^{\text{NESYDM}}(\mathbf{y}^0 \mid \mathbf{x})$ of outputs $\mathbf{y}$ given inputs $\mathbf{x}$ as:

$$p_{\boldsymbol{\theta}}^{\text{NESYDM}}(\mathbf{y}^0 \mid \mathbf{x}) := \sum_{\mathbf{Y}_{\backslash\{0\}}} \sum_{\mathbf{C}} q(\mathbf{c}^1, \mathbf{y}^1) \prod_{k=1}^{T} p_{\boldsymbol{\theta}}(\mathbf{c}^{s(k)} \mid \mathbf{c}^{t(k)}, \mathbf{x}) p_{\boldsymbol{\theta}}(\mathbf{y}^{s(k)} \mid \mathbf{c}^{s(k)}, \mathbf{y}^{t(k)}, \mathbf{x}). \quad (45)$$

Here, $p_{\boldsymbol{\theta}}(\mathbf{c}^s \mid \mathbf{c}^t, \mathbf{x})$ and $p_{\boldsymbol{\theta}}(\mathbf{y}^s \mid \mathbf{c}^s, \mathbf{y}^t, \mathbf{x})$ are defined as in Section 3.1.

First, we define the conditional program $\varphi_{\mathbf{y}^t}$ as the program $\varphi$ that maps concepts to outputs, but always returns $y_i^t$ if dimension $i$ is unmasked in $\mathbf{y}^t$. To be precise,

$$\varphi_{\mathbf{y}^t}(\tilde{\mathbf{c}}^0)_i = \begin{cases} \varphi(\tilde{\mathbf{c}}^0)_i & \text{if } y_i^t = \mathrm{m} \\ y_i^t & \text{if } y_i^t \neq \mathrm{m} \end{cases}. \quad (46)$$

We need this definition in Eq. 4 to ensure the output unmasking model satisfies the carry-over unmasking assumption from Theorem C.1.

Next, we define

$$p_{\boldsymbol{\theta}}(\tilde{\mathbf{y}}^0 \mid \mathbf{c}^s, \mathbf{y}^t, \mathbf{x}) := \sum_{\tilde{\mathbf{c}}^0} p_{\boldsymbol{\theta}}(\tilde{\mathbf{c}}^0 \mid \mathbf{c}^s, \mathbf{x}) \mathbb{1}[\varphi_{\mathbf{y}^t}(\tilde{\mathbf{c}}^0) = \tilde{\mathbf{y}}^0] \quad (47)$$

as the *output unmasking model* such that the MDM defined as $\sum_{\tilde{\mathbf{y}}^0} p_{\boldsymbol{\theta}}(\tilde{\mathbf{y}}^0 \mid \mathbf{c}^s, \mathbf{x}) q(\mathbf{y}^s \mid \mathbf{y}^t, \tilde{\mathbf{y}}^0)$ is equal to $p_{\boldsymbol{\theta}}(\mathbf{y}^s \mid \mathbf{c}^s, \mathbf{y}^t, \mathbf{x})$ as defined in Eq. 4:

$$\sum_{\tilde{\mathbf{y}}^0} p_{\boldsymbol{\theta}}(\tilde{\mathbf{y}}^0 \mid \mathbf{c}^s, \mathbf{x}) q(\mathbf{y}^s \mid \mathbf{y}^t, \tilde{\mathbf{y}}^0) = \sum_{\tilde{\mathbf{y}}^0} \sum_{\tilde{\mathbf{c}}^0} p_{\boldsymbol{\theta}}(\tilde{\mathbf{c}}^0 \mid \mathbf{c}^s, \mathbf{x}) \mathbb{1}[\varphi_{\mathbf{y}^t}(\tilde{\mathbf{c}}^0) = \tilde{\mathbf{y}}^0] q(\mathbf{y}^s \mid \mathbf{y}^t, \tilde{\mathbf{y}}^0) \quad (48)$$

$$= \sum_{\tilde{\mathbf{c}}^0} p_{\boldsymbol{\theta}}(\tilde{\mathbf{c}}^0 \mid \mathbf{c}^s, \mathbf{x}) q(\mathbf{y}^s \mid \mathbf{y}^t, \varphi_{\mathbf{y}^t}(\tilde{\mathbf{c}}^0)) = p_{\boldsymbol{\theta}}(\mathbf{y}^s \mid \mathbf{c}^s, \mathbf{y}^t, \mathbf{x}). \quad (49)$$

Note that $p_{\boldsymbol{\theta}}(\mathbf{y}^s \mid \mathbf{c}^s, \mathbf{y}^t, \mathbf{x})$ does not decompose into a product of marginals, requiring the new results in Section C rather than the standard MDM NELBO derivation.

To be able to use Theorem C.1 and the other lemmas in Section C, we need to ensure that the output unmasking model satisfies the assumptions of Theorem C.1. Since $\varphi_{\mathbf{y}^t}$ maps completely unmasked concepts to completely unmasked outputs, it satisfies zero masking probabilities. Further, the carry-over unmasking assumption is satisfied, since for any unmasked dimension $i \in U_{\mathbf{y}^t}$ and any concept $\tilde{\mathbf{c}}^0$, $\varphi_{\mathbf{y}^t}(\tilde{\mathbf{c}}^0)_i = \mathbf{y}_i^t$ by Eq. 46 and hence $p_{\boldsymbol{\theta}}(\tilde{y}_i^0 = y_i^t \mid \mathbf{c}^s, \mathbf{y}^t, \mathbf{x}) = 1$. Importantly, the carry-over unmasking assumption would *not* hold if we used $\varphi(\tilde{\mathbf{c}}^0)$ instead of $\varphi_{\mathbf{y}^t}(\tilde{\mathbf{c}}^0)$ in Eq. 47, and we would not have been able to use the results in Section C.

## D.2 NELBO derivation

**Theorem D.1.** *Let $p_{\boldsymbol{\theta}}(\tilde{\mathbf{c}}^0 \mid \mathbf{c}^t, \mathbf{x})$ be a concept unmasking model with zero masking probabilities and carry-over unmasking as defined in Theorem C.1, $\varphi : [V_{\mathbf{c}}]^C \to [V_{\mathbf{y}}]^Y$ be a given program, $q_{\boldsymbol{\theta}}(\mathbf{c}^0 \mid \mathbf{y}^0, \mathbf{x})$ be a variational distribution, and $\alpha_t$ be a noising schedule. Then we have that the data log-likelihood as $T \to \infty$ is bounded as $\lim_{T \to \infty} -\log p_{\boldsymbol{\theta}}^{\text{NESYDM}}(\mathbf{y}^0 \mid \mathbf{x}) \leq \mathcal{L}_{\text{NESYDM}}$, where*

$$
\begin{aligned}
\mathcal{L}_{\text{NESYDM}} = \mathbb{E}_{t \sim [0,1], q_{\boldsymbol{\theta}}(\mathbf{c}^0, \mathbf{c}^t \mid \mathbf{x}, \mathbf{y}^0)} \bigg[ & \underbrace{\frac{\alpha_t'}{1 - \alpha_t} \sum_{i \in M_{\mathbf{c}^t}} \log p_{\boldsymbol{\theta}}(\tilde{c}_i^0 = c_i^0 \mid \mathbf{c}^t, \mathbf{x})}_{\mathcal{L}_{\mathbf{c}}:\ \text{Concept unmasking loss}} \\
& + \underbrace{\alpha_t' \sum_{i=1}^{Y} \log \sum_{\tilde{\mathbf{c}}^0} p_{\boldsymbol{\theta}}(\tilde{\mathbf{c}}^0 \mid \mathbf{c}^t, \mathbf{x}) \mathbb{1}[\varphi(\tilde{\mathbf{c}}^0)_i = y_i^0]}_{\mathcal{L}_{\mathbf{y}}:\ \text{Output unmasking loss}} \bigg] - \underbrace{\text{H}[q_{\boldsymbol{\theta}}(\mathbf{c}^0 \mid \mathbf{y}^0, \mathbf{x})]}_{\mathcal{L}_{\text{H}[q]}:\ \text{Variational entropy}}
\end{aligned}
\tag{50}
$$

*Proof.*

$$
\begin{aligned}
& -\log p_{\boldsymbol{\theta}}^{\text{NESYDM}}(\mathbf{y}^0 \mid \mathbf{x}) \\
& = -\log \sum_{\mathbf{Y}_{\backslash 0}} \sum_{\mathbf{C}} p_{\boldsymbol{\theta}}(\mathbf{C}, \mathbf{Y} \mid \mathbf{x}) \leq -\mathbb{E}_{q_{\boldsymbol{\theta}}(\mathbf{C}, \mathbf{Y}_{\backslash 0} \mid \mathbf{y}^0, \mathbf{x})} \left[ \log \frac{p_{\boldsymbol{\theta}}(\mathbf{C}, \mathbf{Y} \mid \mathbf{x})}{q_{\boldsymbol{\psi}}(\mathbf{C}, \mathbf{Y}_{\backslash 0} \mid \mathbf{y}^0, \mathbf{x})} \right]
\end{aligned}
\tag{51}
$$

Next, we again use the trick from Eq. 21, both for $\mathbf{C}$ and $\mathbf{Y}$, and we expand the model $p_{\boldsymbol{\theta}}$ following the graphical model in Fig. 2:

$$
= \mathbb{E}_{q_{\boldsymbol{\theta}}(\mathbf{C}, \mathbf{Y}_{\backslash 0} \mid \mathbf{y}^0, \mathbf{x})} \left[ \log \frac{p_{\boldsymbol{\theta}}(\mathbf{c}^0, \mathbf{y}^0 \mid \mathbf{c}^{\frac{1}{T}}, \mathbf{y}^{\frac{1}{T}}, \mathbf{x}) p(\mathbf{c}^1, \mathbf{y}^1)}{q_{\boldsymbol{\theta}}(\mathbf{c}^1, \mathbf{c}^0 \mid \mathbf{x}, \mathbf{y}^0) q(\mathbf{y}^1 \mid \mathbf{y}^0)} \frac{\prod_{k=2}^{T} p_{\boldsymbol{\theta}}(\mathbf{c}^s, \mathbf{y}^s \mid \mathbf{c}^t, \mathbf{y}^t, \mathbf{x})}{\prod_{t=2}^{T} q(\mathbf{c}^s \mid \mathbf{c}^t, \mathbf{c}^0) q(\mathbf{y}^s \mid \mathbf{y}^t, \mathbf{y}^0)} \right]
$$

$$
= \mathbb{E}_{q_{\boldsymbol{\theta}}(\mathbf{c}^0 \mid \mathbf{x}, \mathbf{y}^0)} \bigg[ \underbrace{\mathbb{E}_{q(\mathbf{y}^{\frac{1}{T}} \mid \mathbf{y}^0)} \left[ -\log p_{\boldsymbol{\theta}}(\mathbf{y}^0 \mid \mathbf{c}^0, \mathbf{y}^{\frac{1}{T}}, \mathbf{x}) \right]}_{\mathcal{L}_{\text{rec}, \mathbf{y}, T}:\ \mathbf{y}^0 \text{ reconstruction}} - \underbrace{\mathbb{E}_{q(\mathbf{c}^{\frac{1}{T}} \mid \mathbf{c}^0)} \left[ \log \frac{p_{\boldsymbol{\theta}}(\mathbf{c}^0 \mid \mathbf{c}^{\frac{1}{T}}, \mathbf{x})}{q_{\boldsymbol{\theta}}(\mathbf{c}^0 \mid \mathbf{x}, \mathbf{y}^0)} \right]}_{\mathcal{L}_{\text{rec}, \mathbf{c}, T}:\ \mathbf{c}^0 \text{ reconstruction}} +
$$

$$
\sum_{k=2}^{T} \underbrace{\mathbb{E}_{q(\mathbf{c}^t \mid \mathbf{c}^0)} \left[ \text{KL}[q(\mathbf{c}^s \mid \mathbf{c}^t, \mathbf{c}^0) \| p_{\boldsymbol{\theta}}(\mathbf{c}^s \mid \mathbf{c}^t, \mathbf{x})] \right]}_{\mathcal{L}_{\text{unm}, \mathbf{c}, T, k}:\ \mathbf{c} \text{ unmasking}} +
$$

$$
\underbrace{\mathbb{E}_{q(\mathbf{c}^s, \mathbf{y}^t \mid \mathbf{c}^0, \mathbf{y}^0)} \left[ \text{KL}[q(\mathbf{y}^s \mid \mathbf{y}^t, \mathbf{y}^0) \| p_{\boldsymbol{\theta}}(\mathbf{y}^s \mid \mathbf{y}^t, \mathbf{c}^s, \mathbf{x})] \right]}_{\mathcal{L}_{\text{unm}, \mathbf{y}, T, k}:\ \mathbf{y} \text{ unmasking}} \bigg] + C
$$

where $C = \mathbb{E}_{q(\mathbf{c}^1, \mathbf{y}^1 \mid \mathbf{y}^0, \mathbf{x})} \log \frac{p(\mathbf{c}^1, \mathbf{y}^1)}{q_{\boldsymbol{\theta}}(\mathbf{c}^1, \mathbf{y}^1 \mid \mathbf{x}, \mathbf{y}^0)}$ is a constant and equal to 0 if $p(\mathbf{c}^1, \mathbf{y}^1) = \mathbb{1}[\mathbf{c}^1 = \mathbf{m} \wedge \mathbf{y}^1 = \mathbf{m}]$, which is true by assumption.

**Discrete-time NELBO.** Next, we use Lemma C.2 to rewrite $\mathcal{L}_{\text{rec}, \mathbf{y}, T}$ and $\mathcal{L}_{\text{rec}, \mathbf{c}, T}$. Then, the first two terms are the reconstruction losses for $\mathbf{y}^0$ and $\mathbf{c}^0$ respectively, and the third term is the entropy of the variational distribution.

$$
\begin{aligned}
\mathcal{L}_{\text{rec}, \mathbf{y}, T} + \mathcal{L}_{\text{rec}, \mathbf{c}, T} = \mathbb{E}_{q_{\boldsymbol{\theta}}(\mathbf{c}^0 \mid \mathbf{x}, \mathbf{y}^0)} \bigg[ & \mathbb{E}_{q(\mathbf{y}^{\frac{1}{T}} \mid \mathbf{y}^0)} \left[ -\log p_{\boldsymbol{\theta}}(\tilde{\mathbf{y}}^0 = \mathbf{y}^0 \mid \mathbf{c}^0, \mathbf{y}^{\frac{1}{T}}, \mathbf{x}) \right] + \\
& \mathbb{E}_{q(\mathbf{c}^{\frac{1}{T}} \mid \mathbf{c}^0)} \left[ -\log p_{\boldsymbol{\theta}}(\tilde{\mathbf{c}}^0 = \mathbf{c}^0 \mid \mathbf{c}^{\frac{1}{T}}, \mathbf{x}) \right] + \log q_{\boldsymbol{\theta}}(\mathbf{c}^0 \mid \mathbf{y}^0, \mathbf{x}) \bigg],
\end{aligned}
\tag{52}
$$

Similarly, using Lemma C.3, we have the **concept unmasking loss** as

$$
\sum_{k=2}^{T} \mathcal{L}_{\text{unm}, \mathbf{c}, T, k} = \sum_{k=2}^{T} \mathbb{E}_{q_{\boldsymbol{\theta}}(\mathbf{c}^0, \mathbf{c}^s, \mathbf{c}^t \mid \mathbf{x}, \mathbf{y}^0)} \left[ -\log p_{\boldsymbol{\theta}}(\tilde{\mathbf{c}}_{U_{\mathbf{c}^s} \backslash U_{\mathbf{c}^t}}^0 = \mathbf{c}_{U_{\mathbf{c}^s} \backslash U_{\mathbf{c}^t}}^s \mid \mathbf{c}^t, \mathbf{x}) \right]
\tag{53}
$$

For the **output unmasking loss** $\mathcal{L}_{\text{unm},\mathbf{y},T}$, again using Lemma C.3, we have

$$\sum_{k=2}^{T} \mathcal{L}_{\text{unm},\mathbf{y},T,k} = \sum_{k=2}^{T} \mathbb{E}_{q_{\boldsymbol{\theta}}(\mathbf{c}^s,\mathbf{y}^s,\mathbf{y}^t|\mathbf{x},\mathbf{y}^0)}\left[ - \log p_{\boldsymbol{\theta}}(\tilde{\mathbf{y}}_{U_{\mathbf{y}^s}\setminus U_{\mathbf{y}^t}}^0 = \mathbf{y}_{U_{\mathbf{c}^s}\setminus U_{\mathbf{c}^t}}^s \mid \mathbf{c}^s,\mathbf{y}^t,\mathbf{x}) \right]. \tag{54}$$

Summing Eqs. 52 to 54, we get the discrete-time NELBO.

**Continuous-time NELBO.** Using Lemma C.4 twice and adding the **entropy of the variational distribution** in Eq. 52, and then using Lemma C.5 twice, we get the continuous-time NELBO:

$$\mathcal{L}^{\text{NeSyDM}'} = \mathbb{E}_{t\sim[0,1],q_{\boldsymbol{\theta}}(\mathbf{c}^0,\mathbf{c}^t|\mathbf{x},\mathbf{y}^0)}\left[ \frac{\alpha_t'}{1-\alpha_t}\underbrace{\left( \sum_{i\in M_{\mathbf{c}^t}} \log p_{\boldsymbol{\theta}}(\tilde{c}_i^0 = c_i^0 \mid \mathbf{c}^t,\mathbf{x}) + \right.}_{\mathcal{L}_{\mathbf{c}}:\text{ Concept denoising loss}}$$

$$\underbrace{\mathbb{E}_{q(\mathbf{y}^t|\mathbf{y}^0)} \sum_{i\in M_{\mathbf{y}^t}} \log p_{\boldsymbol{\theta}}(\tilde{y}_i^0 = y_i^0 \mid \mathbf{c}^t,\mathbf{y}^t,\mathbf{x}) }_{\mathcal{L}_{\mathbf{y}}:\text{ Output denoising loss}} \left. \right) \right] - \underbrace{\text{H}[q_{\boldsymbol{\theta}}(\mathbf{c}^0 \mid \mathbf{y}^0,\mathbf{x})]}_{\mathcal{L}_{\text{H}[q]}:\text{ Variational entropy}}. \tag{55}$$

Next, we will further simplify the **output unmasking loss** $\mathcal{L}_{\mathbf{y}}$ with a Rao-Blackwellisation to get the form given in Theorem 3.1. Using Eq. 47,

$$\mathcal{L}_{\mathbf{y}} = \mathbb{E}_{t\sim[0,1],q_{\boldsymbol{\theta}}(\mathbf{y}^t,\mathbf{c}^t|\mathbf{x},\mathbf{y}^0)}\left[ \frac{\alpha_t'}{1-\alpha_t} \sum_{i\in M_{\mathbf{y}^t}} \log p_{\boldsymbol{\theta}}(\tilde{y}_i^0 = y_i^0 \mid \mathbf{c}^t,\mathbf{y}^t,\mathbf{x}) \right]$$

$$= \mathbb{E}_{t\sim[0,1],q_{\boldsymbol{\theta}}(\mathbf{y}^t,\mathbf{c}^t|\mathbf{x},\mathbf{y}^0)}\left[ \frac{\alpha_t'}{1-\alpha_t} \sum_{i\in M_{\mathbf{y}^t}} \log \sum_{\tilde{\mathbf{y}}^0,\tilde{y}_i^0=y_i^0} \sum_{\tilde{\mathbf{c}}^0} p_{\boldsymbol{\theta}}(\tilde{\mathbf{c}}^0 \mid \mathbf{c}^t,\mathbf{x})\mathbb{1}[\varphi_{\mathbf{y}^t}(\tilde{\mathbf{c}}^0) = \tilde{\mathbf{y}}^0] \right]$$

$$= \mathbb{E}_{t\sim[0,1],q_{\boldsymbol{\theta}}(\mathbf{y}^t,\mathbf{c}^t|\mathbf{x},\mathbf{y}^0)}\left[ \frac{\alpha_t'}{1-\alpha_t} \sum_{i\in M_{\mathbf{y}^t}} \log \sum_{\tilde{\mathbf{c}}^0} p_{\boldsymbol{\theta}}(\tilde{\mathbf{c}}^0 \mid \mathbf{c}^t,\mathbf{x}) \sum_{\tilde{\mathbf{y}}^0,\tilde{y}_i^0=y_i^0} \mathbb{1}[\varphi_{\mathbf{y}^t}(\tilde{\mathbf{c}}^0) = \tilde{\mathbf{y}}^0] \right]$$

$$= \mathbb{E}_{t\sim[0,1],q_{\boldsymbol{\theta}}(\mathbf{c}^t|\mathbf{x},\mathbf{y}^0)q(\mathbf{y}^t|\mathbf{y}^0)}\left[ \frac{\alpha_t'}{1-\alpha_t} \sum_{i\in M_{\mathbf{y}^t}} \log \sum_{\tilde{\mathbf{c}}^0} p_{\boldsymbol{\theta}}(\tilde{\mathbf{c}}^0 \mid \mathbf{c}^t,\mathbf{x})\mathbb{1}[\varphi_{\mathbf{y}^t}(\tilde{\mathbf{c}}^0)_i = y_i^0] \right],$$

where in the last step we use that only a single $\tilde{\mathbf{y}}^0$ satisfies $\varphi_{\mathbf{y}^t}(\tilde{\mathbf{c}}^0) = \tilde{\mathbf{y}}^0$, and it appears in the sum only if exactly that $\tilde{\mathbf{y}}^0$ has $\tilde{y}_i^0 = y_i^0$, and the conditional independence in Fig. 2 of $\mathbf{y}^t$ and $\mathbf{c}^t$ given $\mathbf{y}^0$.

Next, define the following inductive hypothesis based on the value of $Y$:

$$\mathcal{L}_{\mathbf{y}} = \mathbb{E}_{t\sim(0,1]}\mathbb{E}_{q_{\boldsymbol{\theta}}(\mathbf{c}^t|\mathbf{x},\mathbf{y}^0)}\left[ \alpha_t' \sum_{i=1}^{Y} \log \sum_{\tilde{\mathbf{c}}^0} p_{\boldsymbol{\theta}}(\tilde{\mathbf{c}}^0 \mid \mathbf{c}^t,\mathbf{x})\mathbb{1}[\varphi(\tilde{\mathbf{c}}^0)_i = y_i^0] \right]. \tag{56}$$

**Base case $Y = 1$:** The only elements in the support of $q(\mathbf{y}^t \mid \mathbf{y}^0)$ are $y_1^t = y_1^0$ and $y_1^t = \text{m}$. If it is $y_1^0$ (probability $\alpha_t$), the set of unmasked values is empty, and so the loss is zero. If it is m (probability $1 - \alpha_t$), the only masked dimension is $i = 1$. Furthermore, there are no unmasked dimensions in $\mathbf{y}^t$, hence $\varphi_{\mathbf{y}^t} = \varphi$ and so the loss is

$$\mathcal{L}_{\mathbf{y}} = \mathbb{E}_{t\sim(0,1]}\mathbb{E}_{q_{\boldsymbol{\theta}}(\mathbf{c}^t|\mathbf{x})}\left[ \alpha_t' \log \sum_{\tilde{\mathbf{c}}^0} p_{\boldsymbol{\theta}}(\tilde{\mathbf{c}}^0 \mid \mathbf{c}^t,\mathbf{x})\mathbb{1}[\varphi(\tilde{\mathbf{c}}^0)_1 = y_1^0] \right].$$

**Inductive step $Y > 1$:** Assume the result holds for $Y - 1$. Like in the base case, $q(y_Y^t = y_Y^0 \mid \mathbf{y}^0) = \alpha_t$ and $q(y_Y^t = \text{m} \mid \mathbf{y}^0) = 1 - \alpha_t$. Then, let $\hat{\mathbf{y}}^t$ denote all variables in $\mathbf{y}^t$ except $y_Y^t$, and we assume the inductive hypothesis holds for $\hat{\mathbf{y}}^t$. We again consider the two cases: Either $y_Y^t = y_Y^0$ with

probability $\alpha_t$ or $y_Y^t = \text{m}$ with probability $1 - \alpha_t$.

$$
\begin{aligned}
\mathcal{L}_{\mathbf{y}} =& \mathbb{E}_{t\sim(0,1]}\mathbb{E}_{q_{\boldsymbol{\theta}}(\mathbf{c}^t,\mathbf{y}^t|\mathbf{x},\mathbf{y}^0)}\left[\frac{\alpha'_t}{1-\alpha_t}\sum_{i\in M_{\mathbf{y}^t}}\log\sum_{\tilde{\mathbf{c}}^0}p_{\boldsymbol{\theta}}(\tilde{\mathbf{c}}^0\mid\mathbf{c}^t,\mathbf{x})\mathbb{1}[\varphi_{\mathbf{y}^t}(\tilde{\mathbf{c}}^0)_i = y_i^0]\right]\\
=& \mathbb{E}_{t\sim(0,1]}\mathbb{E}_{q_{\boldsymbol{\theta}}(\mathbf{c}^t|\mathbf{x})}\left[\sum_{\hat{\mathbf{y}}^t}q(\hat{\mathbf{y}}^t\mid\mathbf{y}^0)\left(\frac{\alpha_t\alpha'_t}{1-\alpha_t}\sum_{i\in M_{\hat{\mathbf{y}}^t}}\log\sum_{\tilde{\mathbf{c}}^0}p_{\boldsymbol{\theta}}(\tilde{\mathbf{c}}^0\mid\mathbf{c}^t,\mathbf{x})\mathbb{1}[\varphi_{\hat{\mathbf{y}}^t}(\tilde{\mathbf{c}}^0)_i = y_i^0]\right.\right.\\
&+ \left.\left.\frac{(1-\alpha_t)\alpha'_t}{1-\alpha_t}\sum_{i\in M_{\hat{\mathbf{y}}^t}\cup\{Y\}}\log\sum_{\tilde{\mathbf{c}}^0}p_{\boldsymbol{\theta}}(\tilde{\mathbf{c}}^0\mid\mathbf{c}^t,\mathbf{x})\mathbb{1}[\varphi_{\hat{\mathbf{y}}^t}(\tilde{\mathbf{c}}^0)_i = y_i^0]\right)\right]
\end{aligned}
$$

Note now that the second term contains the same sum over the $i \in M_{\hat{\mathbf{y}}^t}$ as in the first term, but in addition it contains the dimension $Y$. We next move the other terms into the first term, leaving with a weight of $\frac{\alpha'_t}{1-\alpha_t}$ for the first term:

$$
\begin{aligned}
\mathcal{L}_{\mathbf{y}} =& \mathbb{E}_{t\sim(0,1]}\mathbb{E}_{q_{\boldsymbol{\theta}}(\mathbf{c}^t|\mathbf{x})}\left[\sum_{\hat{\mathbf{y}}^t}q(\hat{\mathbf{y}}^t\mid\mathbf{y}^0)\left(\frac{\alpha'_t}{1-\alpha_t}\sum_{i\in M_{\hat{\mathbf{y}}^t}}\log\sum_{\tilde{\mathbf{c}}^0}p_{\boldsymbol{\theta}}(\tilde{\mathbf{c}}^0\mid\mathbf{c}^t,\mathbf{x})\mathbb{1}[\varphi_{\hat{\mathbf{y}}^t}(\tilde{\mathbf{c}}^0)_i = y_i^0]\right.\right.\\
&+ \left.\left.\alpha'_t\log\sum_{\tilde{\mathbf{c}}^0}p_{\boldsymbol{\theta}}(\tilde{\mathbf{c}}^0\mid\mathbf{c}^t,\mathbf{x})\mathbb{1}[\varphi_{\hat{\mathbf{y}}^t}(\tilde{\mathbf{c}}^0)_Y = y_Y^0]\right)\right]
\end{aligned}
$$

Next, we apply the inductive hypothesis to the first term. After, note that the second term is independent of the value of $\hat{\mathbf{y}}^t$ as the result of $\varphi_{\hat{\mathbf{y}}^t}(\tilde{\mathbf{c}}^0)_Y$ does not depend on $\hat{\mathbf{y}}^t$.

$$
\begin{aligned}
\mathcal{L}_{\mathbf{y}} =& \mathbb{E}_{t\sim(0,1]}\mathbb{E}_{q_{\boldsymbol{\theta}}(\mathbf{c}^t|\mathbf{x})}\left[\alpha'_t\sum_{i=1}^{Y-1}\log\sum_{\tilde{\mathbf{c}}^0}p_{\boldsymbol{\theta}}(\tilde{\mathbf{c}}^0\mid\mathbf{c}^t,\mathbf{x})\mathbb{1}[\varphi_{\hat{\mathbf{y}}^t}(\tilde{\mathbf{c}}^0)_i = y_i^0]\right.\\
&+ \left.\sum_{\hat{\mathbf{y}}^t}q(\hat{\mathbf{y}}^t\mid\mathbf{y}^0)\alpha'_t\log\sum_{\tilde{\mathbf{c}}^0}p_{\boldsymbol{\theta}}(\tilde{\mathbf{c}}^0\mid\mathbf{c}^t,\mathbf{x})\mathbb{1}[\varphi_{\hat{\mathbf{y}}^t}(\tilde{\mathbf{c}}^0)_Y = y_Y^0]\right]\\
=& \mathbb{E}_{t\sim(0,1]}\mathbb{E}_{q_{\boldsymbol{\theta}}(\mathbf{c}^t|\mathbf{x})}\left[\alpha'_t\sum_{i=1}^{Y}\log\sum_{\tilde{\mathbf{c}}^0}p_{\boldsymbol{\theta}}(\tilde{\mathbf{c}}^0\mid\mathbf{c}^t,\mathbf{x})\mathbb{1}[\varphi(\tilde{\mathbf{c}}^0)_i = y_i^0]\right],
\end{aligned}
$$

completing the inductive proof.

Finally, replacing Eq. 56 for $\mathcal{L}_{\mathbf{y}}$ in Eq. 55 completes the proof. $\qquad\square$

# E   Gradient estimation details

In this section, we provide additional details and formalisation on our gradient estimation procedure, extending the discussion in Section 3.4.

Given some input-output pair $(\mathbf{x}, \mathbf{y}^0) \sim \mathcal{D}$, the gradient of the loss is given by [70, 82]

$$\nabla_{\boldsymbol{\theta}} \mathcal{L}^{\text{NESYDM}} = \underbrace{\nabla_{\boldsymbol{\theta}} \text{H}[q_{\boldsymbol{\theta}}(\mathbf{c}^0 \mid \mathbf{x}, \mathbf{y}^0)]}_{\text{Gradient of \textbf{variational entropy} } \nabla_{\boldsymbol{\theta}} \mathcal{L}_{\text{H}[q]}} +$$

$$\mathbb{E}_{t \sim [0,1], q_{\boldsymbol{\theta}}(\mathbf{c}^0, \mathbf{c}^t \mid \mathbf{x}, \mathbf{y}^0)} \left[ \underbrace{\frac{\alpha_t'}{1 - \alpha_t} \sum_{i \in M_{\mathbf{c}^t}} \nabla_{\boldsymbol{\theta}} \log p_{\boldsymbol{\theta}}(\tilde{c}_i^0 = c_i^0 \mid \mathbf{c}^t, \mathbf{x})}_{\text{Gradient of \textbf{concept unmasking loss} } \nabla_{\boldsymbol{\theta}} \mathcal{L}_{\mathbf{c}}} + \right.$$

$$\underbrace{\alpha_t' \sum_{i=1}^{Y} \log \sum_{\tilde{\mathbf{c}}^0} \nabla_{\boldsymbol{\theta}} p_{\boldsymbol{\theta}}(\tilde{\mathbf{c}}^0 \mid \mathbf{c}^t, \mathbf{x}) \mathbb{1}[\varphi(\tilde{\mathbf{c}}^0)_i = y_i^0]}_{\text{Gradient of \textbf{output unmasking loss} } \nabla_{\boldsymbol{\theta}} \mathcal{L}_{\mathbf{y}}} +$$

$$\left. \underbrace{\left( \frac{\alpha_t'}{1 - \alpha_t} \sum_{i \in M_{\mathbf{c}^t}} \log p_{\boldsymbol{\theta}}(\tilde{c}_i^0 = c_i^0 \mid \mathbf{c}^t, \mathbf{x}) + \alpha_t' \sum_{i=1}^{Y} \log \frac{\sum_{j=1}^{S} \mathbb{1}[\varphi(\tilde{\mathbf{c}}_j^0)_i = y_i^0]}{S} \right) \nabla_{\boldsymbol{\theta}} \log q_{\boldsymbol{\theta}}(\mathbf{c}^0 \mid \mathbf{x}, \mathbf{y}^0)}_{\text{Indirect gradient from sampling from variational distribution (ignored)}} \right]$$

**Monte carlo approximation.** We will use a monte carlo approximation to estimate this gradient. We first sample a single $\mathbf{c}^0 \sim q_{\boldsymbol{\theta}}(\mathbf{c}^0 \mid \mathbf{x}, \mathbf{y}^0)$, $t \sim [0, 1]$, and then a single $\mathbf{c}^t \sim q(\mathbf{c}^t \mid \mathbf{c}^0)$ using Eq. 9. Finally, we sample $S$ samples $\tilde{\mathbf{c}}_1^0, \ldots, \tilde{\mathbf{c}}_S^0 \sim p_{\boldsymbol{\theta}}(\tilde{\mathbf{c}}^0 \mid \mathbf{c}^t, \mathbf{x})$ to approximate the gradient of the **output unmasking loss** $\mathcal{L}_{\mathbf{y}}$ with the RLOO estimator [38]. Alternatively, one could use probabilistic circuits to compute this gradient exactly [50, 86].

**Indirect gradient.** The indirect gradient arises from the expectation over the variational distribution which depends on the parameter $\boldsymbol{\theta}$. This term has high variance in a monte-carlo estimator. Firstly, the vanilla score function estimator is known to have high variance, especially without additional variance reduction techniques [59]. However, the reward, which is given between the large braces, is *doubly-stochastic*: it depends on sampling $t$, $\mathbf{c}^t$, and $\tilde{\mathbf{c}}^0, \ldots, \tilde{\mathbf{c}}^S$, making it an inherently noisy process. Furthermore, when using the variational distribution as defined in Section 3.3, the score term $\nabla_{\boldsymbol{\theta}} \log q_{\boldsymbol{\theta}}(\mathbf{c}^0 \mid \mathbf{x}, \mathbf{y}^0)$ is itself a NESYDM for which computing log-likelihoods is intractable, and thus we would require additional approximations to estimate it. Because of the variance, intractability, and to keep the algorithm simple, we ignore the term altogether.

Therefore, our gradient estimate $\mathbf{g}$ is given by

$$\mathbf{g} = \underbrace{\frac{\gamma_{\mathbf{c}}}{C} \frac{\alpha_t'}{1 - \alpha_t} \sum_{i \in M_{\mathbf{c}^t}} \nabla_{\boldsymbol{\theta}} \log p_{\boldsymbol{\theta}}(\tilde{c}_i^0 = c_i^0 \mid \mathbf{c}^t, \mathbf{x})}_{\text{Unbiased estimate of } \nabla_{\boldsymbol{\theta}} \mathcal{L}_{\mathbf{c}}} + \frac{\gamma_{\text{H}}}{C} \underbrace{\nabla_{\boldsymbol{\theta}} \text{H}[q_{\boldsymbol{\theta}}(\mathbf{c}^0 \mid \mathbf{x}, \mathbf{y}^0)]}_{\text{Choose estimate of } \nabla_{\boldsymbol{\theta}} \mathcal{L}_{\text{H}[q]}} +$$

$$\underbrace{\frac{\gamma_{\mathbf{y}}}{Y} \sum_{i=1}^{Y} \frac{1}{(S-1)\mu_i} \sum_{j=1}^{S} \left( \mathbb{1}[\varphi(\tilde{\mathbf{c}}_j^0)_i = y_i^0] - \mu_i \right) \nabla_{\boldsymbol{\theta}} \log p_{\boldsymbol{\theta}}(\tilde{\mathbf{c}}_j^0 \mid \mathbf{c}^t, \mathbf{x})}_{\text{Consistent estimate of } \nabla_{\boldsymbol{\theta}} \mathcal{L}_{\mathbf{y}}},$$

(57)

where $\mu_i = \frac{1}{S} \sum_{j=1}^{S} \mathbb{1}[\varphi(\tilde{\mathbf{c}}_j^0)_i = y_i^0]$ is the empirical mean of the constraints, and $\gamma_{\mathbf{c}}$, $\gamma_{\text{H}}$ and $\gamma_{\mathbf{y}}$ are weighting coefficients. We keep $\gamma_{\mathbf{y}} = 1$, and tune the other two. Additionally, inspired by the local step approach of [32], we average over dimensions rather than summing to stabilise hyperparameter tuning among different problems. This is especially useful in experiments with variable dimension size such as MNISTAdd and Warcraft Path Planning. We discuss how we estimate the gradient of the entropy of the variational distribution in Section 3.4.

**Estimate of $\nabla_{\boldsymbol{\theta}} \mathcal{L}_{\mathbf{y}}$** Next, we derive the consistent gradient estimator for $\nabla_{\boldsymbol{\theta}} \mathcal{L}_{\mathbf{y}}$ using the RLOO estimator [38]. Assuming we have some $\mathbf{x}$, $\mathbf{y}^0$, $t$ and $\mathbf{c}^t$, and using the score-function estimator, the

gradient of the loss is given by

$$
\begin{aligned}
\nabla_{\boldsymbol{\theta}} \mathcal{L}_{\mathbf{y}} =& \alpha'_t \sum_{i=1}^{Y} \nabla_{\boldsymbol{\theta}} \log \sum_{\tilde{\mathbf{c}}^0} p_{\boldsymbol{\theta}}(\tilde{\mathbf{c}}^0 \mid \mathbf{c}^t, \mathbf{x}) \mathbb{1}[\varphi(\tilde{\mathbf{c}}^0)_i = y_i^0] \\
=& \alpha'_t \sum_{i=1}^{Y} \frac{\sum_{\tilde{\mathbf{c}}^0} \nabla_{\boldsymbol{\theta}} p_{\boldsymbol{\theta}}(\tilde{\mathbf{c}}^0 \mid \mathbf{c}^t, \mathbf{x}) \mathbb{1}[\varphi(\tilde{\mathbf{c}}^0)_i = y_i^0]}{\sum_{\tilde{\mathbf{c}}^0} p_{\boldsymbol{\theta}}(\tilde{\mathbf{c}}^0 \mid \mathbf{c}^t, \mathbf{x}) \mathbb{1}[\varphi(\tilde{\mathbf{c}}^0)_i = y_i^0]} \\
=& \alpha'_t \sum_{i=1}^{Y} \frac{\sum_{\tilde{\mathbf{c}}^0} p_{\boldsymbol{\theta}}(\tilde{\mathbf{c}}^0 \mid \mathbf{c}^t, \mathbf{x}) \mathbb{1}[\varphi(\tilde{\mathbf{c}}^0)_i = y_i^0] \nabla_{\boldsymbol{\theta}} \log p_{\boldsymbol{\theta}}(\tilde{\mathbf{c}}^0 \mid \mathbf{c}^t, \mathbf{x})}{\mathbb{E}_{p_{\boldsymbol{\theta}}(\tilde{\mathbf{c}}^0 \mid \mathbf{c}^t, \mathbf{x})}[\mathbb{1}[\varphi(\tilde{\mathbf{c}}^0)_i = y_i^0]]} \\
=& \alpha'_t \sum_{i=1}^{Y} \frac{\mathbb{E}_{p_{\boldsymbol{\theta}}(\tilde{\mathbf{c}}^0 \mid \mathbf{c}^t, \mathbf{x})}[\mathbb{1}[\varphi(\tilde{\mathbf{c}}^0)_i = y_i^0] \nabla_{\boldsymbol{\theta}} \log p_{\boldsymbol{\theta}}(\tilde{\mathbf{c}}^0 \mid \mathbf{c}^t, \mathbf{x})]}{\mathbb{E}_{p_{\boldsymbol{\theta}}(\tilde{\mathbf{c}}^0 \mid \mathbf{c}^t, \mathbf{x})}[\mathbb{1}[\varphi(\tilde{\mathbf{c}}^0)_i = y_i^0]]}
\end{aligned}
\tag{58}
$$

Both the numerator and denominator are expectations under $p_{\boldsymbol{\theta}}(\tilde{\mathbf{c}}^0 \mid \mathbf{c}^t, \mathbf{x})$ of the constraints. A consistent (but not unbiased) estimator is given by sampling $S$ samples $\tilde{\mathbf{c}}_1^0, \ldots, \tilde{\mathbf{c}}_S^0 \sim p_{\boldsymbol{\theta}}(\tilde{\mathbf{c}}^0 \mid \mathbf{c}^t, \mathbf{x})$ and taking averages at each of these $2Y$ expectations separately. Then, we will use RLOO as a *baseline* to reduce the variance of the numerators. A baseline is a constant $b$, where we use that for any distribution $p_{\boldsymbol{\theta}}(x)$, $\mathbb{E}_{p_{\boldsymbol{\theta}}(x)}[b \nabla_{\boldsymbol{\theta}} \log p_{\boldsymbol{\theta}}(x)] = 0$, and so by linearity of expectation, $\mathbb{E}_{p_{\boldsymbol{\theta}}(x)}[(f(x) - b) \nabla_{\boldsymbol{\theta}} \log p_{\boldsymbol{\theta}}(x)] = \mathbb{E}_{p_{\boldsymbol{\theta}}(x)}[f(x) \nabla_{\boldsymbol{\theta}} \log p_{\boldsymbol{\theta}}(x)]$. Since we are using $S$ samples, we choose for each sample $\tilde{\mathbf{c}}_j^0$ and dimension $i$ the baseline $b_{ij}$ to be the empirical mean *over the other samples*, leaving one sample out: $b_{ij} = \frac{1}{S-1} \sum_{l \neq j} \mathbb{1}[\varphi(\tilde{\mathbf{c}}_l^0)_i = y_i^0]$. Then, $(\mathbb{1}[\varphi(\tilde{\mathbf{c}}_j^0)_i = y_i^0] - b_{ij}) \nabla_{\boldsymbol{\theta}} \log p_{\boldsymbol{\theta}}(\tilde{\mathbf{c}}_j^0 \mid \mathbf{c}^t, \mathbf{x})$ is an unbiased estimator of the numerator. Finally, we average over the $S$ different estimators obtained this way to derive the RLOO gradient estimator as:

$$
\begin{aligned}
&\mathbb{E}_{p_{\boldsymbol{\theta}}(\tilde{\mathbf{c}}^0 \mid \mathbf{c}^t, \mathbf{x})}[\mathbb{1}[\varphi(\tilde{\mathbf{c}}^0)_i = y_i^0] \nabla_{\boldsymbol{\theta}} \log p_{\boldsymbol{\theta}}(\tilde{\mathbf{c}}^0 \mid \mathbf{c}^t, \mathbf{x})] \\
\approx& \frac{1}{S} \sum_{j=1}^{S} (\mathbb{1}[\varphi(\tilde{\mathbf{c}}_j^0)_i = y_i^0] - b_{ij}) \nabla_{\boldsymbol{\theta}} \log p_{\boldsymbol{\theta}}(\tilde{\mathbf{c}}_j^0 \mid \mathbf{c}^t, \mathbf{x}) \\
=& \sum_{j=1}^{S} \left( \frac{(S-1) \mathbb{1}[\varphi(\tilde{\mathbf{c}}_j^0)_i = y_i^0] - \sum_{l \neq j} \mathbb{1}[\varphi(\tilde{\mathbf{c}}_l^0)_i = y_i^0]}{S(S-1)} \right) \nabla_{\boldsymbol{\theta}} \log p_{\boldsymbol{\theta}}(\tilde{\mathbf{c}}_j^0 \mid \mathbf{c}^t, \mathbf{x}) \\
=& \sum_{j=1}^{S} \left( \frac{S \mathbb{1}[\varphi(\tilde{\mathbf{c}}_j^0)_i = y_i^0] - \sum_{l=1}^{S} \mathbb{1}[\varphi(\tilde{\mathbf{c}}_l^0)_i = y_i^0]}{S(S-1)} \right) \nabla_{\boldsymbol{\theta}} \log p_{\boldsymbol{\theta}}(\tilde{\mathbf{c}}_j^0 \mid \mathbf{c}^t, \mathbf{x}) \\
=& \frac{1}{(S-1)} \sum_{j=1}^{S} \left( \mathbb{1}[\varphi(\tilde{\mathbf{c}}_j^0)_i = y_i^0] - \mu_i \right) \nabla_{\boldsymbol{\theta}} \log p_{\boldsymbol{\theta}}(\tilde{\mathbf{c}}_j^0 \mid \mathbf{c}^t, \mathbf{x})
\end{aligned}
\tag{59}
$$

Combining Eqs. 58 and 59 gives the gradient estimator:

$$
\nabla_{\boldsymbol{\theta}} \mathcal{L}_{\mathbf{y}} \approx g_{\mathbf{y}^0}(\tilde{\mathbf{c}}_1^0, \ldots \tilde{\mathbf{c}}_S^0) := \alpha'_t \sum_{i=1}^{Y} \frac{1}{\mu_i (S-1)} \sum_{j=1}^{S} \left( \mathbb{1}[\varphi(\tilde{\mathbf{c}}_j^0)_i = y_i^0] - \mu_i \right) \nabla_{\boldsymbol{\theta}} \log p_{\boldsymbol{\theta}}(\tilde{\mathbf{c}}_j^0 \mid \mathbf{c}^t, \mathbf{x})
\tag{60}
$$

**Full gradient estimation algorithm.** In Algorithm 1, we provide the full algorithm for estimating gradients to train NESYDM. The algorithm proceeds by sampling $\mathbf{c}^0$ from the variational distribution, and then sampling a partially masked value $\mathbf{c}^t$. We then compute the gradients of the three individual losses using Eq. 57. This requires sampling $S$ samples from the unmasking model, which is done in line 5. Finally, we weight the gradients appropriately and sum them up.

# F   Sampling details

We use the first-hitting sampler [94] if the configured number of discretisation steps $T$ is larger or equal to the dimension of the concept space $C$. Otherwise, we use a $T$-step time-discretisation of the reverse process [68].

The first-hitting sampler in Algorithm 3 randomly samples the next timestep to unmask at. There, it randomly selects an index to unmask using the concept unmasking model. Note that $\alpha^{-1}$ is the inverse of the noising schedule. Since we do not provide the temperature to our neural networks, this sampler is, in practice, a concept-by-concept decoding process similar to masked models like BERT [23, 94].

---

**Algorithm 3** First-hitting sampler for $p_{\boldsymbol{\theta}}(\mathbf{c}^0 \mid \mathbf{x})$

1: **Input:** $\mathbf{x}$, unmasking model $p_{\boldsymbol{\theta}}(\tilde{\mathbf{c}}^0 \mid \mathbf{x}, \mathbf{c}^t)$
2: $t \leftarrow 1$
3: $\mathbf{c}^1 = \mathbf{m}$
4: **for** $k \leftarrow C$ **to** 1 **do**
5: $\quad s \leftarrow \alpha^{-1}(1 - \sqrt[k]{u}(1 - \alpha_t))$, where $u \sim \mathcal{U}(0,1)$      ▷ Select next timestep to unmask at
6: $\quad i \sim \text{Uniform}(M_{\mathbf{c}^k})$                     ▷ Select a random dimension to unmask
7: $\quad \mathbf{c}^s \leftarrow \mathbf{c}^t$
8: $\quad c_i^s \sim p_{\boldsymbol{\theta}}(\tilde{c}_i^0 \mid \mathbf{x}, \mathbf{c}^t)$                 ▷ Sample the unmasked dimension
9: $\quad t \leftarrow s$
10: **Return** $\mathbf{c}^0$

---

Instead, the time-discretised sampler in Algorithm 4 samples a completely unmasked sample $\tilde{\mathbf{c}}^0$ from the unmasking model at each timestep, then samples $\mathbf{c}^s$ from the reverse process in Eq. 10 to obtain the next timestep. When sampling from the reverse process, the algorithm remasks some of the newly unmasked dimensions in $\tilde{\mathbf{c}}^0$, while keeping the unmasked dimensions in $\mathbf{c}^t$ fixed.

---

**Algorithm 4** Time discretised sampler for $p(\mathbf{c}^0 \mid \mathbf{x})$

1: **Input:** $\mathbf{x}$, unmasking model $p_{\boldsymbol{\theta}}(\tilde{\mathbf{c}}^0 \mid \mathbf{x}, \mathbf{c}^t)$, number of discretisation steps $T$
2: $\mathbf{c}^1 = \mathbf{m}$
3: **for** $k \leftarrow T$ **to** 1 **do**
4: $\quad \tilde{\mathbf{c}}^0 \sim p_{\boldsymbol{\theta}}(\tilde{\mathbf{c}}^0 \mid \mathbf{x}, \mathbf{c}^t)$               ▷ Sample from unmasking model
5: $\quad \mathbf{c}^s \sim q(\mathbf{c}^s \mid \mathbf{c}^t, \mathbf{c}^0 = \tilde{\mathbf{c}}^0)$          ▷ Sample from reverse process in Eq. 10
6: **Return** $\mathbf{c}^0$

---

### F.1 Sampling from the variational distribution

We adapted the two samplers above to sample from our model conditioned on the output $\mathbf{y}^0$. We use a simple resampling approach as described in Section 3.3, which we elaborate on here. First, we recall the relaxed constraint for $\beta > 0$ as

$$r_\beta(\tilde{\mathbf{c}}^0 \mid \mathbf{y}^0) = \exp(-\beta \sum_{i=1}^{Y} \mathbb{1}[\varphi(\tilde{\mathbf{c}}^0)_i \neq y_i^0]). \tag{61}$$

Then, we define the distribution to sample from as

$$q_{\boldsymbol{\theta}}(\tilde{\mathbf{c}}^0 \mid \mathbf{x}, \mathbf{c}^t, \mathbf{y}^0) \propto p_{\boldsymbol{\theta}}(\tilde{\mathbf{c}}^0 \mid \mathbf{x}, \mathbf{c}^t) r_\beta(\tilde{\mathbf{c}}^0 \mid \mathbf{y}^0). \tag{62}$$

Since we cannot tractably sample from this distribution, we use self-normalised importance sampling [12]. In other words, we sample $K$ samples from the unmasking model, compute the relaxed constraint for each sample, and then normalise these values. Finally, we sample from the renormalised distribution. We provide the full algorithm in Algorithm 5.

We note that the distribution $p_{\boldsymbol{\theta}}(\tilde{\mathbf{c}}^0 \mid \mathbf{x}, \mathbf{c}^t)$ does not appear in the importance weights. This holds because we are sampling from it, thus it divides away in the computation of the importance weights.

We implement this algorithm in the two samplers as follows. For the time-discretised sampler, we replace Algorithm 4 of Algorithm 4 with Algorithm 5. Together, this is the algorithm used in [29]. For the first-hitting sampler, we replace Algorithm 3 of Algorithm 3 by first calling Algorithm 5 to obtain some $\tilde{\mathbf{c}}^0$, and then returning the $i$-th dimension $\tilde{c}_i^0$.

**Relation to Markov Logic Networks.** The distribution in Eq. 62 is similar to a Markov Logic Network (MLN) [67]. Particularly, the formulas of the MLN are (1): the different constraints

---
**Algorithm 5** Self-normalised importance sampling for $q_{\boldsymbol{\theta}}(\tilde{\mathbf{c}}^0 \mid \mathbf{x}, \mathbf{c}^t, \mathbf{y}^0)$
---
1: **Input:** $\mathbf{x}$, $\mathbf{y}^t$, unmasking model $p_{\boldsymbol{\theta}}(\tilde{\mathbf{c}}^0 \mid \mathbf{x}, \mathbf{c}^t)$, number of samples $K$
2: $\tilde{\mathbf{c}}_1^0, \ldots, \tilde{\mathbf{c}}_K^0 \sim p_{\boldsymbol{\theta}}(\tilde{\mathbf{c}}^0 \mid \mathbf{x}, \mathbf{c}^t)$           ▷ Sample $K$ samples from unmasking model
3: $w_i = r_\beta(\tilde{\mathbf{c}}_i^0 \mid \mathbf{y}^0)$, for all $i \in [K]$           ▷ Compute importance weights
4: $Z = \sum_{i=1}^K w_i$           ▷ Normalisation constant
5: $i \sim \text{Categorical}(\frac{w_i}{Z})$           ▷ Sample from renormalised distribution
6: **Return** $\tilde{\mathbf{c}}_i^0$
---

$\mathbb{1}[\varphi(\tilde{\mathbf{c}}^0)_i \neq y_i^0]$, each weighted by $-\beta^2$, and (2): the unmasking model $p_{\boldsymbol{\theta}}(\tilde{\mathbf{c}}^0 \mid \mathbf{x}, \mathbf{c}^t)$, defined with formulas $\mathbb{1}[\tilde{c}_i^0 = c_i^0]$ and weight $\log p_{\boldsymbol{\theta}}(\tilde{c}_i^0 = c_i^0 \mid \mathbf{x}, \mathbf{c}^t)$. In particular, this means that $\tilde{\mathbf{c}}^0$'s that violate constraints still have positive energy. However, the energy exponentially shrinks by a factor of $\frac{1}{\exp(\beta)}$ for each violated constraint. Since we use rather high values of $\beta > 10$, the resampling step in Algorithm 5 is *extremely* likely to pick the sample that violates the least number of constraints.

**Numerically stable implementation.** In practice, computing Eq. 61 in Algorithm 5 is numerically highly unstable if multiple constraints are violated. Then, the reward is equal to $\frac{1}{\exp(l\beta)}$, where $l$ is the number of violated constraints. PyTorch floats are roughly between $\exp(-104)$ and $\exp(88)$, meaning that for $\beta > 10$, the reward underflows at $l = 8$. First, we note that when sampling from the reweighted distribution in Algorithm 5, probabilities are computed as the relative proportion of the rewards. Therefore, we can simply rescale all rewards by a constant factor to ensure they do not underflow or overflow. Particularly, we redefine the reward as

$$r_{\beta, L, U}(\tilde{\mathbf{c}}^0 \mid \mathbf{y}^0) = \exp\left(-\max\left(\beta \sum_{i=1}^Y \mathbb{1}[\varphi(\tilde{\mathbf{c}}^0)_i \neq y_i^0] - L, U\right)\right).$$

Here, $L > 0$ acts as a scaling on the reward as $r_{\beta, L, U}(\tilde{\mathbf{c}}^0 \mid \mathbf{y}^0) = r_\beta(\tilde{\mathbf{c}}^0 \mid \mathbf{y}^0)\exp(L)$ if the max is not active. $U > 0$ acts as a floor on the reward such that samples that violate many constraints still have non-zero probability, even if it is extraordinarily unlikely. We fix $U = 100$ to remain in the floating point range. Note that this is ever so slightly biased as it will over-estimate the probability of the samples that violate the least number of constraints. However, this bias is very small as the probability of choosing these samples is extraordinarily low.

We want to choose $L$ to maximise the range of the reward among the samples without overflowing. Within this range, the differences between the samples that violate the least number of constraints is most important: these are the samples we are most likely to choose. Our intuition is as follows: we set $L$ to the average number of violated constraints among the samples in Algorithm 5. However, if this would overflow the best sample, we instead set $L$ such that the best sample has a reward of $\exp(M)$, where $M = 70$ to prevent overflow. Therefore, we choose

$$L = \min\left(\frac{\beta_t}{S} \sum_{k=1}^S \sum_{i=1}^Y \mathbb{1}[\varphi(\tilde{\mathbf{c}}_k^0)_i \neq y_i^0], M + \min_{k=1}^S \beta_t \sum_{i=1}^Y \mathbb{1}[\varphi(\tilde{\mathbf{c}}_k^0)_i \neq y_i^0]\right). \tag{63}$$

# G    Experimental details

NESYDM is implemented in PyTorch. We used RAdam [45] for all experiments except for MNIST Addition, where we used Adam [35]. We did not compare these optimisers in detail, but we do not expect this choice to significantly affect the results. Furthermore, we used default momentum parameters for both optimisers. For all neural networks used to implement the unmasking model $p_{\boldsymbol{\theta}}(\tilde{\mathbf{c}}^0 \mid \mathbf{x}, \mathbf{c}^t)$, we did *not* pass the current time step $t$ to the network, as previous work found minimal impact on performance for doing so (Appendix E.5 of [68]).

For all experiments, we used GPU computing nodes, each with a single lower-end GPU. In particular, we used NVIDIA GeForce GTX 1080 Ti and GTX 2080 Ti GPUs. All our experiments were run with 12 CPU cores, although this was not the bottleneck in most experiments. On the GTX 1080 Ti, our experiments took between 1 and 17 hours, depending on the complexity of the task and number

---

[2]Unlike MLNs, we sum *unsatisfied* constraints rather than satisfied ones to ensure $r_\beta(\tilde{\mathbf{c}}^0 \mid \mathbf{y}^0) \in [0, 1]$.

of epochs. The project required extra compute when testing different variations of the model, and by performing hyperparameter tuning. For repeating our runs and hyperparameter tuning, we further expect around 600 total GPU hours are needed.

## G.1   Hyperparameter tuning

We list all hyperparameters in Table 4. We perform random search over the hyperparameters on the validation set of the benchmark tasks. For the random search, we used fixed ranges for each parameter, from which we sample log-uniformly. For the parameter $\beta$ we sampled uniformly instead. We used a budget of 30 random samples for each problem, although for some problems we needed more when we found the ranges chosen were poor.

Several hyperparameters, namely the minibatch size, $S$, $K$, $T$ and $L$, are compute dependent, and we keep these fixed when tuning depending on the compute budget and the problem size. The hyperparameters we do tune are the learning rate, $\gamma_{\mathbf{c}}$, $\gamma_{\mathrm{H}}$, and $\beta$. We found that low values of $\gamma_{\mathbf{c}}$ were usually fine, and that for large enough $\beta$ above 10, its value did not matter much. Therefore, the most important hyperparameters to tune are $\gamma_{\mathrm{H}}$ and the learning rate, for which the optimal values varied between problems significantly. For an ablation study on the influence of the value of the loss weighting hyperparameters, see Section H.2.

Table 4: All hyperparameters used in the experiments, and rough recommendations for some of their values. We recommend at least tuning learning rate, and $\gamma_{\mathrm{H}}$ and $\gamma_{\mathbf{c}}$ to some extent (leaving $\gamma_{\mathbf{y}}$ at 1). Some hyperparameters are compute dependent, and higher is always better for reducing gradient estimation variance $(S, K, T)$ and majority voting quality $(L, T)$.

| Variable | Recommendation | Description | Range | Definition |
|---|---|---|---|---|
| — | $(0.0001, 0.0005)$ | Overall learning rate | $\mathbb{R}_{>0}$ | — |
| — | — | Minibatch size | $\mathbb{N}$ | — |
| — | — | Epochs | $\mathbb{N}$ | — |
| $\gamma_{\mathbf{y}}$ | $1$ | Weight of concept unmasking loss | $\mathbb{R}_{\geq 0}$ | Eq. 57 |
| $\gamma_{\mathbf{c}}$ | $10^{-5}$ | Weight of output unmasking loss | $\mathbb{R}_{\geq 0}$ | Eq. 57 |
| $\gamma_{\mathrm{H}}$ | $(0.002, 2)$ | Weight of variational entropy | $\mathbb{R}_{\geq 0}$ | Eq. 57 |
| $\beta$ | $10$ | Penalty in soft constraint | $\mathbb{R}_{>0}$ | Section 3.3 |
| $S$ | $\geq 4$ | Number of RLOO samples | $\mathbb{N}$ | Eq. 7 |
| $K$ | $\geq 2$ | Number of SNIS samples for $q_{\boldsymbol{\theta}}$ | $\mathbb{N}$ | Section F.1 |
| $T$ | $\geq \sqrt{C}$ | MDM discretisation steps | $\mathbb{N}$ | Section 3.5 |
| $L$ | $\geq 8$ | Number of majority voting samples | $\mathbb{N}$ | Section 3.5 |

## G.2   MNIST Addition

We use the LeNet architecture [41] for the neural network architecture as is standard in the NeSy literature [52]. As there are no dependencies between the digits in the data generation process, making the neural network conditional on partially unmasked outputs is not useful: merely predicting marginals is sufficient. Therefore, we ignore the conditioning on $\mathbf{c}^t$ when computing $p_{\boldsymbol{\theta}}(\tilde{\mathbf{c}}^0 \mid \mathbf{c}^t, \mathbf{x})$ in Eq. 47.

Since there is no standard dataset for multidigit MNIST addition, we use a generator defined as follows: for some dataset of MNIST images, we permute it randomly, then split it into $2N$ parts and stack them to obtain the different datapoints. This ensures we use each datapoint in the dataset exactly once, ending up in $\lfloor \frac{60000}{2N} \rfloor$ training datapoints.

We tuned hyperparameters in accordance with Section G.1. Since MNIST has no separate validation dataset, we split the training dataset in a training dataset of 50.000 samples and a validation dataset 10.000 samples before creating the addition dataset. We tune with this split, then again train 10 times with the optimised parameters on the full training dataset of 60.000 samples for the reported test accuracy. We tune on $N = 15$, and reuse the same hyperparameters for $N = 2$ and $N = 4$. For the number of epochs, we use 100 for $N = 2$ and $N = 4$ as an epoch is more expensive for smaller $N$ and because $N = 15$ requires moving beyond a cold-start phase. We found all 10 runs moved past this phase within 100 epochs, but needed more time to converge after.

Table 5: Hyperparameters for MNIST Addition and Warcraft Path Planning.

| Variable | MNIST Addition | Path Planning |
|---|---:|---:|
| learning rate | 0.0003 | 0.0005 |
| minibatch size | 16 | 50 |
| epochs | $N = 4$: 100, $N = 15$: 1000 | 40 |
| $\gamma_{\mathbf{c}}$ | $2 \cdot 10^{-5}$ | $10^{-5}$ |
| $\gamma_{\mathrm{H}}$ | 0.01 | 0.002 |
| $\gamma_{\mathbf{y}}$ | 1 | 1 |
| $\beta$ | 20 | 12 |
| $S$ | 1024 | $12 \times 12$: 16, $30 \times 30$: 4 |
| $K$ | 1024 | $12 \times 12$: 4, $30 \times 30$: 2 |
| $T$ | 8 | 20 |
| $L$ | 8 | 8 |

**Baselines.** For all methods, we take the numbers reported in the papers where possible. We obtained numbers for Scallop from the PLIA paper. For A-NeSI, we pick the best-scoring variant as reported, which is Predict for $N = 4$ and Explain for $N = 15$. For DeepSoftLog, A-NeSI and PLIA, we obtained performance on 10 individual runs from the authors to compute the Mann-Whitney U test.

### G.3 Visual Path Planning

Following [80], we use categorical costs for the Visual Path Planning task. We use $V_{\mathbf{c}} = 5$, which corresponds to the possible cost values in the data, $\mathsf{costs} = [0.8, 1.2, 5.3, 7.7, 9.2]$. Then, $\mathbf{c}^0$ corresponds to an index of the cost of each grid cell. That is, $c^0_{Ni+j} \in \{1, ..., 5\}$ corresponds to the cost value $\mathsf{costs}[c^0_{Ni+j}]$ at grid cell $i, j$.

We adapted the ResNet18-based architecture from [62] for the unmasking model $p(\tilde{\mathbf{c}}^0 \mid \mathbf{x}, \mathbf{c}^t)$ over grid costs. This architecture consists of a single convolutional layer to start encoding the image, with batch normalisation and adaptive max-pooling to a grid of size $N \times N$. After this, we have 64-dimensional embeddings for each grid cell. To condition on the currently unmasked values, we add embeddings of $\mathbf{c}^t \in \{1, \ldots, 5, \mathrm{m}\}^{N^2}$ for each cell: we use six 64-dimensional embeddings $\mathbf{e}^C_1, \ldots, \mathbf{e}^C_5, \mathbf{e}^C_{\mathrm{m}}$ for the different costs plus the mask value. Then we add these embeddings to the image embeddings cell-wise. That is, if $\mathbf{e}^I_{i,j}$ is the image embedding at cell $i, j$, then the new embedding is $\mathbf{e}^I_{i,j} + \mathbf{e}^C_{c^t_{Ni+j}}$. After this, a ResNet layer containing two more convolutional layers follows. Finally, we use an output layer that takes the grid cell embeddings and predicts a distribution over the 5 possible costs.

We performed hyperparameter tuning on the validation set of the $12 \times 12$ grid size problem, then reused the same hyperparameters for the $30 \times 30$ grid size problem. We only reduced the number of RLOO samples $S$ and the number of samples for the SNIS algorithm in Section F.1 for the $30 \times 30$ grid size problem to reduce the overhead of many calls to Dijkstra's algorithm. This algorithm quickly becomes the main compute bottleneck on large grids.

For $12 \times 12$, we evaluated test accuracy at 40 epochs, and for $30 \times 30$ we evaluated validation accuracy every 5 epochs within the 40 epoch timeframe, choosing the best performing model for the test accuracy. We found that on $30 \times 30$ the model was sometimes unstable, suddenly dropping in accuracy and recovering after a while. As is common in this task and our baselines, we consider a path prediction correct if the predicted path has the same cost as the gold-truth shortest path given. This is because shortest paths may not be unique.

**Baselines.** We take the numbers for EXAL as reported in the paper [85]. For A-NeSI, I-MLE and A-NeSI + RLOO, we obtained performance on 10 individual runs from the authors to compute the Mann-Whitney U test.

## G.4 RSBench

For all experiments, we adapt the implementation of the benchmark in the RSBench repository [11]. We use the conditional 1-step entropy discussed in Section 3.4. For the MNIST experiments, we brute-force the conditional entropy computation, while for BDD-OIA, we adapt the inference procedure in [11] to obtain the conditional entropy.

### G.4.1 Metrics

For all tasks, we compute the Expected Calibration Error (ECE) over *marginal* concept probabilities [60] as a metric for calibration. Since NESYDM is not tractable, we have to estimate these marginals. Therefore, we use simple maximum-likelihood estimation to obtain approximate marginal probabilities for $p_{\boldsymbol{\theta}}(w_i \mid \mathbf{x})$ by sampling $L$ samples from the model and taking the empirical mean. We used $L = 1000$ throughout to improve the accuracy of the ECE estimate.

For the MNIST tasks, we report both the output accuracy $Acc_{\mathbf{y}}$ and concept accuracy $Acc_{\mathbf{c}}$. In particular, for output accuracy, we compute exact match accuracy over the output predictions. For concept accuracy, we use *micro-averaged* accuracy over the concept predictions (that is, the two digits). This requires NESYDM to output predictions for the digits separately. We tried two different majority voting strategies using the $L$ samples. 1) Take the dimension-wise mode among the samples, or 2) take the most common complete concept vector $\mathbf{c}$ and use the individual dimensions of $\mathbf{c}$ as the predictions. We used the second strategy in Table 3 to ensure the predictions can capture dependencies between digits, and compare the two methods in Section H.1 and Table 8.

For BDD-OIA, we report macro-averaged F1 scores for both the output and concept prediction. For example, for the concept F1 score, we compute the F1 score for each concept separately, then take the unweighted mean of these F1 scores. Similarly, we computed macro-averaged ECE scores for concept prediction. For NESYDM, we computed marginal probabilities for concept and output predictions, that is, per dimension. Furthermore, we recomputed all metrics for the baselines reported in Table 3, as we found bugs in the code for both metrics in the RSBench and BEARS codebases. Note that PNP$^{\perp\!\!\!\perp}$ was called DPL in the BEARS paper. We changed the name as 1) the baseline code did not actually use the DeepProbLog language, and 2) there are many different NeSy predictors that could be implemented in DeepProbLog, so it is not clear which one to compare to.

### G.4.2 Why Expected Calibration Error for reasoning shortcut awareness?

In this section, we motivate the use of concept calibration, in particular using the Expected Calibration Error (ECE), to empirically measure reasoning shortcut (RS) awareness. BEARS [55] introduced RS-awareness as attaining high accuracy on concepts unaffected by RSs, while being calibrated on concepts affected by RSs. In the latter case, perfect concept accuracy is unattainable, and we should aim for high calibration. A model that predicts concepts in such a way by mixing over RSs that cannot be disambiguated from data alone maximises data likelihood [39].

For example, consider the XOR problem from Example 2.2, which has 1 RS that maps MNIST digits of 1s to 0s and MNIST digits of 0s to 1s. This RS cannot be distinguished from the ground-truth mapping. The ideal model under this ambiguity would assign 50% confidence to the ground-truth mapping and 50% to the RS. We can achieve this with a neural network that given two distinct MNIST digits outputs a uniform distribution over (0, 1) and (1, 0), and given two equivalent MNIST digits, outputs a uniform distribution over (0, 0) and (1, 1). In the first case, the XOR function will return 1 with probability 1, and in the second case, it will return 0 with probability 1.

This attains maximum data likelihood as the model returns the correct output label. Furthermore, it always assigns 0.5 probability to 1 and 0. Its concept accuracy will also be around 0.5 in our synthetic setup. Therefore, the ECE will also be around 0, highlighting its calibration, and in practical experiments low ECE values are attainable [39]. Instead, a non RS-aware method finding the RS will attain an ECE of around 1: it always predicts exactly the opposite of the correct concept. Since the non RS-aware method randomly finds the RS or the correct solution, the ECE will be around 0.5 on average [39]. For concepts that are not affected by RSs, the accuracy will be 1 and maximum likelihood will also attain high confidence, resulting also in a low ECE value.

That said, ECE is a proxy for measuring concept calibration, but is not necessarily the only or perfect metric for uncertainty quantification. A further theoretical study evaluating how to best measure RS-awareness could be of significant value.

### G.4.3 Hyperparameters

Table 6: Hyperparameters for RSBench.

| Variable | MNIST Half & Even-Odd | BDD-OIA |
|---|---:|---:|
| learning rate | 0.00009 | 0.0001 |
| minibatch size | 16 | 256 |
| epochs | 500 | 30 |
| $\gamma_{\mathbf{c}}$ | $1.5 \cdot 10^{-6}$ | $5 \cdot 10^{-6}$ |
| $\gamma_{\mathrm{H}}$ | 1.6 | 2.0 |
| $\gamma_{\mathbf{y}}$ | 1 | 1 |
| $\beta$ | 10 | 10 |
| $S$ | 1024 | 1024 |
| $K$ | 1024 | 1024 |
| $T$ | 8 | 22 |
| $L$ | 1000 | 1000 |

For the datasets in RSBench, we tuned on the validation set for all parameters using the conditional entropy. Then, we ran an additional hyperparameter search on just the entropy weight to find the right trade-off between calibration and accuracy. We found the entropy weight can be sensitive, where high values significantly slow down training, while low values may result in uncalibrated models. See Table 10 and Section H.2 for an ablation study on the effect of the entropy weight. For $L$, we use a much higher number of 1000 samples. This is to ensure the Expected Calibration Error is properly estimated (see Section G.4.1 for details). For the runs with the unconditional entropy, we used the same hyperparameters and no additional hyperparameter search.

### G.4.4 Experimental details and architectures

**MNIST Half and MNIST Even-Odd.** We adapted the original architecture from our baselines in [55]. For both experiments, we encode the two individual digits in $\mathbf{x}$ with a convolutional neural network (ReLU activations) of 3 layers, with 32, 64 and 128 channels respectively. Then, we flatten the output, obtaining two embeddings $\mathbf{e}_1$ and $\mathbf{e}_2$. For predicting the unmasking distribution $p(\tilde{c}_1^0 \mid \mathbf{x}, \mathbf{c}^t)$ for the first digit, we concatenate one-hot encodings of $c_1^t$, $c_2^t$ and $\mathbf{e}_1$, while for predicting the distribution of the second digit $p(\tilde{c}_2^0 \mid \mathbf{x}, \mathbf{c}^t)$, we concatenate one-hot encodings of $c_2^t$, $c_1^t$ and $\mathbf{e}_2$. Note the order here: This is to ensure permutation equivariance, as the sum is a commutative operation. Therefore, like our baselines, we have a disentangled architecture that uses the same neural network to classify the two digits, while still incorporating the currently unmasked values. Finally, using the concatenated vector, we use a linear output layer and a softmax to obtain the distribution over the possible digits.

**BDD-OIA.** We used early stopping by running for 30 epochs, testing on the validation set every epoch and picking the model with the highest validation accuracy. As in [55], we used preprocessed embeddings of the dashcam images from a Faster-RCNN [66]. This Faster-RCNN was pre-trained on MS-COCO and fine-tuned on BDD-100k. These are provided in the RSBench dataset, and were also used for BEARS. For the unmasking model $p(\tilde{\mathbf{c}}^0 \mid \mathbf{x}, \mathbf{c}^t)$, we adapted the MLP from [55], using a single hidden layer with a dimensionality of 512, by simply concatenating a one-hot encoding of $\mathbf{c}^t \in \{0, 1, \mathrm{m}\}^{21}$ to the input embedding of $\mathbf{x}$. Note that, since the concepts are binary, this one-hot encoding is a 3-dimensional vector, as it can be 0, 1, or the mask value m.

**Baselines.** We obtained results of the 5 individual runs used for each method in [55] and re-evaluated them to obtain 4 digits of precision for all reported results, as [55] only reported 2 digits of precision. Furthermore, we used these results to compute statistical significance tests. We have different results than reported in [55] for BDD-OIA as we found bugs in the code for the metrics.

# H  Further ablation experiments

## H.1  Other majority voting strategies

As stated in the main text, computing the exact mode $\text{argmax}_{\mathbf{y}^0} \, p_{\boldsymbol{\theta}}^{\text{NESYDM}}(\mathbf{y}^0 \mid \mathbf{x})$ is intractable in general, also for representations supporting tractable marginals [2, 84]. Throughout this paper, we used the majority voting strategy described in Section 3.5. However, when observing the results on MNIST-Even-Odd, one might be puzzled by the relatively high performance on concept accuracy while the output accuracy is low. We hypothesised that this was due to our chosen majority voting strategy, and repeated the evaluation of the models using different strategies, which we describe here. All assume access to a set of samples $\mathbf{c}_1^0, \ldots, \mathbf{c}_L^0 \sim p_{\boldsymbol{\theta}}(\mathbf{c}^0 \mid \mathbf{x}, \mathbf{c}^1 = \mathbf{m})$.

- *Program-then-true-mode (PTM)*: The strategy described in Eq. 8, and the main one used in this paper. We emphasise that this is the "correct" strategy according to the generative process of NeSy predictors.

- *Program-then-marginal-mode (PMM)*: Similar to above, we feed all sampled concepts into the program, but rather than taking the most likely output, we choose the most likely output dimension-wise:

$$\hat{y}_i = \text{argmax}_{y_i} \sum_{l=1}^{L} \mathbb{1}[\varphi(\mathbf{c}_l^0)_i = y_i] \tag{64}$$

- *True-mode-then-program (TMP)*: Find the mode of the sampled concepts, then feed that into the program:

$$\hat{\mathbf{y}} = \varphi \left( \text{argmax}_{\mathbf{c}} \sum_{l=1}^{L} \mathbb{1}[\mathbf{c}_l^0 = \mathbf{c}] \right) \tag{65}$$

- *Marginal-mode-then-program (MMP)*: Compute the dimension-wise mode of the concepts $\hat{c}_i$, combine them into a single concept $\hat{\mathbf{c}}$, and feed that into the program:

$$\hat{c}_i = \text{argmax}_{c_i} \sum_{l=1}^{L} \mathbb{1}[c_{l,i}^0 = c_i], \quad \hat{\mathbf{y}} = \varphi(\hat{\mathbf{c}}) \tag{66}$$

Table 7: Output accuracy, both in- and out-of-distribution, for different majority voting strategies on the MNIST-Half and MNIST-Even-Odd datasets.

| Strategy | Half, ID | Half, OOD | Even-Odd, ID | Even-Odd, OOD |
|---|---|---|---|---|
| **NESYDM, Conditional entropy** | | | | |
| PTM | 99.12± 0.18 | 28.45± 0.90 | **98.65± 0.31** | 0.02± 0.04 |
| PMM | 99.12± 0.18 | 28.44± 0.91 | **98.65± 0.31** | 0.02± 0.04 |
| TMP | 98.87± 0.23 | 28.46± 0.91 | 97.94± 0.49 | 0.18± 0.14 |
| MMP | 60.16± 4.77 | 33.15± 1.40 | 25.14± 2.81 | **5.39± 0.45** |
| **NESYDM, Unconditional entropy** | | | | |
| PTM | 99.12± 0.10 | 10.95± 0.05 | 97.52± 0.44 | 0.00± 0.00 |
| PMM | 99.12± 0.10 | 10.95± 0.05 | 97.52± 0.44 | 0.00± 0.00 |
| TMP | **99.26± 0.26** | 15.71± 0.49 | 98.10± 0.37 | 0.02± 0.02 |
| MMP | 79.42± 3.14 | **44.11± 4.87** | 87.64± 0.37 | **5.27± 0.52** |

We evaluated these strategies on the validation set of all benchmarks, and found that they all performed similar, or at most marginally worse than the PTM strategy used in this paper. However, we found exceptions in MNIST-Half and MNIST-Even-Odd, where MMP significantly outperforms the other strategies in the OOD setting, while significantly *under*performing in the ID setting, as highlighted in Table 7. This result holds for both NESYDM with the conditional entropy and the unconditional entropy.

ID performance of MMP takes a rather significant hit because there are strong statistical dependencies between the concepts in the construction of the ID datasets. Especially the Even-Odd OOD dataset

is rather adversarially constructed, as highlighted by the extremely low OOD performance of all methods. However, because NESYDM has relatively high concept accuracy OOD, using MMP still results in some correct outputs.

We performed a similar analysis for the two strategies for predicting concepts in Table 8. Here we find that, overall, the true mode strategy usually performs better, except that we find a significant difference between TM and MM on the OOD dataset of MNIST-Half.

Table 8: Concept accuracy, both in- and out-of-distribution, for different majority voting strategies on the MNIST-Half and MNIST-Even-Odd datasets.

| Strategy | Half, ID | Half, OOD | Even-Odd, ID | Even-Odd, OOD |
|---|---|---|---|---|
| **NESYDM, Conditional entropy** | | | | |
| TM | 71.16± 1.77 | 61.84± 0.89 | **20.33± 1.33** | **15.60± 0.99** |
| MM | 66.78± 2.84 | 62.76± 0.77 | **19.65± 2.37** | 14.56± 0.86 |
| **NESYDM, Unconditional entropy** | | | | |
| TM | **79.41± 6.58** | 57.22± 0.49 | 0.36± 0.39 | 4.65± 0.49 |
| MM | **80.56± 5.12** | **70.40± 2.71** | 0.39± 0.44 | 1.16± 0.44 |

## H.2 Effect of loss weighting hyperparameters

In this appendix, we investigate the effect of the loss weighting hyperparameters $\gamma_{\mathbf{c}}$, $\gamma_{\mathrm{H}}$ and $\gamma_{\mathbf{y}}$ on the performance of NESYDM. We experimented with the conditional entropy version of NESYDM on the MNIST-Half dataset with three repeated runs. Here, we kept the output unmasking weight $\gamma_{\mathbf{y}} = 1$ and tuned the other two hyperparameters.

For the concept unmasking weight $\gamma_{\mathbf{c}}$ in Table 9, we find that all tested values achieve high label accuracy. However, its value significantly influences the concept accuracy both in and out of distribution, and the OOD label accuracy. We observe values below 1 are effective. We suspect the concept unmasking loss can provide the model information that it is useful, but it should not dominate the loss, as this results in significantly lower concept accuracy and OOD performance. Furthermore, it results in poor calibration, suggesting that the model converged onto a single reasoning shortcut.

We observe a significant influence of the entropy weight $\gamma_{\mathrm{H}}$ in Table 10. All values below 1.3 seem to converge on a single reasoning shortcut, exhibiting poor calibration and worse concept accuracy. These runs do not balance the maximisation of entropy as the weight is too low, resulting in models with low entropy. By increasing the entropy weight beyond the value we used (1.6), we find that the entropy loss can also impact the performance when above 2.0, resulting in reduced label and concept accuracy and calibration. As the optimal range of values is quite tight, we recommend tuning the entropy weight when using NESYDM.

Table 9: Effect of concept unmasking weight on label and concept accuracies (in- and out-of-distribution) and calibration (ECE) performance on MNIST-Half. Bold values indicate best results per column. We used 1e-06 in the experiments.

| $\gamma_{\mathbf{c}}$ | $\mathrm{Acc}_{\mathbf{y}} \uparrow$ | $\mathrm{Acc}_{\mathbf{c}} \uparrow$ | $\mathrm{Acc}_{\mathbf{y},\mathrm{OOD}} \uparrow$ | $\mathrm{Acc}_{\mathbf{c},\mathrm{OOD}} \uparrow$ | $\mathrm{ECE}_{\mathbf{c},\mathrm{ID}} \downarrow$ | $\mathrm{ECE}_{\mathbf{c},\mathrm{OOD}} \downarrow$ |
|---|---|---|---|---|---|---|
| 1e-08 | 99.38± 0.27 | 70.45± 2.32 | 33.92± 5.10 | 65.13± 2.72 | 8.54± 4.46 | 12.23± 1.18 |
| 1e-07 | **99.61± 0.13** | 71.41± 0.72 | **37.16± 1.22** | **67.74± 0.56** | 7.77± 1.09 | 10.67± 0.98 |
| **1e-06** | 99.54± 0.80 | **72.88± 0.85** | 33.21± 7.06 | 64.86± 3.92 | 5.31± 1.19 | 11.00± 0.87 |
| 1.5e-06 | 99.12± 0.10 | 71.16± 1.77 | 28.44± 0.90 | 62.76± 0.89 | 4.18± 2.56 | 11.74± 1.18 |
| 1e-05 | 99.00± 0.53 | 69.79± 3.09 | 29.72± 3.00 | 63.10± 2.11 | 6.33± 3.87 | 11.39± 1.76 |
| 0.0001 | 99.23± 0.13 | 72.07± 0.77 | 36.46± 6.60 | 66.84± 4.04 | 4.90± 0.69 | **10.22± 1.96** |
| 0.001 | **99.61± 0.13** | 72.03± 0.84 | 32.75± 6.51 | 64.60± 4.44 | 4.98± 3.15 | 10.46± 2.86 |
| 0.01 | 99.15± 0.48 | 71.60± 0.35 | 31.99± 7.80 | 63.57± 5.41 | **3.80± 1.52** | 10.81± 1.07 |
| 0.1 | 99.46± 0.35 | 71.88± 0.81 | 36.76± 8.27 | 66.85± 5.05 | 6.96± 2.35 | 12.35± 4.01 |
| 1.0 | 98.84± 0.61 | 41.55± 0.12 | 5.70± 0.24 | 38.65± 0.14 | 56.87± 0.25 | 61.09± 0.11 |
| 10.0 | 99.38± 0.13 | 41.59± 0.13 | 5.64± 0.19 | 38.59± 0.19 | 57.00± 0.23 | 60.97± 0.08 |

Table 10: Effect of entropy weight on label and concept accuracies (in- and out-of-distribution) and calibration (ECE) performance on MNIST-Half. Bold values indicate best results per column. We used 1.6 in the experiments.

| $\gamma_{\mathrm{H}}$ | $\mathrm{Acc_y} \uparrow$ | $\mathrm{Acc_c} \uparrow$ | $\mathrm{Acc_{y,OOD}} \uparrow$ | $\mathrm{Acc_{c,OOD}} \uparrow$ | $\mathrm{ECE_{c,ID}} \downarrow$ | $\mathrm{ECE_{c,OOD}} \downarrow$ |
|---|---|---|---|---|---|---|
| 0.001 | 99.15± 0.13 | 41.63± 0.07 | 5.61± 0.16 | 38.63± 0.03 | 57.05± 0.12 | 61.08± 0.16 |
| 0.01 | 99.00± 0.35 | 41.74± 0.18 | 5.79± 0.18 | 38.66± 0.03 | 56.80± 0.21 | 60.91± 0.11 |
| 0.1 | 98.77± 0.48 | 41.67± 0.12 | 5.61± 0.32 | 38.60± 0.23 | 56.98± 0.26 | 60.97± 0.09 |
| 0.5 | 99.31± 0.23 | 41.63± 0.07 | 5.58± 0.14 | 38.59± 0.15 | 56.94± 0.09 | 61.09± 0.01 |
| 1.0 | 99.23± 0.13 | 41.63± 0.13 | 5.85± 0.14 | 38.73± 0.12 | 56.90± 0.25 | 61.06± 0.09 |
| 1.3 | 99.77± 0.23 | 67.05± 1.99 | 35.51± 2.78 | 65.79± 1.79 | 9.33± 1.74 | 12.26± 1.31 |
| **1.6** | 99.12± 0.10 | **71.16± 1.77** | 28.44± 0.90 | 62.76± 0.89 | **4.18± 2.56** | 11.74± 1.18 |
| 2.0 | 99.61± 0.13 | 72.26± 0.41 | 29.35± 1.73 | 62.19± 1.03 | 3.79± 0.48 | 11.42± 0.61 |
| 3.0 | 89.81± 5.84 | 46.53± 2.50 | 17.03± 8.97 | 51.23± 7.03 | 12.70± 5.87 | 14.56± 2.71 |
| 5.0 | 85.73± 1.94 | 39.74± 1.33 | 11.24± 0.14 | 44.04± 1.39 | 12.56± 1.33 | 24.22± 2.12 |
| 10.0 | 85.73± 0.35 | 34.22± 1.27 | 10.81± 0.19 | 41.25± 1.03 | 15.40± 2.25 | 24.01± 1.56 |

