# OpenReview forum: "Neurosymbolic Diffusion Models"
_NeurIPS.cc/2025/Conference — NeurIPS 2025 poster_

### Official Review · Reviewer_QdAB · 2025-07-01

**Clarity:** 2
**Significance:** 1
**Originality:** 2
**Rating:** 5
**Confidence:** 2

**Summary:**

The paper introduces neurosymbolic diffusion models that are a novel class of neurosymbolic models based on diffusion process. The key contributions of these models is that they overcome the "independence assumption" and reuse the "independence assumption" from NeSy predictors at each step of deffusion process for scalable learning. The method shows close to SOTA performance on multiple synthetic datasets.

**Questions:**

I have mentioned my concerns and two main questions in the weakness section.

**Ethical Concerns:**

["NO or VERY MINOR ethics concerns only"]

**Final Justification:**

- The authors clarified my doubts about **independence assumption**.
- **About the metric ECE,** I think the authors did give a reasonable justification for it on a toy example. However, it still does not stand as a clear proof for me. However, given that RS-awareness is only one of the multiple contributions of NesyDM, I lean towards acceptance.
- My confidence remains 2, as although I am reasonably aware of NeSy literature, I am not so well-versed with Diffusion models, and some important aspects of the proposed model may have been missed by me.

**Limitations:**

I think the limitations  are fairly addressed wrt the methodology proposed by the authors

**Quality:**

3

**Strengths And Weaknesses:**

***Strengths:***

Novel use of diffusion models in a NeSy setting

***Weakness:***

**Independence Assumption**

My main problem with the paper is statements like  *"independence assumption haunting NeSy predictors...."* can be misleading in my current understanding of the literature: Weighted Model Counting is fully expressive wrt any discrete distribution, given that one adds additional parameters and this is already addressed in the original paper on WMC [1], this is also true for Problog [2, Section 2.6].  Hence, I find the statement that NeSy predictors can not express something because of the "independence assumption" quite misleading, and infact "independence assumption" should more appropriately be called "multiplicative parameterization". The claim that, *"The only maximisers of Eq. 1 for the independent model are to either deterministically return (0, 1) or (1, 0). However, there is no maximiser that can simultaneously assign probability mass to both cases, meaning independent models cannot be RS-aware"*,  seems wrong to me. I demonstrate it in the following:

If you define $P_{\theta}(c = (c_1,c_2))$ as a WMC distribution, by defining $w(c_i) = p_i$ if $c_i =1$ and $w(c_i) = 1-p_i$ if $c_i =0$, and

$$P_{\theta}(c = (c_1,c_2)) = \frac{w_{1}(c_1)w_{2}(c_2)}{\sum_{c' \models y} w_{1}(c'_1)w_2(c'_2)}$$


where $w(c_i) \in [0,1]$, then for $ w_i = 0.5$ you get uniform distribution on (0,1) and (1,0) and this parameterization maximizes equation 1, infact equation 1 achieves the maximum value of 1, always with this equation. The only NeSy model in which Equation 1 actually reduces to only multiplicative weights with no partition function is semantic loss, and again semantic loss is essentially a regularizer which is trained with other loss terms. So it is not meaningful to analyze its expressivity in isolation.  If required, I can also provide a WCM reduction that is, it can express any possible distribution.

Infact, theoretically WMC is already much more powerful than diffusion, if analyzed wrt Theorem C.1. As Theorem C.1 only guarentees full expressivity as $T \rightarrow \infty$ which in my understanding requires infinitely many parameters to be modeled. Whereas with WMC you can model any distribution on n discrete boolean random variables with $2^{n}-1$ parameters.

Question: Given that WMC and Problog are already fully expressive, with better bounds on the number of required parameters than what is proposed in the paper, then why should one use diffusion?

**RS-Awareness**

This critique could be because of my general misunderstanding of the literature. However, it is not obvious to me that RS-awareness as measured by ECE is indeed a correct metric. Note that parameters in NeSy models are supposed to weight predictions in a way such that they agree with constraints and (as usual) maximize data likelihood. Whereas ECE, in literature, and in the paper is supposed to measure if the model's accuracy on a given concept is indeed same as the predicted confidence.

Question: Can the authors show that these two notions,  RS-Awareness while maximizing data likelihood and constraint satisfaction, and ECE are mutually compatible or even consistent with each other?

**Experiments**

As mentioned earlier, I am not sure of ECE is a meaningful metric in this setting. Now removing ECE, the experimental results for accuracy look quite mixed on the rather toy datasets (common in NeSy literature).


[1] On probabilistic inference by weighted model counting 2008. Mark Chavira and Adnan Darwiche.

[2] Foundations of Probabilistic Logic Programming. Fabrizio Riguzzi [Section 2.6]

---

> ### Author Rebuttal · Authors · 2025-07-31
>
> We thank the reviewer for their valuable comments, and are glad they found our approach novel.
>
> The reviewer wonders about the benefit of Neurosymbolic Diffusion Models (NeSyDMs) compared to increasing the number of parameters in weighted model counting (WMC). We highlight how NeSyDMs use neural networks that predict the same number of parameters as baseline SotA NeSy methods that independence over concepts. Instead, increasing the number of WMC parameters also requires larger neural networks. This allows it to **scale to more challenging reasoning problems**. The reviewer also asked for a motivation for the ECE measure in studying Reasoning Shortcut Awareness. We motivate this measure in this context and provide an example. **We will add both clarifications to our manuscript**, and provide detailed answers below.
>
> > Question: Given that WMC and Problog are already fully expressive, with better bounds on the number of required parameters than what is proposed in the paper, then why should one use diffusion?
>
> **The number of parameters is precisely our motivation**. We assume a fixed number of concepts, say $n$ binary variables. Over these, we assume conditional independence, predict parameters with a neural diffusion model from an input $\mathbf{x}$, and estimate $n$ WMC parameters, as is standard in our baselines. One could alternatively increase the expressivity by introducing additional variables and changing the program. Then one can indeed achieve full expressivity at $2^{n}-1$ parameters. However, in NeSy one needs to predict these additional parameters with the neural network, significantly increasing the neural network size and training requirements. For full expressivity on our largest path planning task, our diffusion method requires a neural network that predicts only $5*900$ rather than $5^{900}-1$ parameters. We explain how to think of these numbers below.
>
> > Question: Can the authors show that these two notions, RS-Awareness while maximizing data likelihood and constraint satisfaction, and ECE are mutually compatible or even consistent with each other?
>
> Certainly! We understand RS-awareness as in BEARS [1]: we aim for high accuracy on concepts unaffected by reasoning shortcuts, while being calibrated on concepts that are affected; therefore, perfect concept accuracy is unattainable. A model which predicts concepts like this while mixing over reasoning shortcuts also maximises data likelihood. To measure calibration over ambiguous concepts, we use ECE. We provide a detailed example below.
>
> # Detailed response
>
> ## Independence assumption
> Thank you for the useful comments. We will clarify our understanding of this problem and highlight the benefit of using diffusion for NeSy.
>
> > Weighted Model Counting is fully expressive wrt any discrete distribution, given that one adds additional parameters
>
> **The reviewer is correct that WMC can be more expressive only by adding variables apart from those used to characterise concepts and changing the constraint.**
> - We understand independence in our paper as follows: given a fixed program and a fixed number of concepts, we define a factorised distribution over them (conditioned on the input $\mathbf{x}$). In the graphical modelling literature, one would therefore say these concepts are conditionally independent.
> - By increasing the number of variables beyond those characterising concepts, we can indeed represent more flexible distributions over concepts than conditionally independent ones. This is an alternative to diffusion for increasing the expressivity beyond independence, which is done in eg SPL [1] (mentioned in paper). SPL cannot be directly applied to our setting where concepts are latent, and all NeSy baselines we compare against do not introduce these additional variables to make WMC more expressive. **As a matter of fact, all baselines we consider (except BEARS) assume concepts to be independent and hence are limited in expressiveness and uncertainty quantification**.
> - We agree that this possibility is not clearly highlighted in the paper; we will clarify this point in the camera-ready version.
>
> Secondly, the reviewer shows a parameterisation that attains the uniform distribution. Note that this parameterisation requires access to the label $y$ to compute the partition function. But in our NeSy setting, the model does not know what the label $y$ is at test time, as that is what the model needs to predict. Therefore, this particular parameterisation is not applicable here.
>
> It is possible to have an unconditioned uniform WMC distribution when using three parameters; however, this would go beyond what we understand in our paper as the independence assumption.
>
> Now, since theoretically WMC can be fully expressive, why use diffusion? As the reviewer mentions, to attain increased expressivity via WMC, we need to increase the number of WMC parameters. However, consider our NeSy setting: these WMC parameters are predicted from the high-dimensional input $\mathbf{x}$ using our neural network. So we would need to train a (potentially very large) neural network that predicts $2^n-1$ WMC parameters, which is certainly infeasible for the problems we study. For the largest path planning task in our experiments, this requires predicting $5^{900}-1$ WMC parameters for full expressivity.
>
> Instead, by having our neural network predict an independent distribution / multiplicative parameterisation, the network only needs to predict $5*900$ WMC parameters, creating a practically viable expressive parameterisation.
>
> > As Theorem C.1 only guarentees full expressivity as  $T\rightarrow \infty$ which in my understanding requires infinitely many parameters to be modeled
>
> Increasing $T$ does **not** increase any requirements of parameter counts. $T\rightarrow \infty$, mentioned in Theorem C.1 refers to the number of diffusion timesteps, not number of parameters. We assume $T\rightarrow \infty$ (continuous-time diffusion) throughout our paper.
>
> ## ECE validity
> > Question: Can the authors show that these two notions, RS-Awareness while maximizing data likelihood and constraint satisfaction, and ECE are mutually compatible or even consistent with each other?
>
> **We give an example of consistency.** Consider again the XOR problem, which has 1 reasoning shortcut (RS) that maps MNIST digits of 1s to 0s and MNIST digits of 0s to 1s. This RS cannot be distinguished from the ground-truth mapping. The ideal model under this ambiguity would assign 50% confidence to the ground-truth mapping and 50% to the RS.
>
> How would one achieve this? Using a neural network that, 1) given two distinct MNIST digits outputs a uniform distribution over (0, 1) and (1, 0), and 2) given two equivalent MNIST digits, outputs a uniform distribution over (0, 0) and (1, 1). Then for case 1), running the XOR function returns 1 with probability 1, and for case 0), it will return 0 with probability 1.
>
> This attains maximum data likelihood: The model will always return the correct output label. Furthermore, it always assigns 0.5 probability to 1 and 0. Its concept accuracy will also be ~0.5 in our synthetic setup. Therefore, the ECE will also be ~0, highlighting its calibration.
> Instead, a non RS-aware method finding the RS will attain an ECE of ~1: it always predicts exactly the opposite of the correct concept.
>
> For concepts that are not affected by RSs, the accuracy will be 1 and maximum likelihood will also attain high confidence, resulting also in a low ECE value.
>
> Now, ECE is a proxy for measuring RS-awareness and also not the only / perfect metric for uncertainty quantification. A future study in the limitations of ECE for measuring RS-awareness could be of significant value, even if it is useful for the problems studied in this paper. We would be happy to add alternative metrics the reviewer suggests.
>
> We will add a paragraph motivating the ECE measure in the appendix with this example included.
>
> 1. Ahmed, K., Teso, S., Chang, K. W., Van den Broeck, G., & Vergari, A. (2022). Semantic probabilistic layers for neuro-symbolic learning. Advances in Neural Information Processing Systems, 35, 29944-29959.

---

> > ### Comment · Reviewer_QdAB · 2025-08-03
> > **Thanks for the detailed review. I will upgrade my score.**
> >
> > **Independence Assumption:** The authors have clarified my misunderstanding on this aspect of WMC. Indeed *the authors are right* and computing partition function without ground truth labels is non-trivial. However, I hope that authors will clarify this nuance, i.e., that they are dealing with a specific (and a very substantial) class of NeSy models.
> >
> > **ECE validity:** *I am still not sure about the fact that ECE and NeSy's notions of parameters/confidence align.* More formally, is it true that an optimal (say fully expressive) NeSy model will always maximize ECE for all constraints? Note that ECE as written in BEARS Appendix A.2 does not take care of constraints.* However, I do agree that given that RS-awareness is not the only objective of the paper at hand, the authors response is enough for me to raise my score :)
> >
> > I hope the authors will clarify the notion of "independence assumption" and the class of models in the final version.

---

> > > ### Author Response · Authors · 2025-08-04
> > >
> > > We thank the reviewer for their positive response! We confirm we will incorporate the clarifications regarding the independence assumption and class of models in our manuscript.
> > >
> > > Concerning the ECE metric, we note that we want to minimise it over the concepts, therefore before any constraint takes place. We will clarify this in the text.

---

### Official Review · Reviewer_cdKS · 2025-07-02

**Clarity:** 3
**Significance:** 3
**Originality:** 3
**Rating:** 5
**Confidence:** 3

**Summary:**

The paper introduces Neurosymbolic Diffusion Models (NeSy-DMs), which combine discrete diffusion processes with symbolic reasoning to model dependencies between symbolic concepts more effectively. Unlike traditional NeSy models that assume conditional independence among concepts, NESYDMs leverage a denoising diffusion framework to sample structured symbolic predictions iteratively. This allows them to handle uncertainty better and generalize more robustly to ambiguous or out-of-distribution data. The authors provide a principled derivation of the training objective, along with a thorough theoretical analysis of its NELBO formulation. Experiments on visual reasoning and path-planning tasks demonstrate significant improvements in both accuracy and calibration.

**Questions:**

Given the weaknesses above, I would ask:

- How would NESYDMs behave on tasks whose outputs are not easily factorised and whose programs lack a compilable circuit? A deeper discussion on this matter would clarify better the scope and the limitations of the approach.

- How do the authors envision applying NESYDMs to reasoning-heavy tasks like visual question answering, where the symbolic structure is more complex and less explicit?

- Since L_y , L_c , and L_H are critical to performance (L234), what is the effect of varying these weights? Can the authors provide ablation studies or sensitivity analysis?

**Ethical Concerns:**

["NO or VERY MINOR ethics concerns only"]

**Final Justification:**

The authors provided a thorough explanation and conducted additional experiments that addressed my initial concerns. Their rebuttal was clear and responsive, and I recommend incorporating the new material into the manuscript. Based on this, I have increased my score.

**Limitations:**

yes

**Paper Formatting Concerns:**

No major formatting issues.

**Quality:**

3

**Strengths And Weaknesses:**

Strengths:

- The paper tackles the independence assumption flaw that limits existing NeSy predictors and presents a principled and theoretically grounded solution.

- The paper proposes a theoretically grounded explanation of how the local independence assumption within diffusion steps balances scalability and expressiveness. A continuous-time NELBO is proved (Thm 3.1) and linked to masked diffusion losses, providing a solid learning objective

- The paper is clear and well organized.

Weaknesses:

- The training objective (Eq. 5) requires either (i) a decomposable output space or (ii) a compiled circuit to evaluate weighted model counts. What would happen in tasks where neither is available? How do NESYDMs address such problems?

- The experiments focus mostly on visual planning and rule-based driving. It would be beneficial for the paper if the authors discussed how NESYDMs could be used for further tasks that require reasoning, such as visual-question answering.

- The loss-component weights are "critical to the performance" as reported by the authors in L234. The paper would benefit from ablation studies for the terms and the hyper-parameter sensitivity analysis of the components of the loss term.

---

> ### Author Rebuttal · Authors · 2025-07-31
>
> We thank the reviewer for their helpful comments and feedback, and are glad they found our paper **clear and well-organized**. They believe NeSyDM is a **theoretically grounded solution** with a solid learning objective, while providing **significant experimental improvements**.
>
> The reviewer had several important questions on the fit of NeSyDMs to other tasks and applications, which we discuss in our response below. They also gave the valuable suggestion to **include an ablation study for the weighting of our loss components**, which we ran and also provide below. We will add these clarifications and new results to our manuscript.
>
> > How would NESYDMs behave on tasks whose outputs are not easily factorised and whose programs lack a compilable circuit? A deeper discussion on this matter would clarify better the scope and the limitations of the approach.
>
> **Yes, this also is a challenge for _all_ our NeSy competitors. However, the strength of NeSyDM lies in its compatibility with approximate inference methods for WMC [1, 2].** So, if one has a problem for which a good WMC approximation is available, this can be directly applied in NeSyDM by updating the gradient estimation procedure.
>
> In particular, we believe a significant scalability boost can be obtained by using pretrained models. For instance, Reinforcement Learning with Verifiable Rewards (RLVR) methods use pretrained LLMs that already have a large probability of obtaining a positive reward. Similarly, we can use pretrained diffusion models, such as LLADA or MMADA [3, 4], to bootstrap the learning process.
>
> We will include this discussion in the camera-ready version.
>
>
> > How do the authors envision applying NESYDMs to reasoning-heavy tasks like visual question answering, where the symbolic structure is more complex and less explicit?
>
> Indeed, when the symbolic structure is not explicit, we have the additional challenge of predicting a suitable symbolic structure. **Luckily, this can be neatly integrated with NeSyDMs.** For instance, we can use a pretrained model (again, like LLADA or MMADA [3, 4]) that can already generate programs ($\approx$symbolic structures). Then, we can evaluate the resulting program to predict the answer. If this aligns with the ground-truth answer, we can reinforce the probability of this program. This is similar to popular approaches like ViperGPT, VISPROG, Faithful CoT and Logic-LM [5-8], which can improve the quality of the generated program. We will include a discussion of this extension in our camera-ready.
>
> > Since L_y , L_c , and L_H are critical to performance (L234), what is the effect of varying these weights? Can the authors provide ablation studies or sensitivity analysis?
>
> **Yes we can provide those, see below!** We ran the conditioned model NeSyDM_Cond with 3 repeated runs on MNISTHalf with various values of the entropy weight and concept unmasking weight to understand their effect on the various metrics. We will include these results and our analysis in our appendix.
>
> L234 is somewhat imprecise, we only experimented with the concept unmasking weight and the variational entropy weight, keeping the output unmasking weight to 1 (affected only by learning rate). We will update L234 and refer to the new appendix.
>
> | Concept unmasking weight | Label accuracy ↑  | Concept accuracy ↑| Label accuracy OOD ↑| concept accuracy OOD ↑ | ECE ID ↓ | ECE OOD ↓ |
> |------------------|------------|-----------|-------------|------------|----------|-----------|
> | 1e-08 | 99.38±0.27 | 70.45±2.32 | 33.92±5.10 | 65.13±2.72 | 8.54±4.46 | 12.23±1.18 |
> | 1e-07 | **99.61±0.13** | 71.41±0.72 | **37.16±1.22** | **67.74±0.56** | 7.77±1.09 | 10.67±0.98 |
> | 1e-06 | 99.54±0.80 | **72.88±0.85** | 33.21±7.06 | 64.86±3.92 | 5.31±1.19 | 11.00±0.87 |
> | 1.5e-06 (used) | 99.12±0.10 | 71.16±1.77 | 28.44±0.90 | 62.76±0.89 | 4.18±2.56                                      | 11.74±1.18 |
> | 1e-05 | 99.00±0.53 | 69.79±3.09 | 29.72±3.00 | 63.10±2.11 | 6.33±3.87 | 11.39±1.76 |
> | 0.0001 | 99.23±0.13 | 72.07±0.77 | 36.46±6.60 | 66.84±4.04 | 4.90±0.69 | **10.22±1.96** |
> | 0.001 | **99.61±0.13** | 72.03±0.84 | 32.75±6.51 | 64.60±4.44 | 4.98±3.15 | 10.46±2.86 |
> | 0.01 | 99.15±0.48 | 71.60±0.35 | 31.99±7.80 | 63.57±5.41 | **3.80±1.52** | 10.81±1.07 |
> | 0.1 | 99.46±0.35 | 71.88±0.81 | 36.76±8.27 | 66.85±5.05 | 6.96±2.35 | 12.35±4.01 |
> | 1.0 | 98.84±0.61 | 41.55±0.12 | 5.70±0.24 | 38.65±0.14 | 56.87±0.25 | 61.09±0.11 |
> | 10.0 | 99.38±0.13 | 41.59±0.13 | 5.64±0.19 | 38.59±0.19 | 57.00±0.23 | 60.97±0.08 |
>
>
> For the output unmasking weight  $\gamma_{\mathbf{c}}$, we find that all test values achieve high label accuracy. However, its value significantly influences the concept accuracy both in-distribution and OOD, and the OOD label accuracy. We observe lower values (<1.0) are effective, which is in line with our observation on L947. We suspect the output unmasking loss can provide the model information that it is useful, but it should not dominate the loss, as this results in significantly lower concept accuracy and OOD performance, and very poor calibration. This suggests the model converged onto a single reasoning shortcut.
>
> | Entropy weight | Label accuracy ↑  | Concept accuracy ↑| Label accuracy OOD ↑| concept accuracy OOD ↑ | ECE ID ↓ | ECE OOD ↓ |
> |------------------|------------|-----------|-------------|------------|----------|-----------|
> | 0.001 | 99.15±0.13 | 41.63±0.07 | 5.61±0.16 | 38.63±0.03 | 57.05±0.12 | 61.08±0.16 |
> | 0.01 | 99.00±0.35 | 41.74±0.18 | 5.79±0.18 | 38.66±0.03 | 56.80±0.21 | 60.91±0.11 |
> | 0.1 | 98.77±0.48 | 41.67±0.12 | 5.61±0.32 | 38.60±0.23 | 56.98±0.26 | 60.97±0.09 |
> | 0.5 | 99.31±0.23 | 41.63±0.07 | 5.58±0.14 | 38.59±0.15 | 56.94±0.09 | 61.09±0.01 |
> | 1.0 | 99.23±0.13 | 41.63±0.13 | 5.85±0.14 | 38.73±0.12 | 56.90±0.25 | 61.06±0.09 |
> | 1.3 | **99.77±0.23** | 67.05±1.99 | **35.51±2.78** | **65.79±1.79** | 9.33±1.74 | 12.26±1.31 |
> | 1.6 (used)             | 99.12±0.10                        | 71.16±1.77                        | 28.44±0.90                                   | 62.76±0.89                                   | 4.18±2.56                                      | 11.74±1.18                                     |
> | 2.0 | 99.61±0.13 | **72.26±0.41** | 29.35±1.73 | 62.19±1.03 | **3.79±0.48** | **11.42±0.61** |
> | 3.0 | 89.81±5.84 | 46.53±2.50 | 17.03±8.97 | 51.23±7.03 | 12.70±5.87 | 14.56±2.71 |
> | 5.0 | 85.73±1.94 | 39.74±1.33 | 11.24±0.14 | 44.04±1.39 | 12.56±1.33 | 24.22±2.12 |
> | 10.0 | 85.73±0.35 | 34.22±1.27 | 10.81±0.19 | 41.25±1.03 | 15.40±2.25 | 24.01±1.56 |
>
> We also observe a significant influence of the entropy weight $\gamma_{\text{H}}$. All values <1.3 seem to converge on a single reasoning shortcut, exhibiting very poor calibration and worse concept accuracy. These runs do not balance the maximisation of entropy as the weight is too low, resulting in models with low entropy. Increasing the entropy weight beyond the value we used (1.6), we find that this can also impact the performance (>2.0), resulting in reduced label and concept accuracy and calibration. As the optimal range of values is quite tight, we recommend tuning $\gamma_{\text{H}}$ (as mentioned in L949).
>
> **Citations**
> 1. van Krieken, E., Thanapalasingam, T., Tomczak, J., Van Harmelen, F., & Ten Teije, A. (2023). A-nesi: A scalable approximate method for probabilistic neurosymbolic inference. Advances in Neural Information Processing Systems, 36, 24586-24609.
> 2. De Smet, L., Sansone, E., & Zuidberg Dos Martires, P. (2023). Differentiable sampling of categorical distributions using the catlog-derivative trick. Advances in Neural Information Processing Systems, 36, 30416-30428.
> 3. Nie, S., Zhu, F., You, Z., Zhang, X., Ou, J., Hu, J., ... & Li, C. (2025). Large language diffusion models. arXiv preprint arXiv:2502.09992.
> 4. Yang, L., Tian, Y., Li, B., Zhang, X., Shen, K., Tong, Y., & Wang, M. (2025). Mmada: Multimodal large diffusion language models. arXiv preprint arXiv:2505.15809.
> 5. Surís, D., Menon, S., & Vondrick, C. (2023). Vipergpt: Visual inference via python execution for reasoning. In Proceedings of the IEEE/CVF international conference on computer vision (pp. 11888-11898).
> 6. Gupta, T., & Kembhavi, A. (2023). Visual programming: Compositional visual reasoning without training. In Proceedings of the IEEE/CVF conference on computer vision and pattern recognition (pp. 14953-14962).
> 7. Lyu, Q., Havaldar, S., Stein, A., Zhang, L., Rao, D., Wong, E., ... & Callison-Burch, C. (2023, November). Faithful chain-of-thought reasoning. In The 13th International Joint Conference on Natural Language Processing and the 3rd Conference of the Asia-Pacific Chapter of the Association for Computational Linguistics (IJCNLP-AACL 2023).
> 8. Pan, L., Albalak, A., Wang, X., & Wang, W. Y. (2023). Logic-lm: Empowering large language models with symbolic solvers for faithful logical reasoning. arXiv preprint arXiv:2305.12295.

---

> > ### Comment · Reviewer_cdKS · 2025-08-05
> >
> > I thank the authors for their thorough explanation and the additional experiments, which have addressed my concerns. I encourage the authors to incorporate these into their manuscript. Based on the authors’ rebuttal, I will raise my score.

---

### Official Review · Reviewer_MBEg · 2025-07-02

**Clarity:** 4
**Significance:** 4
**Originality:** 4
**Rating:** 5
**Confidence:** 3

**Summary:**

Briefly summarize the paper and its contributions. This is not the place to critique the paper; the authors should generally agree with a well-written summary. This is also not the place to paste the abstract—please provide the summary in your own understanding after reading.

The authors propose neurosymbolic diffusion models (NeSyDMs), which are a masked-diffusion-based class of neurosymbolic (NeSy) predictors operating both in concept space and in output space.

Here, a neurosymbolic (NeSy) predictor is (informally) a model that extracts a concept $c$ from its input $x$ (normally with $c$ having conditionally independent coordinates; i.e. a “disentangled embedding” in some sense), and then feeds $c$ as input to a program $\phi$, obtaining a final output $y = \phi(c)$. The paper works with NeSy predictors in a probabilistic framework, so that concept extraction is seen as sampling from $p(c | x)$, and the posterior output distribution is $p(y|x) = \int 1_{\phi(c)=y} p_\theta(dc|x)$. This distribution is known as a weighted model count (WMC).

In practice, the conditional independence assumption on concepts $c$ can be limiting, as many realistic tasks can indeed have interactions between ground truth concepts, and are potentially solvable even if the learned input-to-concept mapping is spurious (“Reasoning Shortcuts”, RS). Prior work has tried to get around Reasoning Shortcuts by building NeSy predictors that express uncertainty across concepts consistent with the input-output mapping. However, correctly expressing this uncertainty can be at odds with maximizing $p(y|x)$ for conditionally independent concept extractors.

The authors address this limitation by proposing a NeSy architecture allowing for expressive inter-dependencies between the coordinates of $c$. A natural expressive model class for this is that of masked diffusion models. Their model has the following main components:

- A concept unmasking model $p_\theta(\tilde{c}^0 | c^s, x)$, which is essentially a masked diffusion model.
- A variational approximation $q_\theta(c^0 | y^0, x)$ of the distribution of concepts compatible with an input-output pair. This shares parameters with the concept unmasking model, except that sampling is done through controlled generation, conditioned on $y^0$.
- An output reverse process (Eq. 4), which marginalizes over (masked) concepts according to the concept unmasking model.

The authors derive a NELBO for their model, which consists of a concept unmasking loss term (essentially the MDM loss for $p_\theta(\tilde{c}^0 \mid c^s, x)$), an output unmasking loss (which requires computing weighted model counts), and a variational entropy term for $q_\theta(c^0 \mid y^0, x)$.

The output unmasking loss is non-trivial to compute, and requires using a REINFORCE Leave-One-Out estimator, which essentially involves sampling multiple concepts $\tilde{c}^0_1, \dots, \tilde{c}^0_S$ from the reverse diffusion model, computing a per-dimension binary reward $\mathbf{1}[\varphi(\tilde{c}^0_j)_i = y^0_i]$ for each sample, subtracting the mean reward over the other $S-1$ samples as a leave-one-out baseline, and weighting each sample’s reward by the score-function gradient $\nabla_\theta \log p_\theta(\tilde{c}^0_j \mid c^t, x)$ to obtain an unbiased, variance-reduced gradient estimate.

Computing the variational entropy term is also challenging, since $q_\theta$ is defined by performing controlled sampling on the concept unmasking model, with conditioning on $y^0$, which makes computing exact likelihoods intractable. The authors address this by working with 1-step entropies, i.e. using $T=1$ discretization steps in the diffusion process.

Sampling on a given input $x$ is done by simply denoising several concepts $c_i$ and taking a majority vote over the resulting outputs $y_i := \phi(c_i)$.

Experiments aim to answer (RQ1) whether NESYDM scales to high-dimensional reasoning and (RQ2) whether its expressiveness reduces reasoning shortcuts. For RQ1, NESYDM is compared to A-NeSI, Scallop, EXAL and I-MLE on multidigit MNIST addition and visual path planning (12×12 and 30×30 grids); it matches baseline accuracy on MNIST addition (92.49±0.98 for N=4, 77.29±1.40 for N=15) and exceeds others on 30×30 planning (97.40±1.23 vs ≤67.57±36.76). For RQ2, on the RSBench tasks (MNIST Half, MNIST Even-Odd, BDD-OIA) NESYDM is evaluated against DPL, Semantic Loss, DeepProbLog and BEARS, reporting concept and output accuracy plus ECE; it achieves comparable or higher accuracy with lower calibration error than these baselines. Overall, the results indicate that joint modeling via diffusion can scale without assuming conditional independence and produces more calibrated predictions under ambiguity.

**Questions:**

- I don’t have any major questions for the authors, and believe the exposition in the paper is already very thorough. Please feel free to point out e.g. any misunderstandings on my part in the summary.
- Out of curiosity, what was the authors’ process for constructing the presented formulation of NeSyDM’s? In particular:
  - Is it somehow unavoidable to denoise outputs in addition to concepts? If not unavoidable, did you try at any point in your exploration process to only use diffusion for the concepts? What were the limitations?
  - How did you arrive at the RLOO estimator as a solution to computing $\mathcal{L}_y$? As in, what was the reasoning process for trying this approach specifically?

**Ethical Concerns:**

["NO or VERY MINOR ethics concerns only"]

**Final Justification:**

My recommendation of acceptance follows from the points listed in **Strengths**. Also, the authors agreed to include a more "linear" summary of their method, addressing what I considered to me the main missing element for this paper.

**Limitations:**

The authors extensively discuss limitations and future work in the conclusion.

**Paper Formatting Concerns:**

None.

**Quality:**

4

**Strengths And Weaknesses:**

**Strengths:**
- The method is well-motivated by a very concrete research goal: removing the need for conditional independence in NeSy predictors to enable reasoning-shortcut-awareness and better uncertainty quantification.
- Extensive exposition of related background: as someone with a background in diffusion models but not neurosymbolic prediction, I found the authors did a good job of introducing both relevant lines of research and how they interact in their formulation of NeSyDMs.
- Significant technical novelty: in order to get NeSyDM training to actually be tractable, the authors architect a training process including non-obvious algorithmic tricks (e.g. RLOO estimator for the output unmasking loss). This is a significant value add to the body of research in the area, as re-discovering this kind of trick can be very hard and time-consuming, so that knowing the tricks in advance can significantly save time from research efforts which might have struggled with these tractability issues.
- Overall very high quality of presentation, exposition, formatting, and so on.

**Weaknesses:**
- The method is fairly complex and requires a lot of e.g. gradient approximation tricks in order to work. These tricks are explained separately in the main paper, but I believe it would facilitate understanding if the authors included a “linear” summary of what training and sampling look like at the end of the day when all components are put together.

---

> ### Author Rebuttal · Authors · 2025-07-31
>
> We thank the reviewer for their encouraging comments, and would like to confirm their summary is pretty spot-on! Furthermore, we are happy they thought our paper is **well-motivated**, discussing the related work properly while providing **extensive significant technical novelty**, and appreciate they believe **the exposition is thorough**.
>
> We will extend our paper to give a linear overview of the loss computation and sampling, and provide extra discussion on our RLOO gradient estimator, as suggested by the reviewer.
>
> > I believe it would facilitate understanding if the authors included a “linear” summary of what training and sampling look like at the end of the day when all components are put together.
>
> That is a great suggestion, thank you. **We will add this to our paper**.
>
> > Is it somehow unavoidable to denoise outputs in addition to concepts? If not unavoidable, did you try at any point in your exploration process to only use diffusion for the concepts? What were the limitations?
>
> **It is not unavoidable, but it makes the resulting algorithms more tricky.** If outputs are not denoised, they’d from a generative perspective only come in for completely clean data. This requires full simulation of the diffusion model during training, which is more akin to standard RL training in LLMs and diffusion. While this is theoretically possible, it is highly compute-intensive and has much a more challenging credit assignment problem: Ie, which of the many denoising steps were relevant to producing a good result?
>
> > How did you arrive at the RLOO estimator as a solution to computing ? As in, what was the reasoning process for trying this approach specifically?
>
> **RLOO is the simplest RL-based estimator to implement, really.** It scales well with number of samples, and is unbiased [1]. Recent work in RLHF also shows it is a great, simple alternative to, e.g., PPO [2] and it is closely related to (but simpler than) the currently very popular GRPO algorithm [3]. For more expensive symbolic programs, one could also consider LOOP [4], a combination of PPO and RLOO. We will add this discussion to our paper.
>
> 1. Kool, Wouter, Herke van Hoof, and Max Welling. "Buy 4 reinforce samples, get a baseline for free!." (2019).
> 2. Schulman, J., Wolski, F., Dhariwal, P., Radford, A., & Klimov, O. (2017). Proximal policy optimization algorithms. arXiv preprint arXiv:1707.06347.
> 3. Shao, Z., Wang, P., Zhu, Q., Xu, R., Song, J., Bi, X., ... & Guo, D. (2024). Deepseekmath: Pushing the limits of mathematical reasoning in open language models. arXiv preprint arXiv:2402.03300.
> 4. Gupta, S., Ahuja, C., Lin, T. Y., Roy, S. D., Oosterhuis, H., de Rijke, M., & Shukla, S. N. (2025). A simple and effective reinforcement learning method for text-to-image diffusion fine-tuning. arXiv preprint arXiv:2503.00897.

---

> > ### Comment · Reviewer_MBEg · 2025-08-04
> >
> > Thank you for your reply and for clarifying how you arrived at the RLOO estimator! I agree that including this in the paper will make it more intuitive to readers, and also more instructive to practitioners regarding considerations for picking a baseline RL estimator to try in their training setups.
> >
> > I maintain my recommendation of acceptance.

---

### Official Review · Reviewer_N1ag · 2025-07-03

**Clarity:** 3
**Significance:** 2
**Originality:** 3
**Rating:** 4
**Confidence:** 1

**Summary:**

Disclaimer: I am no expert in neurosymbolic models or diffusion models. I am unsure why I was assigned this paper and flagged this to the AC.

The paper proposes to use discrete diffusion models to overcome the conditional independence assumption commonly made for context variables in neurosymbolic models.
The paper proposes to do this by having a diffusion model to have global dependencies between concepts.
The work then derives the update rules and loss functions, before empirically evaluating e.g. on MNIST classification.

**Questions:**

See weaknesses

**Ethical Concerns:**

["NO or VERY MINOR ethics concerns only"]

**Final Justification:**

nothing to be added

**Limitations:**

yes

**Quality:**

2

**Strengths And Weaknesses:**

Strengths:
- The work seems well motivated
- The paper is clearly written, technically sound, and can be followed
- The work shows clear empirical improvements over prior work
- Code is attached

Weaknesses:
- The experimental validation is limited to few tasks
- The largest problem studied is MNIST classification. To me, these experiments do not show the "scalability" of the approach, as minst classification is a rather small-scale problem.

---

> ### Author Rebuttal · Authors · 2025-07-31
>
> We thank the reviewer for their comments and feedback, and appreciate they believe our work is well-motivated and clear, and technically sound.
>
> >The largest problem studied is MNIST classification.
> > To me, these experiments do not show the "scalability" of the approach, as minst classification is a rather small-scale problem.
>
> **The scalability described here is regarding reasoning.** Our method also significantly surpasses baseline methods on large reasoning problems like visual path planning that are challenging for current NeSy methods [1-3]. Furthermore, we also study real-world images with the BDD-OIA dataset [4].
>
> > The experimental validation is limited to few tasks
>
> We are very happy to add additional tasks if the reviewer has a concrete list in mind.
>
> 1. van Krieken, E., Thanapalasingam, T., Tomczak, J., Van Harmelen, F., & Ten Teije, A. (2023). A-nesi: A scalable approximate method for probabilistic neurosymbolic inference. Advances in Neural Information Processing Systems, 36, 24586-24609.
> 2. Verreet, V., De Smet, L., De Raedt, L., & Sansone, E. (2024). EXPLAIN, AGREE, LEARN: Scaling Learning for Neural Probabilistic Logic. AAAI 2025.
> 3. Maene, J., Derkinderen, V., & Martires, P. Z. D. (2024). Klay: Accelerating arithmetic circuits for neurosymbolic ai. ICLR 2025.
> 4. Xu, Y., Yang, X., Gong, L., Lin, H. C., Wu, T. Y., Li, Y., & Vasconcelos, N. (2020). Explainable object-induced action decision for autonomous vehicles. In Proceedings of the IEEE/CVF Conference on Computer Vision and Pattern Recognition (pp. 9523-9532).

---

> > ### Comment · Reviewer_N1ag · 2025-08-03
> >
> > As pointed out in my review, I am no expert in the field and cannot recommend additional tasks.
> > I maintain my score.

---

### Note · Authors · 2025-08-15

We thank the reviewers for their in-depth reviews. We appreciate that the reviewers agree that the paper is **clearly written**, **easy to follow** (N1ag03, MBEg, cdKS) and **well-organised** (MBEg02, cdKS), with reviewer MBEg mentioning the **‘exposition in the paper is very thorough’**. They believe the paper is **well motivated** (N1ag03, MBEg) and **provides a principled, theoretically grounded solution** (N1ag03, cdKS) with **significant technical novelty** (MBEg, QdAB). Furthermore, it provides **clear empirical improvements** over prior work (N1ag03, MBEg, cdKS), while discussing limitations thoroughly.

During the discussion, we **resolved reviewer QdAB’s question** about the benefit of Neurosymbolic Diffusion Models (NeSyDMs) compared to increasing the number of parameters in weighted model counting (WMC). We highlighted how NeSyDMs use neural networks that predict the same number of parameters as baseline SotA NeSy methods that independence over concepts. Instead, increasing the number of WMC parameters also requires larger neural networks. This allows it to **scale to more challenging reasoning problems**. Reviewer QdAB also asked for a motivation for the ECE measure in studying Reasoning Shortcut Awareness. We motivated this measure in this context and provide an example. **We will add both clarifications to our manuscript.**

Furthermore, we will **extend our paper using the valuable suggestions of the reviewers**. First, we will add a linear overview of the loss computation and sampling (MBEg). We will also add our new ablation study on the impact of the weighting of the different loss components (cdKS), clarified our RLOO gradient estimator for reviewer MBEg and include further discussion on future work and applications of NeSyDMs (cdKS). In particular, reviewer cdKS mentioned their concerns were addressed.

---

### Decision · Program_Chairs · 2025-09-17

**Decision:**

Accept (poster)

**Comment:**

(a) This paper introduces Neuro-symbolic Diffusion Models (NESYDMs), a new class of NeSy predictors that drop the restrictive independence assumption while staying scalable. The key idea is to use discrete masked diffusion to model dependencies over symbolic concepts, while integrating the symbolic program directly into both the reverse process and the loss. The authors derive a continuous-time NELBO that incorporates a program-aware weighted model count term, and they prove a new result extending diffusion NELBOs to non-factorized unmasking distributions. Empirical results are demonstrated on hard reasoning problems including path planning and RSBench.

(b) The derivation of a program-aware diffusion NELBO and the new theoretical extension to non-factorized joints is a clear and makes sense. The integration of symbolic programs directly inside the reverse process, variational posterior via constrained resampling, and gradient estimation via decomposed RLOO go beyond MDM + some penalty. The method outperforms baselines in high-dimensional reasoning (30×30 path planning) and improves calibration/RS-awareness on various datasets.

(c) Personally I think the experimental results are okay but needs to demonstrate on more realistic and challenging reasoning problems, where it is difficult to solve the problem otherwise. Empirical validation is limited to RSBench and path-planning - broader real-world tasks are needed for wider impact.

(d) The paper does address a fundamental independence assumption limitation of NeSy. So for everyone interested in this problem domain, the assumptions and experimental results should be interesting.

(e)
- One reviewer raised whether NeSyDMs truly add value beyond just scaling up weighted model counting with more parameters. The authors clarified that NeSyDMs predict the same number of parameters as standard NeSy baselines, but can scale to larger reasoning problems without requiring exponentially larger neural networks

- A second concern was about using ECE as the measure for reasoning-shortcut awareness. The authors gave a concrete example showing how calibration reflects whether probability mass is spread correctly over ambiguous concepts.

- There were also two side issues: one reviewer flagged potential AI-assistance in a review (handled separately, no impact on this paper), and another noted they weren’t an expert in this area, which limited their feedback but didn’t affect the consensus.